# No Free Lunch: Non-Asymptotic Analysis of Prediction-Powered Inference

**Pranav Mani** [1]   **Peng Xu** [1]   **Zachary Lipton** [2 1]   **Michael Oberst** [3 1]

## Abstract

Prediction-Powered Inference (PPI) is a popular strategy for combining gold-standard and possibly noisy pseudo-labels to perform statistical estimation. Prior work has shown an asymptotic "free lunch" for PPI++, an adaptive form of PPI, showing that the *asymptotic* variance of PPI++ is always less than or equal to the variance obtained from using gold-standard labels alone. Notably, this result holds *regardless of the quality of the pseudo-labels*. In this work, we demystify this result by conducting an exact finite-sample analysis of the estimation error of PPI++ on the mean estimation problem. We give a "no free lunch" result, characterizing the settings (and sample sizes) where PPI++ has provably worse estimation error than using gold-standard labels alone. Specifically, PPI++ will outperform if and only if the correlation between pseudo- and gold-standard is above a certain level that depends on the number of labeled samples ($n$). In some cases our results simplify considerably: For Gaussian data, for instance, the correlation must be at least $1/\sqrt{n-2}$ in order to see improvement. More broadly, by providing exact non-asymptotic expressions for the variance of PPI++ under sample splitting, we aim to empower practitioners to transparently reason about the benefits of PPI++ in specific applications. In experiments, we illustrate that our theoretical findings hold on real-world datasets.

## 1. Introduction

Mean estimation is an ever-present problem: Medical researchers wish to understand the prevalence of disease, machine learning engineers want to understand the average performance of their models, and so on. Often, practitioners have access to a large set of unlabeled examples, e.g., clinical notes that may or may not record the presence of a disease, images with unknown labels, etc. A typical approach is to gather (expensive) gold-standard labels for a small number of randomly selected examples, and take a simple average of the relevant metric across them.

In many scenarios, it is possible to construct cheap pseudo-labels using machine learning models or other heuristics. For instance, large language models (LLMs) and vision-language models (VLMs) can achieve reasonable "zero shot" performance on a variety of tasks (Radford et al., 2021). These pseudo-labels have the potential to improve estimation: For instance, if they were perfect substitutes for gold-standard labels, they would dramatically expand our sample size without introducing bias. In fact, it is already becoming a routine practice to use pseudo-labels from LLMs in "LLM-as-Judge" evaluations (Rahmani et al., 2025).

However, the quality of these pseudo-labels is often unknown a priori, and the naive approach of treating them as true standard labels can lead to erroneous conclusions. Prediction-Powered Inference (PPI) (Angelopoulos et al., 2023a) proposes an estimator that is unbiased for any set of pseudo-labels, by using the small labeled dataset (on which pseudo-labels can also be obtained) to estimate and correct for any pseudo-label bias. While PPI is unbiased, it is not guaranteed to improve variance / statistical efficiency, particularly when the pseudo-labels have weak correlation with the true labels. To this end, Angelopoulos et al. (2023b) proposed Power-Tuned PPI (PPI++), which uses the labeled dataset to infer the pseudo/true label correlation and discount the pseudo-labels if this correlation is small. Informed by analysis showing that the (asymptotic) variance of PPI++ is never greater than the variance of the *classical* estimator that only uses labeled examples, they note (emphasis added)

> *[our] methods automatically adapt to the quality of available predictions, yielding easy-to-compute confidence sets... that **always improve on classical intervals using only the labeled data.*** (Angelopoulos et al., 2023b)

However, this guarantee is asymptotic in nature, applying in the regime where the number of labeled examples becomes

---

[1]Abridge AI, San Francisco, CA, USA [2]Machine Learning Department, Carnegie Mellon University, Pittsburgh, PA, USA [3]Department of Computer Science, Johns Hopkins University, Baltimore, MD, USA. Correspondence to: Michael Oberst <moberst@jhu.edu>.

*Proceedings of the $43^{rd}$ International Conference on Machine Learning*, Seoul, South Korea. PMLR 306, 2026. Copyright 2026 by the author(s).

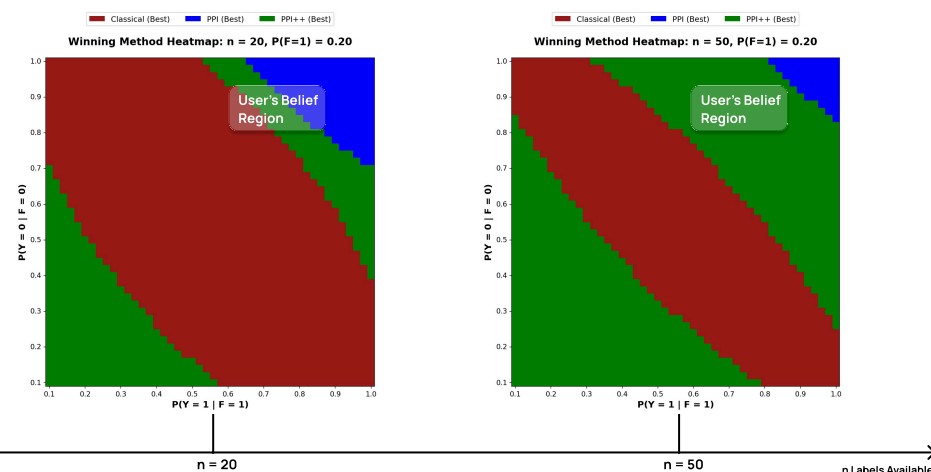

*Figure 1.* An illustrative example of how our analytical results can guide practical decision-making. For any user-specified belief regarding the quality of pseudo-labels, we can determine analytically whether or not PPI++ (with cross-fitting, see Definition 3.5) will improve over classical estimation or PPI. Specifying the quality of pseudo-labels $F$ can be done in the binary setting using three parameters: The prevalence $P(F = 1)$ and the positive / negative predictive value $P(Y = 1 \mid F = 1), P(Y = 0 \mid F = 0)$. For any value of these three parameters, and the sample size of labels $n$, we can compute the variance of PPI & PPI++ relative to the variance of the classical approach. For modest sample sizes, there are distinct regions (color coded) where each method performs best. Given *a priori* practitioner beliefs about plausible accuracy of the pseudo-label (the "belief region"), this visualization allows for an informed choice as to which method to use: At $n = 20$ (left), there is no dominant method, while at $n = 50$ (right), PPI++ is superior across the belief region.

large, while the highest-value use of PPI++ comes precisely when labels are scarce. This leaves an open question: If the labeled sample is already too small to give a reliable estimate of the mean, when can we trust its assessment of the pseudo-labels?

We resolve this question by **providing an exact non-asymptotic expression for the finite-sample variance of PPI++** under a sample-splitting implementation, and in the regime where the unlabeled dataset is taken to be arbitrarily large. Our expression holds for any joint distribution of pseudo- and true labels that has bounded fourth moments.

This analysis yields practical insights. First, our analysis produces a "no free lunch" result: PPI++ pays a price attributable to errors in estimating the correlation between pseudo- and true labels. If this correlation is small, then the costs of PPI++ outweigh the benefits. From our general results, we derive expressions for the *minimum number of labeled samples* that are required for PPI++ to improve upon the classical estimator. While our results apply to any distribution, they simplify considerably for Gaussian and binary labels. For Gaussian labels, the correlation must be larger than $(n - 2)^{-1/2}$ for PPI++ to improve over the classical estimator, where $n$ is the labeled sample size.[1]

---

[1]As discussed in Corollary 4.1, this threshold is for an implementation without sample splitting. In the Gaussian setting with sample-splitting and cross-fitting, the threshold is slightly higher, at $(n/2 - 2)^{-1/2}$.

Second, our results allow for direct calculations of the benefit / downside of using PPI vs PPI++ vs. classical estimators, depending on the assumed quality of pseudo labels and available size of labeled samples. Thus, practitioners can operationalize our framework into a recommended course of action by applying broad assumptions of the region in which the quality of their pseudo-labels lie. This process is described in Figure 1. However, absent any assumption of the quality of the pseudo-labels, our method does not purport to identify the best performing method: that would fall into the same free-lunch trap that we caution against. Our conclusion is that PPI++ is a useful tool, but requires practitioners to implicitly assume that their pseudo-labels are "good enough", a notion we define precisely in this work. Our overall contributions can be summarized as follows:

- In Section 4.1, we analyze PPI++ in the Gaussian setting, and provide interpretable bounds on the correlation/sample size required for PPI++ to outperform the classical estimator that only uses the labeled sample.

- In Section 4.2, we analyze PPI++ with sample splitting and cross-fitting, for general joint pseudo/true label distributions. We provide an exact expression for the variance of this unbiased estimator with finite labeled samples, and showing that similar intuitions hold as in the Gaussian case.

- In Section 4.3, we return to PPI++ without sample splitting, showing that it has two undesirable properties: It

is biased, and standard variance-estimation approaches can produce overly optimistic estimates of its variance.

- In Section 5, we illustrate our findings on real data, highlighting settings where the use of PPI++ hurts variance, and show that PPI++ without sample splitting tends to draw overly narrow confidence intervals.

## 2. Related Work

The term "Prediction-Powered Inference" was coined by Angelopoulos et al. (2023a), who introduced the method in the context of both mean estimation and more general inference tasks. Recent work has proposed various extensions to PPI and PPI++, including their use with multiple estimands using additional shrinkage (Li & Ignatiadis, 2025), their use with cross-fitting of predictors (Zrnic & Candès, 2024b), and their use in the context of active statistical inference (Zrnic & Candès, 2024a). Others have proposed the use of PPI for application in clinical trials (Poulet et al., 2025), causal inference (Demirel et al., 2024), social science research (Broska et al., 2024), and the efficient evaluation of AI systems (Boyeau et al., 2024; Gligoric et al., 2025).

PPI shares similarities with doubly-robust estimators in causal inference, as noted by Zrnic & Candès (2024a), and can be seen as a special case of the Augmented Inverse Propensity Weighted (AIPW) estimator (Bang & Robins, 2005). Through this lens, the unbiased nature of PPI (even when pseudo-labels are uninformative) is mirrored by the consistency of AIPW estimators under incorrect outcome-regression models. The core idea of PPI++ (taking linear combinations of *unbiased* estimators to minimize variance) dates back at least as far as Bates & Granger (1969). However, the theoretical analysis of these estimators is often limited to potentially loose non-asymptotic error bounds (Lavancier & Rochet, 2016) or study of their asymptotic efficiency. A distinct but related problem arises in the study of combining *biased* and *unbiased* estimators, where the bias of the former can be estimated from data. In that setting, a similar "no free lunch" phenomenon has been observed, where these methods improve asymptotic variance for any fixed bias, but perform poorly when the bias is comparable in magnitude to $1/\sqrt{n}$ (Yang et al., 2023; Chen et al., 2021; Oberst et al., 2022). While our analysis strategy and problem setting differs substantially, our core result shows a conceptually similar point, that PPI++ tends to under-perform the classical estimator when the correlation between pseudo-labels and true labels is smaller than $\approx 1/\sqrt{n}$.

PPI++ can also be seen as a control variate method, where an unbiased estimator using $n$ samples is additively combined with a mean-zero, correlated control variable to reduce variance. In the control variate literature, it is well known that estimating the optimal coefficient for the control vari-

ate from the same $n$ samples can lead to variance inflation. For instance, Lavenberg & Welch (1981) give a variance-inflation factor for Gaussian data which corresponds precisely to the threshold for improvement that we recover in our "warm up" on the Gaussian setting in Section 4.1 for the single-sample PPI++ estimator with infinite $N$. We go further, however, in providing thresholds in the Gaussian setting for both a cross-fit variant of PPI++ and the finite $N$ case. Sample-splitting is also proposed as a "remedy" in that literature for bias and over-optimistic variance estimates that can arise from re-using the same sample (Nelson, 1990), analogous to the cross-fitting variant of PPI++ we analyze. However, to our knowledge, prior work characterizes variance only under the simplified setting of Gaussian data or asymptotically, while our main contribution (Section 4.2) is an exact, distribution-free finite-sample expression.

Close connections exist between PPI, PPI++, and estimators from the survey sampling literature. In particular, as noted by Mozer (2026), the PPI estimator for the mean is equivalent to the *difference estimator*, and the PPI++ estimator is equivalent to the generalized regression estimator (GREG) (Cassel et al., 1976; Särndal et al., 1989). Both estimators arise from a similar motivation (in survey sampling) to the motivation of PPI: The availability of *auxiliary information* (i.e., covariates) on many unsurveyed (i.e., unlabeled) units, and the use of predictive models to incorporate that information has a long history (Breidt & Opsomer, 2017).

However, despite the depth of literature on the topic of similar estimators across causal inference, control variate methods, and survey sampling, analysis of performance is often asymptotic in nature, with the characterization of the exact finite-sample variance (or mean squared estimation error) perceived to be intractable. For instance, Tille (2020), a textbook on survey sampling estimators, states (see Chapter 9.4., page 203) that "It is not possible to accurately calculate the expectation and variance of the regression estimator" before introducing standard approximations. We perform that seemingly intractable exact calculation in our work, using the insight that this analysis becomes tractable precisely in the setting that motivates PPI++, where the model generating pseudo-labels is fixed, and an extremely large unlabeled dataset is available.

The theoretical characterization of PPI++ in other related work is largely limited to characterizing *upper bounds* on the benefits. Dorner et al. (2024) studies the limits of the benefits of PPI++, but focus on establishing upper bounds on the theoretical benefit, in the (optimistic) scenario where the optimal weight is chosen exactly. Likewise, Chaganty et al. (2018), whose proposed method is largely identical to PPI++ in the mean estimation setting, do not consider finite-sample behavior of their estimator beyond establishing that the bias is $O(1/n)$.

Also of interest is Cross-PPI (Zrnic & Candès, 2024b) (distinct from the cross-fitting variant of PPI++ discussed in this work) which uses a portion of the data to train a predictor. We find there is no need to "demystify" the performance claim of Cross-PPI, since Zrnic & Candès (2024b) already gives some sense of the trade-offs that come with poor pseudo-labels. In contrast, PPI++ stands out for its claim to "always" do no worse than classical, even if the predictions are poor. That said, if there existed a version of Cross-PPI with a similar *"power-tuning"* component, we believe our core insight (no free lunch) would carry over.

## 3. Setup and Preliminaries

**Notation**: We use upper-case letters to denote a random variable (e.g., $Y$) and lower-case letters to denote a particular value (e.g., $y$). We use $P_Y$ to denote the distribution of a variable $Y$. We denote the mean of a random variable $Y$ by $\mu_y$, the variance by $\sigma_y^2$, and the covariance between two random variables $Y, X$ by $\sigma_{yx}$ or $\sigma_{xy}$. Given $n$ observations $\{x_i, y_i\}_{i=1}^n$ of two random variables $X, Y$, the empirical covariance is given by $\hat{\sigma}_{xy} = \widehat{\text{Cov}}(X, Y) = \frac{1}{n-1} \sum_{i=1}^n (x_i - \bar{x}_n)(y_i - \bar{y}_n)$ where $\bar{x}_n$ and $\bar{y}_n$ are the sample means of $x$ and $y$ respectively over $n$ samples.

**Problem Setting**: We focus on the *mean estimation* problem: Our goal is to estimate the expectation of a variable (or "label") $Y$, denoted $\theta := \mathbb{E}[Y]$.[2] We have access to a dataset of size $n$ (typically small), containing both covariates $X$ and true labels $Y$. Meanwhile, we have access to a dataset of size $N$ (typically much larger), which only contains unlabeled examples $X$. In this setting, we make the standard assumption that both samples are drawn from a common distribution. This assumption holds in the common scenario where a random subset of the entire dataset is chosen for labeling. We assume the ability to obtain cheap pseudo-labels $F$ on both the labeled and unlabeled datasets. One way to obtain such labels is via a function $f(X)$ (e.g., querying an LLM) that can be queried to get pseudo-labels, which motivates our use of upper-case $F$ to denote these pseudo-labels. The *classical* estimator simply takes the average of the labeled examples to estimate the mean.

**Definition 3.1** (Classical Estimator). Let $\{y_i\}_{i=1}^n$ be $n$ samples drawn independently from $P_Y$, where the mean of $Y$ is given by $\theta := \mathbb{E}[Y]$ and the variance by $\sigma_y^2 := \mathbb{E}[(Y - \theta)^2]$. The classical estimator of $\theta$ is the sample average $\hat{\theta}_{\text{Classical}} = \frac{1}{n} \sum_{i=1}^n y_i$ and is unbiased with variance $\sigma_y^2/n$.

**Prediction Powered Inference**: Prediction Powered Inference is a family of estimators that combine the labeled dataset of $n$ samples $\mathcal{D}_n = \{y_i, f_i\}_{i=1}^n$ with the larger set of $N$ pseudo-label samples $\{f_j\}_{j=1}^N$. For mean estimation,

---

[2] We use $\theta$ here to align with notation in the PPI literature, but note that $\theta$ and $\mu_y$ are the same quantity.

the original PPI estimator takes the following form.

**Definition 3.2** (Prediction Powered Inference (PPI) (Angelopoulos et al., 2023a)). Let $\mathcal{D}_n$ denote $n$ samples of $F, Y$ drawn from $P_{FY}$, and let $\mathcal{D}'_N$ denote $N$ samples of $F$ drawn from $P_F$. The PPI estimator is then given by $\hat{\theta}_{\text{PPI}} = \frac{1}{n} \sum_{i \in \mathcal{D}_n} y_i + \frac{1}{N} \sum_{j \in \mathcal{D}'_N} f_j - \frac{1}{n} \sum_{i \in \mathcal{D}_n} f_i$

Notice that PPI is unbiased for any $f$. That is, $\mathbb{E}[\hat{\theta}_{\text{PPI++}}] = \mathbb{E}[Y] + \mathbb{E}[F] - \mathbb{E}[F] = \mathbb{E}[Y]$. However, the variance of PPI depends on the variance of $Y - F$, which can be worse than the variance of $Y$ alone if $F$ is not sufficiently correlated with $Y$. Motivated by this concern, the PPI++ estimator (Angelopoulos et al., 2023b) uses a data-driven parameter $\lambda$ to weight the contribution of the pseudo-labels. For mean estimation, the PPI++ estimator takes the following form.

**Definition 3.3** (Power Tuned PPI (PPI++) (Angelopoulos et al., 2023b)). Let $\mathcal{D}_n, \mathcal{D}'_N$ be defined as in Definition 3.2. The PPI++ estimator for mean estimation is then given by the following $\hat{\theta}_{\text{PPI++}} = \frac{1}{n} \sum_{i \in \mathcal{D}_n} y_i + \hat{\lambda} \left( \frac{1}{N} \sum_{j \in \mathcal{D}'_N} f_j - \frac{1}{n} \sum_{i \in \mathcal{D}_n} f_i \right)$ with $\hat{\lambda} = \widehat{\text{Cov}}(Y, F)/((1 + \frac{n}{N})\widehat{\text{Var}}(F))$

Note that if $\lambda = 0$, then PPI++ reduces to Classical, and if $\lambda = 1$, then PPI++ reduces to PPI. Angelopoulos et al. (2023b) note that the variance-optimal choice of $\lambda^*$ is given by $\lambda^* = (\sigma_{fy}/((1 + \frac{n}{N})\sigma_f^2)$, which motivates the "plug-in" estimator of $\hat{\lambda}$ above. We make two observations: First, the optimal value $\lambda^*$ increases with higher covariance between the pseudo-labels and the true labels ($\sigma_{fy}$), and with lower variance in the pseudo-labels ($\sigma_f^2$). Both quantities need to be estimated from data, but the covariance in particular needs to be estimated using *labeled data*: Notably, the true labels $Y$ are used for both (a) estimating the mean, and (b) estimating the quality of the pseudo-labels.

In many practical applications, the unlabeled sample size $N$ is extremely large, and in our work, we consider it to be effectively infinite, such that $\sigma_f^2$ and $\mu_f$ can be estimated with arbitrary precision, and the optimal tuning parameter $\lambda^*$ is given by $\sigma_{yf}/\sigma_f^2$. Notably, even when $N$ is infinite, PPI++ still relies heavily on the small labeled sample of size $n$, which in practice may be very small. We now describe two variants of PPI++, which vary in how they use the labeled data to obtain their estimate of the optimal $\lambda$. The first approach is to use the entire labeled sample for both (a) estimating $\hat{\lambda}$, and (b) estimating $\hat{\theta}$, which we refer to as Single Sample PPI++.

**Definition 3.4** (Single-Sample PPI++, Infinite $N$). Let $\mathcal{D}_n$ denote $n$ samples of $F, Y$ drawn from $P_{FY}$. The Single-

Sample PPI++ estimate is given by

$$\hat{\theta}_{\text{Single-PPI++}} = \frac{1}{n} \sum_{i \in \mathcal{D}_n} y_i + \hat{\lambda}\left(\mu_f - \frac{1}{n} \sum_{i \in \mathcal{D}_n} f_i\right)$$

$$\hat{\lambda} = \widehat{\text{Cov}}(Y, F; \mathcal{D}_n)/\sigma_f^2$$

This estimator is simple, but the lack of sample splitting can complicate variance estimation and the construction of confidence intervals. For example, as we show in Proposition 4.6, when the same data are used to both (a) construct $\hat{\lambda}$, which minimizes the empirical variance, and (b) estimate the empirical variance while treating $\hat{\lambda}$ as fixed, the resulting variance estimates can be overly optimistic. Cross-fitting is a standard solution to this type of problem: splitting the labeled data, using one half to estimate $\lambda$, and the other to construct $\hat{\theta}$ using the chosen $\hat{\lambda}$. This procedure is repeated by swapping the roles of each half and the two estimates are averaged to produce a final estimate.

**Definition 3.5** (Cross-fit PPI++ Estimator, Infinite $N$). Let $\mathcal{D}_{n/2}^{(1)}, \mathcal{D}_{n/2}^{(2)}$ denote two independent $n/2$ samples of $F, Y$ drawn from $P_{FY}$, and $\hat{\theta}^{(k)}$ the PPI++ Estimator which uses $\mathcal{D}_{n/2}^{(k)}$ to estimate $\hat{\lambda}$ and $\mathcal{D}_{n/2}^{(k')}$ ($k \neq k'$) to estimate $\hat{\theta}$, s.t.,

$$\hat{\theta}^{(k)} := \frac{1}{n/2} \sum_{i \in \mathcal{D}_{n/2}^{(k')}} y_i + \hat{\lambda}^{(k)}\left(\mu_f - \frac{1}{n/2} \sum_{i \in \mathcal{D}_{n/2}^{(k')}} f_i\right)$$

$$\hat{\lambda}^{(k)} := \widehat{\text{Cov}}(F, Y; \mathcal{D}_{n/2}^{(k)})/\sigma_f^2,$$

and the Cross-fit PPI++ estimator is

$$\hat{\theta}_{\text{CF-PPI++}} = (\hat{\theta}^{(1)} + \hat{\theta}^{(2)})/2$$

We note that the cross-fit estimator uses only $n/2$ samples per-fold, and thus the estimate of the covariance $\widehat{\text{Cov}}(f, Y; \mathcal{D}_{n/2}^{(k)})$ is computed using $n/2$ labels.

*Remark* 3.1 (Mean Estimation vs M-Estimation). We note that our results focus primarily on mean estimation, whereas Angelopoulos et al. (2023b) also consider more general convex M-estimation. Our core insights still apply in that setting, since the asymptotic claims of PPI / PPI++ for M-estimation still proceed by a "mean estimation variance reduction" argument. For instance, the main result (Theorem 1) of Angelopoulos et al. (2023b) proceeds in two steps: First, it is argued that their approach leads to reduced variance in estimation of the expected gradient (instead of considering $Y, F$, they use the per-sample gradients $\nabla \ell(X, Y)$ and $\nabla \ell(X, F)$ calculated using $Y$ and $F$). Second, they link reduced variance in the estimate of the expected gradient to variance in the estimated parameter. The steps involved in the second step are similar to those conventionally used in standard proofs of asymptotic normality for M-estimators. As a result, our results would apply directly to a "no-free

lunch" in that first stage of the argument, with the caveat that the linkage between "variance in mean estimation of the gradient" and "variance in parameter estimation" is harder to characterize exactly in a non-asymptotic framework.

## 4. Theoretical Analysis

To understand the relative performance of the classical and PPI++ estimators, we focus on an exact characterization of $\mathbb{E}[(\hat{\theta} - \theta)^2]$, the finite-$n$ estimation error. Since both the Classical and Cross-fit PPI++ estimators are unbiased, analyzing the finite-$n$ estimation error is equivalent to the finite-$n$ variance $\text{Var}(\hat{\theta}) := \mathbb{E}[(\hat{\theta} - \mathbb{E}[\hat{\theta}])^2]$.

Our main result (Theorem 4.1) makes no substantive assumptions regarding the distribution of $Y$ and $F$. That said, we warm-up in Section 4.1 by analyzing the special case where labels and pseudo-labels are Gaussian, and the analysis / results are far simpler. In this setting, both the Single Sample PPI++ and Cross-fit PPI++ estimators are both unbiased, and their variances are comparable. The behavior in this setting is qualitatively similar to the more general results we derive later on.

In Section 4.2, we move on to analyze the variance of the Cross-fit PPI++ estimator for general distributions, where it remains unbiased. We decompose the variance into four parts: The variance of the classical estimator, a term that represents the *gain* in efficiency (i.e., a decrease in variance) attributable to pseudo-label quality, a term that represents the *loss* in efficiency due to errors in estimating the quality of the pseudo-labels and an additional term that captures the dependence between the two folds of the cross-fit estimator. The gain term depends directly on the covariance between the labels and the pseudo-labels while the loss term depends on the error in estimating this covariance from the labeled samples. The core of our no-free lunch result is that if *this efficiency loss in estimating the covariance from labeled data exceeds the gain offered by the covariance itself, then Cross-fit PPI++ will perform worse than Classical*.

A complete specification of the finite-sample variance of the Cross-fit PPI++ estimator requires an analysis of the efficiency loss term. We motivate this analysis with the illustrative case when the pseudo-labels are completely random. Here, the variance of Cross-fit PPI++ is always greater than the variance of the classical estimator, by a multiplicative factor of $O(1/n)$. Then, in Section 4.2.2, we characterize the variance of the Cross-fit PPI++ estimator for any joint distribution of gold-standard and pseudo-labels. Finally, in Section 4.3, we return to the discussion of the Single Sample PPI++ estimator, which is more difficult to analyze theoretically but has some unappealing properties from the perspective of confidence interval construction: It is biased

in finite samples, and estimates of the variance that treat $\lambda$ as fixed can lead to misleading optimism. We provide complete proofs for all theoretical results in Section A.

### 4.1. Warm Up: Gaussian Labels

We begin with the special case where $Y$ and $F$ are normally distributed, for two reasons. First, this special case allows for a unified analysis of both the Single Sample PPI++ and the Cross-fit PPI++ estimators, and establishes the existence of settings where both estimators under-perform Classical. Second, this special case provides simple expressions for the variance of each estimator that capture the fundamental "no free lunch" nature of the problem, yielding simple conditions where PPI++ outperforms the classical estimate. In the Gaussian setting, both estimators are **unbiased**, and so we consider their variance as a measure of performance.[3]

**Proposition 4.1** (Var of PPI++, Gaussian Case). *Let $Y, F$ be jointly Gaussian random variables, and consider the Single Sample PPI++ , Cross-fit PPI++ and Classical estimators which all make use of $n$ labeled samples. Then,*

$$\text{Var}(\hat{\theta}_{PPI++}) \tag{1}$$
$$= \text{Var}(\hat{\theta}_{Classical})\left(1 + \frac{1}{c \cdot n - 1}\right) - \frac{c \cdot n - 2}{n(c \cdot n - 1)} \cdot \frac{\sigma_{fy}^2}{\sigma_f^2}$$

*where $c = 1$ for Single Sample PPI++ and $c = 1/2$ for Cross-fit PPI++. Note that $\text{Var}(\hat{\theta}_{Classical}) = \sigma_y^2/n$*

Proposition 4.1 shows the exact variance of the PPI++ variants in the Gaussian setting. We see that the variance is reduced by a term that depends on the correlation between $F$ and $Y$. However, the variance is also inflated by a multiplicative factor in the first term, and hence we require the correlation to be greater than a certain level for the overall variance to be lower than that of the classical estimator. Equation (1) makes it easy to arrive at this required level, which we present in the below corollary:

**Corollary 4.1** (Condition for Improvement, Gaussian Case). *Let $Y, F$ be jointly Gaussian random variables, and consider the Single Sample PPI++ and Cross-fit PPI++ estimators which both make use of $n$ labeled samples. Then,*

$$\text{Var}(\hat{\theta}_{PPI++}) < \text{Var}(\hat{\theta}_{Classical}) \iff \frac{1}{\sqrt{c \cdot n - 2}} < \left|\frac{\sigma_{fy}}{\sigma_y \sigma_f}\right|$$

*where $c = 1$ for Single Sample PPI++ and $c = 1/2$ for Cross-fit PPI++.*

---

[3]The setting of Gaussian labels is unique: The empirical covariance of normally distributed random variables is independent of their sample means. This property reduces the dependence between the estimate of $\lambda$ (which depends on the estimated covariance) and the estimated value itself. As a result, Single Sample PPI++ is also unbiased in this setting. However, in the non-Gaussian case, Single Sample PPI++ is biased, while the use of sample-splitting retains the unbiasedness of Cross-fit PPI++, as we will see in Section 4.3

Corollary 4.1 provides a unifying perspective on the "no free lunch" phenomenon: If the correlation $\rho_{yf} := \sigma_{fy}/(\sigma_y \sigma_f)$ is non-zero, then pseudo-labels **can** improve our variance, but there is a minimum sample size at which this improvement kicks in.[4] However, if the correlation is zero, then both PPI++ estimators always perform worse than the classical estimator. This conclusion follows from a direct application of Proposition 4.1, where plugging a covariance of zero ($\sigma_{fy} = 0$) into Equation (1) yields a variance of PPI++ that is $(1 + 1/(c \cdot n - 1))$ times the variance of the classical estimator. Intuitively, this additional variance arises from estimation of $\lambda$. If the covariance is zero, then $\lambda^* = 0$, and this choice of $\lambda$ would yield the same variance as the classical estimator. In practice, however, $\hat{\lambda}$ is not exactly zero, leading to higher variance. As we will see in Proposition 4.4, the same gap arises for Cross-fit PPI++ under general distributions, and we discuss the intuition further in Section 4.2.1.

For the remainder of the paper, we consider the setting where $N$ is taken to be arbitrarily large. However, we note that in the Gaussian setting, it is possible to give exact expressions even when $N$ is finite.

**Proposition 4.2** (Condition for Improvement, Gaussian Case, Finite $N$). *Under the same conditions as Corollary 4.1, consider the Single Sample PPI++ where $N$ is finite. Then the condition for improvement becomes*

$$\text{Var}(\hat{\theta}_{Single\text{-}PPI++}) < \text{Var}(\hat{\theta}_{Classical}) \iff$$
$$\frac{1}{\sqrt{n - 2 - 8\frac{n-1}{N-1}}} < \left|\frac{\sigma_{fy}}{\sigma_y \sigma_f}\right|$$

### 4.2. Finite-Sample Variance of Cross-fit PPI++

Before we present our main results, we briefly note the *unbiasedness* of Cross-fit PPI++.

**Proposition 4.3** (Bias of Crossfit-PPI++). *The Cross-fit PPI++ estimator is unbiased, i.e., $\mathbb{E}[\hat{\theta}_{CF\text{-}PPI++}] = \theta$.*

This is a trivial result included for completeness, and for contrast with Proposition 4.5, where we will characterize the non-zero bias of Single Sample PPI++. The unbiasedness of Cross-fit PPI++ follows by noting that the estimate on each fold after sample-splitting is unbiased since $\hat{\lambda}$ is fixed within a fold, and so the simple average is unbiased by linearity of expectations. We now proceed to focus on the finite sample variance of Cross-fit PPI++ for general distributions. We first present our main result.

**Theorem 4.1** (Variance of Crossfit-PPI++). *Given a sample size of $n$ labels, the variance of the Cross-fit PPI++*

---

[4]Notably, the condition for Cross-fit PPI++ uses $n/2$, the size of a single fold, while the condition for Single Sample PPI++ uses $n$, which illustrates a cost of sample splitting in the Gaussian setting.

*estimator (Definition 3.5) is given by*

$$\text{Var}(\hat{\theta}_{\text{CF-PPI++}}) = \underbrace{\frac{\sigma_y^2}{n}}_{\text{Var}\hat{\theta}_c} - \underbrace{\frac{\sigma_{fy}^2}{n\sigma_f^2}}_{\text{(Eff. Gain)}} + \underbrace{\frac{1}{n\sigma_f^2}\mathbb{E}[(\hat{\sigma}_{fy} - \sigma_{fy})^2]}_{\text{(Eff. Loss)}}$$

$$+ \underbrace{\frac{2}{n^2\sigma_f^4}(\sigma_{yf^2} - 2\sigma_{fy}\mu_f)^2}_{\text{Cross-term b/w folds}} \quad (2)$$

The first term is the variance of the classical estimator. The second term (Eff. Gain) captures the gain in efficiency from using a predictor $f$ that is correlated with the true label $Y$, and the third term (Eff. Loss) captures the loss of efficiency that arises from the expected error in estimating the covariance from finite samples. In Lemma 4.1 we will characterize the loss term in Equation (2), which can be combined with Theorem 4.1 to give the exact variance of the Cross-fit PPI++ estimator. For now, Equation (2) yields intuition for when Cross-fit PPI++ fails to provide benefit.

**Corollary 4.2.**

$$\mathbb{E}[(\hat{\sigma}_{fy} - \sigma_{fy})^2] > \sigma_{fy}^2 \implies \text{Var}(\hat{\theta}_{CF\text{-}PPI++}) > \sigma_y^2/n$$

In other words, there is no free lunch, but rather an intuitive trade-off: If the error in estimating the covariance $\sigma_{fy}$ is higher than the magnitude of $\sigma_{fy}$, then Cross-fit PPI++ performs worse than using the $n$ labeled examples alone.

In the following, we characterize the covariance estimation error $\mathbb{E}[(\hat{\sigma}_{fy} - \sigma_{fy})^2]$. To build intuition, we first revisit the setting where $Y, F$ are independent, and prove that (in general), PPI++ performs worse than classical in this setting. In Section 4.2.2 we then give an explicit characterization (in Lemma 4.1) of the covariance estimation error term.

### 4.2.1. WARM-UP: PERFORMANCE GAP WITH RANDOM PSEUDO-LABELS

We revisit the case where $Y$ and $F$ are independent, but now allow for non-Gaussian distributions.

**Proposition 4.4.** *Given $n$ samples $\{y_i, f_i\}_{i=1}^n$ drawn from $P_{FY}$, where $Y$ and $F$ are independent with bounded second moments, the variance of Cross-fit PPI++ is given by*

$$\text{Var}(\hat{\theta}_{CF\text{-}PPI++}) = \frac{\sigma_y^2}{n}\left(1 + \frac{2}{n-2}\right)$$

Note that this non-Gaussian result is identical to the analogous result in the Gaussian case (obtained by plugging in $\sigma_{yf} = 0$ to Equation (1), where $c = 1/2$ for Cross-fit PPI++). Proposition 4.4 can help build intuition that bridges the non-asymptotic and asymptotic regimes: when pseudo-labels are independent of the true labels, the *finite-sample variance* of the Cross-fit PPI++ estimator is always

greater than the variance of the classical estimator, even if the *asymptotic variance* is identical.

### 4.2.2. FINITE-SAMPLE COVARIANCE ESTIMATION ERROR WITH GENERAL PSEUDO-LABELS

Before introducing our main result, we introduce additional notation. For two variables (e.g., $F$ and $Y$) and two integers $a, b$, we use $\sigma_{f^a y^b}$ to refer to the covariance between the random variable $F^a$ and the random variable $Y^b$. For a single variable $Y$, we use $\sigma_{Y^a}^2$ to denote the variance of $Y^a$.

First, we characterize the error in covariance estimation in Lemma 4.1, which, when plugged into Theorem 4.1, **yields an analytical expression for the variance of Cross-fit PPI++** in terms of terms involving the mean and covariance of $Y, F$ and their higher-order moments.

**Lemma 4.1** (Covariance Estimation Error, General Case)**.** *Let $Y$ and $F$ have bounded fourth-moments. Given $n$ IID samples $\{y_i, f_i\}_{i=1}^n$ drawn from $P_{YF}$, we have*

$$\mathbb{E}[(\hat{\sigma}_{yf} - \sigma_{yf})^2] = \frac{1}{n}\sigma_{f^2 y^2} + \frac{1}{n-1}\sigma_f^2\sigma_y^2 - \frac{(n-2)}{n(n-1)}\sigma_{fy}^2$$

$$- \frac{2}{n}[\sigma_{y^2 f}\mu_f + \sigma_{f^2 y}\mu_y - 2\sigma_{fy}\mu_f\mu_y]$$

Note that while here the covariance estimate is formed from $n$ IID samples, Cross-fit PPI++ forms this estimate from only $n/2$ samples assuming $n$ total labels are available.

### 4.2.3. ANALYTICAL EXPRESSION FOR THE FINITE-SAMPLE VARIANCE

Combining Lemma 4.1 with Theorem 4.1 provides an analytical expression for the exact variance of Cross-fit PPI++. While we do not write the entire expression here due to space constraints, we make the following observation: Of the four components of the variance in Equation (2), the first two are $O(1/n)$, and the last two are $O(1/n^2)$, since the covariance estimation error in Lemma 4.1 is $O(1/n)$

$$\text{Var}(\hat{\theta}_{\text{CF-PPI++}}) = \frac{1}{n}\left(\sigma_y^2 - \frac{\sigma_{fy}^2}{\sigma_f^2}\right)$$

$$+ \frac{1}{n}\left(\frac{1}{\sigma_f^2}\overbrace{\mathbb{E}[(\hat{\sigma}_{fy} - \sigma_{fy})^2]}^{O(1/n) \text{ by Lemma 4.1}}\right)$$

$$+ \frac{1}{n^2}\left(\frac{2}{\sigma_f^4}(\sigma_{yf^2} - 2\sigma_{fy}\mu_f)^2\right)$$

$$= \frac{1}{n}\left(\sigma_y^2 - \frac{\sigma_{fy}^2}{\sigma_f^2}\right) + O\left(\frac{1}{n^2}\right)$$

The full expression for the variance, expanding the covariance estimation error term, is a simple corollary of Theorem 4.1 and Lemma 4.1, and is provided in Corollary A.1

in the Appendix. The decomposition above provides further intuition regarding the mismatch between asymptotic results, as established in e.g., Angelopoulos et al. (2023b), which only consider the $O(1/n)$ term since the $O(1/n^2)$ terms vanishes asymptotically, and our finite-sample results, which take the $O(1/n^2)$ terms into account.

### 4.3. Bias and Over-Optimistic Variance Estimation of Single-Sample PPI++

We now turn to some less desirable properties of the Single Sample PPI++ estimator. Practitioners conducting statistical inference are often concerned with the construction of confidence intervals. Typically, we construct asymptotically valid confidence intervals using the variance of a consistent estimator, and hence lower variance yields tighter intervals. While such intervals are generally not guaranteed to maintain nominal coverage in finite samples, Single Sample PPI++ introduces two additional challenges. First, while asymptotically consistent, it is biased in finite sample, and second, it can produce overly optimistic variance estimates. We first characterize the exact bias[5] in Proposition 4.5, and in Proposition 4.6 we illustrate that under mild conditions, standard methods for estimating the variance of Single Sample PPI++ will necessarily yield estimates that are lower than the classical variance, even in scenarios (as illustrated in Section 4.1) where we know the MSE of Single Sample PPI++ is higher than that of the classical estimator.

**Proposition 4.5** (Bias of Single Sample). *The bias of the Single Sample PPI++ defined in Definition 3.4 is given by* $\mathbb{E}[\hat{\theta}_{Single\text{-}PPI++} - \theta] = \left(2\sigma_{yf}\mu_f - \sigma_{yf^2}\right)/(n\sigma_f^2)$

While in Section 4.2, we exactly characterized the variance of Cross-fit PPI++, we omit a similar analysis of Single Sample PPI++, since the latter case is far less tractable to analyze, owing to complex dependencies that arise from estimating $\lambda$ on the same data used to form the final estimate. We focus instead on the typical approach to *estimating* the variance that practitioners use to derive confidence intervals, and give the following result on its overly optimistic nature.

In Proposition 4.6 we consider a variance estimation strategy that treats $\lambda$ as fixed, and uses plug-in estimators to estimate each component of the variance (namely, $\hat{\sigma}_y$, $\hat{\sigma}_f$, and $\hat{\sigma}_{yf}$). In the appendix, we present a similar result for another common method of variance estimation, that similarly treats the estimated $\lambda$ as fixed, and uses the empirical variance of $y - \lambda f$ in $\mathcal{D}_n$ and $\lambda f$ in $\mathcal{D}'_N$.

**Proposition 4.6** (Optimistic Variance Estimation of Single Sample PPI++). *The plug-in variance estimator for Single*

---

[5]It is generally known that Single Sample PPI++ is biased in finite samples (Chaganty et al., 2018; Eyre & Madras, 2025), but we do not know of other results giving the exact form, so we produce it here. While Chaganty et al. (2018) show that the bias of a similar estimator is $O(1/n)$, they do not give the exact form.

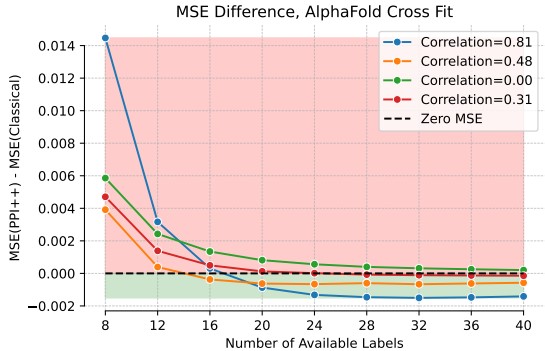

*(a)* Cross-fit PPI++ Estimator

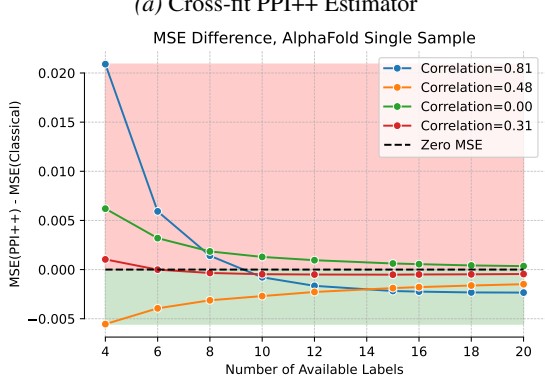

*(b)* Single-Sample PPI++ Estimator

*Figure 2.* MSE diff: $\mathsf{MSE}(\hat{\theta}_{\text{PPI++}}) - \mathsf{MSE}(\hat{\theta}_{\text{Classical}})$ vs. Sample Size on the Alphafold Dataset for PPI++ estimators with pseudo-labels of varying quality. Each line represents PPI++ with pseudo-labels of different correlations. The blue line uses the original set, which has strong predictive performance, but only improves estimation error at $n \geq 20$ (a) and $n \geq 10$ (b).

*Sample PPI++ given by* $\hat{\sigma}^2_{Single\text{-}PPI++} = (1/n)[\hat{\sigma}^2_y + \lambda^2 \sigma^2_f - 2\lambda\hat{\sigma}_{fy}]$ *reduces to* $(1/n)[\hat{\sigma}^2_y - \hat{\sigma}^2_{fy}/\sigma^2_f]$, *which is always less than the plug-in estimate of the classical variance* $\hat{\sigma}^2_y/n$.

As a result of Proposition 4.6, the asymptotic confidence intervals constructed for Single Sample PPI++ will always be narrower than those of Classical, even if the true variance is larger. In Section 5, we show empirically that these *always tighter intervals* can result in significant drops in coverage.

## 5. Experiments

We now illustrate our theory with experiments on the Alphafold (Jumper et al., 2021) dataset used in Angelopoulos et al. (2023b), using a similar experimental setup where we use bootstrapping to simulate multiple draws from a common distribution, and use the empirical mean of the (larger) original dataset set for establishing ground truth. Note that we focus on the low sample size regime where n < 50. See Section B for more details. See Section C for additional results on the galaxies (Angelopoulos et al., 2023b) dataset.

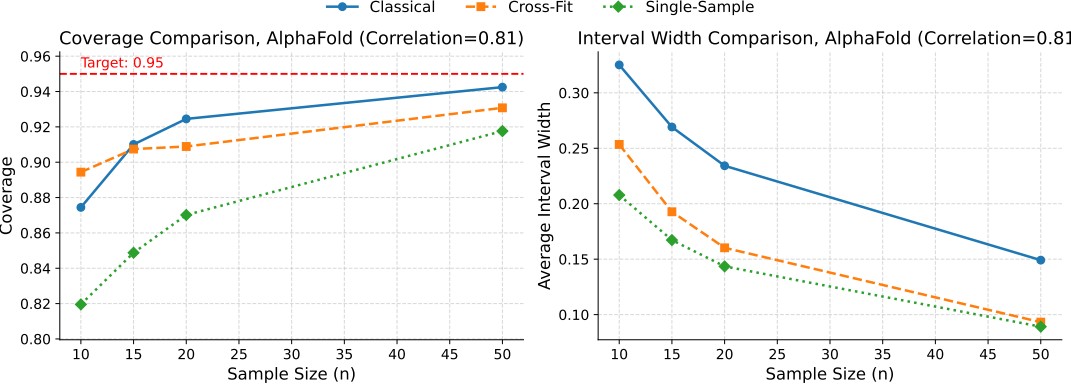

*Figure 3.* **AlphaFold.** Coverage (left) and interval width (right) for Cross-fit PPI++ vs. Single Sample PPI++ using pseudo-labels from Alphafold (corr = 0.81). Single Sample PPI++ under-covers relative to Cross-fit PPI++ and Classical.

In Figure 2, we show the difference between MSE of PPI++ and classical as a function of sample size $n$. We use MSE to reflect the fact that while Cross-fit PPI++ is unbiased, Single Sample PPI++ is not. For MSE difference, the region below 0 marks improvement. We consider pseudo-labels with various correlation levels (0.81 is the original set in Alphafold). The results are in line with our theoretical developments for Cross-fit PPI++, and illustrate that similar trends hold for Single Sample PPI++. Using pseudo-labels with varying quality, the MSE either (a) starts off worse than classical, but starts to improve over classical past a certain 'tipping point' sample size, (b) starts off worse than classical, but closes the gap as dataset size increases, or (c) starts off better than classical and stays there. For all non-zero correlation sets, there exists some $n$ after which PPI++ has improved MSE over classical. We note that for the pseudo-label models shown here, Single Sample PPI++ tends to achieve lower MSE than Cross-fit PPI++.

In Figure 3, we show the coverage (left) and average widths of intervals (right) constructed via Single Sample PPI++ and Cross-fit PPI++ over $n$. Here we present the results with the original pseudo-label set in Alphafold with correlation 0.81, while a further-noised set is presented in Section D. We note that the Single Sample PPI++ method produces intervals that are narrower than those from the Classical. However, these substantively tighter intervals come at the cost of inadequate coverage. For instance, for the original pseudo-labels (top), at $n = 20$, the realized coverage with Single Sample PPI++ is 87%. In Section D, Figure 6, we see that both Single Sample PPI++ and Cross-fit PPI++ show wider intervals as the correlation drops, and while Cross-fit PPI++ widens enough to match the coverage of Classical, Single Sample PPI++ still falls short in this regard.

## 6. Discussion

Having presented our main theoretical results, we now step back to summarize what these results mean for applied use of prediction-powered methods.

**A Cautionary Note (No Free Lunch):** The practical appeal of PPI is clear: when pseudo-labels *are informative*, they can increase the effective sample size. Our work seeks to establish the same spirit for PPI++, guiding practitioners away from the pitfall of believing that pseudo-labels *always* help, even when they are close to uninformative.

**An Optimistic Note**: Our results show that the required dependence between pseudo-labels and true labels is not fixed, but depends on the number of labeled samples, and is not necessarily an onerous requirement: Taking the Gaussian case as a general rule-of-thumb, the correlation lower-bound of $(n/2 - 2)^{-1/2}$ for Cross-fit PPI++ may be reasonable to achieve for moderate $n$. The key is that *some* assumption on pseudo-label performance is required for improvement, but such assumptions may be quite reasonable.

**A Practical Note**: Our work provides more than just a "no free lunch" result, or a "rule-of-thumb" for gauging the benefit of pseudo-labels. Theorem 4.1, combined with Lemma 4.1, can be used to derive the exact relative variance of Cross-fit PPI++. While the full form depends on some higher-order covariances (e.g., between $Y$ and $F^2$), the full expression can be computed analytically in many common settings, and used to construct visualizations like the one shown in Figure 1 for the binary setting. Thus, if the practitioner is willing to posit a reasonable assumption on the quality of the pseudo-labels, and the number of labels they have, our framework can determine if there is an upside to use PPI++ or PPI and the magnitude of improvement.

Overall, our hope is that this work helps practitioners use techniques like PPI++ with confidence, making transparent the implicit assumptions required to see practical benefit.

## Impact Statement

Given the availability and ease of use of LLMs and VLMs, it has never been easier to construct *some* predictor that appears to plausibly have "good enough" performance, even without using labeled data. Thus, pseudo-labels are already a routine and central part of the decision-making processes within many high-impact applications. For instance, LLM-as-judge evaluations in industry production pipelines (Zhou et al., 2025; Rahmani et al., 2025; Li et al.; Zheng et al., 2023), LLMs as proxies for human responses in social science experimentation (Hewitt et al., 2024; Manning et al., 2024; Anthis et al., 2025), the use of foundation models in randomized controlled trials (Goldenholz et al., 2025; Lai et al., 2024), clinical trials (Poulet et al., 2025), PPI-based content moderation (Waldetoft et al., 2025; Ludwig et al., 2024), LLMs-as-peer-reviewers (Tyser et al., 2024; Goldberg et al., 2024), etc. While procedures like PPI++ are important tools, the promise of these methods to *always* turn these synthetic signals, however noisy they may be, into statistically valid inferences might tempt practitioners to throw in "the kitchen sink" of black-box pseudo labels, confident that it cannot hurt. Given the already widespread use of LLM-labels, such misplaced optimism could prove very costly. Our theoretical findings narrate a "no-free lunch" style cautionary tale: proof that there are practical settings where PPI++ can hurt and PPI++ confidence intervals fail to meet advertised rates of coverage. On the other hand, our theory provides a quantitative framework that converts the assumptions that a practitioner is willing to make about the quality of their pseudo-labels into a quantification of the magnitude of the upside, if any.

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

# A. Proofs

In this appendix section, we include formal proofs for all the mathematical statements in the main paper. We structure the section as follows. After giving the full expression for the variance of Cross-Fit PPI++, we begin with an introduction of some useful lemmas on properties of moments, variances and covariances. Some lemmas are interesting in their own right while other lemmas are introduced solely to establish some results that recur in the workings of other proofs. We then consider the bias of Cross-fit PPI++ and Single Sample PPI++. We show that Cross-fit PPI++ is unbiased while Single Sample PPI++ is biased and we derive this bias that we presented in Proposition 4.5. We then present a full proof of the finite sample variance of the Cross-fit PPI++ and all the statements in Section 4.2 including conditions where Cross-fit PPI++ has higher variance than Definition 3.1. Following this, we construct the proof for the covariance estimation error presented in Lemma 4.1. With these proofs in place, we are able to apply them to the Gaussian setting to complete the proofs in Section 4.1. Finally, we provide proofs for the optimistic variance estimate of Single Sample PPI++ that we presented in Proposition 4.6.

## A.1. Full Expression for the Variance of Cross-Fit PPI++

**Corollary A.1.** *Under the conditions of Theorem 4.1 and Lemma 4.1, we have*

$$
\begin{aligned}
\mathrm{Var}(\hat{\theta}_{CF\text{-}PPI++}) =& \frac{1}{n}\left(\sigma_y^2 - \frac{\sigma_{fy}^2}{\sigma_f^2}\right) \\
&+ \frac{1}{n\sigma_f^2}\left(\frac{1}{n/2}\sigma_{f^2y^2} + \frac{1}{n/2-1}\sigma_f^2\sigma_y^2 - \frac{(n/2-2)}{n/2(n/2-1)}\sigma_{fy}^2 - \frac{2}{n/2}[\sigma_{y^2f}\mu_f + \sigma_{f^2y}\mu_y - 2\sigma_{fy}\mu_f\mu_y]\right) \\
&+ \frac{1}{n^2}\left(\frac{2}{\sigma_f^4}(\sigma_{yf^2} - 2\sigma_{fy}\mu_f)^2\right) \\
=& \frac{1}{n}\left(\sigma_y^2 - \frac{\sigma_{fy}^2}{\sigma_f^2}\right) + O\left(\frac{1}{n^2}\right)
\end{aligned}
$$

*Proof.* The statement follows by plugging in Lemma 4.1 (using $n/2$ samples) into Theorem 4.1. $\square$

## A.2. Foreword and Useful Lemmas

*Remark* A.1 (Use of subscripts). We often make use of subscripts $y_i, f_j$, etc in two contexts: First, we use these subscripts in the standard context as part of a sum. For instance, we may write $\mathbb{E}[\bar{y}\bar{f}] = \frac{1}{n}\sum_{i=1}^n \mathbb{E}[y_i\bar{f}]$. Second, we use these subscripts as short-hand to distinguish between variables corresponding to different samples. Continuing the previous example, we may write $\mathbb{E}[\bar{y}\bar{f}] = \frac{1}{n}\sum_{i=1}^n \mathbb{E}[y_i\bar{f}] = \frac{1}{n}n \cdot \mathbb{E}[y_i\bar{f}]$. Here, we make implicit use of the fact that for any $i \in \{1,\ldots,n\}$, the expectation $\mathbb{E}[y_i\bar{f}]$ is identical by exchangeability. This notation is particularly useful when indicies differ. For instance $\mathbb{E}[y_i\bar{f}] = \frac{1}{n}\mathbb{E}[y_i(f_i + \sum_{j\neq i}f_j)] = \frac{1}{n}\mathbb{E}[y_if_i] + \frac{(n-1)}{n}\mathbb{E}[y_if_j]$, where $\mathbb{E}[y_if_j]$ denotes the expectation of $y_i$ (from some sample $i$) and $f_j$ (drawn from a distinct sample $j \neq i$), such that $\mathbb{E}[y_if_j] = \mathbb{E}[y]\mathbb{E}[f]$ by independence, while $\mathbb{E}[y_if_i] = \mathbb{E}[yf]$.

**Lemma A.1** (Covariance Representation). *The empirical covariance estimated from $n$-samples can be equivalently written*

$$
\frac{1}{n-1}\sum_{i=1}^n (y_i - \bar{y})(f_i - \bar{f}) = \frac{1}{n-1}\sum_{i=1}^n (y_if_i - \bar{y}\bar{f}) \tag{3}
$$

*Proof.*

$$
\frac{1}{n-1}\sum_{i=1}^n (y_i - \bar{y})(f_i - \bar{f}) = \frac{1}{n-1}\sum_{i=1}^n (y_if_i - y_i\bar{f} - f_i\bar{y} + \bar{y}\bar{f}) \tag{4}
$$

and $\sum_i y_i\bar{f} = n\bar{y}\bar{f} = \sum_i \bar{y}\bar{f}$, and similarly with $f_i\bar{y}$. $\square$

**Lemma A.2** (Cross-term with n-sample means)**.** *Let $Y$ and $F$ be random variables with bounded second-moments. Suppose that we observe $n$ IID samples $\{y_i, f_i\}_{i=1}^n$ from the joint distribution of $Y$ and $F$. Let $\bar{y}_n, \bar{f}_n$ denote the sample means of $Y, F$ on the observed n-sample. Furthermore, let $\hat{\lambda}$ be independent of $\{y_i, f_i\}_{i=1}^n$ and computed on an independent n-sample. Then we have,*

$$\mathbb{E}\left[\hat{\lambda}(\bar{y}_n - \theta)(\mu_f - \bar{f}_n)\right] = -\frac{\sigma_{yf}^2}{n\sigma_f^2}$$

*Proof.*

$$\mathbb{E}\left[\hat{\lambda}(\bar{y}_n - \theta)(\mu_f - \bar{f}_n)\right]$$
$$= \mathbb{E}[\hat{\lambda}]\mathbb{E}[(\bar{y}_n - \theta)(\mu_f - \bar{f}_n)] \qquad\qquad \text{Independence of } \hat{\lambda}$$
$$= \mathbb{E}[\hat{\lambda}]\mathbb{E}\left[\bar{y}_n(\mu_f - \bar{f}_n) - \theta(\mu_f - \bar{f}_n)\right]$$
$$= \mathbb{E}[\hat{\lambda}]\mathbb{E}\left[\bar{y}_n(\mu_f - \bar{f}_n)\right] - \mathbb{E}[\hat{\lambda}]\mathbb{E}\left[\theta(\mu_f - \bar{f}_n)\right] \qquad\qquad \text{Linearity of Expectation}$$
$$= \mathbb{E}[\hat{\lambda}]\mathbb{E}\left[\bar{y}_n(\mu_f - \bar{f}_n)\right] - \mathbb{E}[\hat{\lambda}]\theta\mathbb{E}\left[(\mu_f - \bar{f}_n)\right] \qquad\qquad \theta \text{ is constant}$$
$$= \mathbb{E}[\hat{\lambda}]\mathbb{E}\left[\bar{y}_n(\mu_f - \bar{f}_n)\right] - \mathbb{E}[\hat{\lambda}]\theta(\mathbb{E}[\mu_f] - \mathbb{E}[\bar{f}_n]) \qquad\qquad \text{Linearity of Expectation}$$

Since $\mu_f = \mathbb{E}[\bar{f}_n]$, the second term vanishes. Further $\hat{\lambda} = \widehat{\mathrm{Cov}}(Y, F)/\sigma_f^2$. where $\widehat{\mathrm{Cov}}(Y, F)$ is an unbiased estimate of the covariance between $Y, F$. Therefore, $\mathbb{E}[\hat{\lambda}] = \frac{\sigma_{fy}}{\sigma_f^2}$. Plugging these in,

$$\mathbb{E}\left[\hat{\lambda}(\bar{y}_n - \theta)(\mu_f - \bar{f}_n)\right] = \frac{\sigma_{fy}}{\sigma_f^2}\mathbb{E}\left[\bar{y}_n(\mu_f - \bar{f}_n)\right]$$
$$= \frac{\sigma_{fy}}{\sigma_f^2}\left[\mathbb{E}[\bar{y}_n\mu_f] - \mathbb{E}[\bar{y}_n\bar{f}_n]\right] \qquad\qquad \text{Linearity of Expectation}$$

Note that $\mathbb{E}[\bar{y}_n\mu_f] = \mu_y\mu_f$, and $\mathbb{E}[\bar{y}_n\bar{f}_n]$ can be written as

$$\mathbb{E}\left[\bar{y}_n\bar{f}_n\right] = \frac{1}{n^2}\mathbb{E}\left[\sum_i\sum_j y_i f_j\right]$$
$$= \frac{1}{n^2}\mathbb{E}\left[\sum_{i=j} y_i f_j + \sum_{i\neq j} y_i f_j\right]$$
$$= \frac{1}{n^2}\left(n\mathbb{E}[y_i f_j] + n(n-1)\mathbb{E}[y_i f_j]\right)$$
$$= \frac{\sigma_{yf}}{n} + \frac{n}{n^2}\mu_f\mu_y + \frac{n(n-1)}{n^2}\mu_f\mu_y$$
$$= \frac{\sigma_{fy}}{n} + \mu_f\mu_y$$

Where we have used the fact that in the expansion of $\bar{y}\bar{f}$, there are $n$ terms where $i = j$, such that $E[y_i f_i] = \sigma_{fy} + \mu_f\mu_y$ by definition of covariance, and there are $n(n-1)$ terms where $i \neq j$, such that $E[y_i f_j] = \mu_f\mu_y$, since $y_i \perp f_j$. Putting these together, we have

$$\mathbb{E}[\bar{y}_n\mu_f] - \mathbb{E}[\bar{y}_n\bar{f}_n] = \mu_y\mu_f - \frac{\sigma_{fy}}{n} - \mu_f\mu_y = -\frac{\sigma_{fy}}{n}$$

Therefore,

$$\mathbb{E}\left[\hat{\lambda}(\bar{y}_n - \theta)(\mu_f - \bar{f}_n)\right] = \frac{\sigma_{fy}}{\sigma_f^2}\left[\mathbb{E}[\bar{y}_n\mu_f] - \mathbb{E}[\bar{y}_n\bar{f}_n]\right] = \frac{\sigma_{fy}}{\sigma_f^2}[-\frac{\sigma_{fy}}{n}] = -\frac{\sigma_{fy}^2}{n\sigma_f^2}$$

which completes the proof.

**Note:** if the sample means where computed using $n' = c \cdot n$ samples, then we would simply get:

$$\mathbb{E}\left[\hat{\lambda}(\bar{y}_{n'} - \theta)(\mu_f - \bar{f}_{n'})\right] = -\frac{\sigma_{fy}^2}{n'\sigma_f^2} = -\frac{\sigma_{fy}^2}{cn\sigma_f^2} \tag{5}$$

$\square$

**Lemma A.3.** *Let $\bar{f}, \bar{y}$ denote sample means computed on $n$ IID samples.*

*We have it that*

$$\mathbb{E}[\hat{\sigma}_{yf}\bar{f}] = \frac{1}{n}\left(\sigma_{yf^2} + (n-2)\sigma_{yf}\mu_f\right)$$

$$\mathbb{E}[\hat{\sigma}_{yf}\bar{y}] = \frac{1}{n}\left(\sigma_{y^2f} + (n-2)\sigma_{yf}\mu_y\right)$$

*Proof.* We demonstrate for $\mathbb{E}[\hat{\sigma}_{yf}\bar{f}]$, and the case with $\bar{y}$ follows by symmetry.

$$\mathbb{E}[\widehat{\text{Cov}}(y,f)\bar{f}] = \mathbb{E}\left[\frac{1}{n-1}\sum_{i=1}^{n}(y_i - \bar{y})(f_i - \bar{f})\bar{f}\right] = \frac{1}{n-1}\sum_{i=1}^{n}\mathbb{E}[(y_if_i - \bar{f}\bar{y}))\bar{f}]$$

where have we used Lemma A.1 and linearity of expectation. We can then use the fact that

$$\bar{f}\bar{y} = \frac{1}{n^2}\sum_{j}\sum_{k}f_jy_k = \frac{1}{n^2}\left(\sum_{j=k}f_jy_k + \sum_{j\neq k}f_jy_k\right),$$

to obtain

$$\mathbb{E}[\widehat{\text{Cov}}(y,f)\bar{f}] = \frac{1}{n-1}\sum_{i=1}^{n}\mathbb{E}[(y_if_i - \bar{f}\bar{y}))\bar{f}]$$

$$= \frac{1}{n-1}\sum_{i=1}^{n}\mathbb{E}\left[\frac{1}{n^2}\left(n^2(y_if_i) - \sum_{j=k}f_jy_k - \sum_{j\neq k}f_jy_k\right)\bar{f}\right]$$

$$= \frac{1}{n-1}\sum_{i=1}^{n}\left(\frac{n(n-1)}{n^2}\mathbb{E}[y_if_i\bar{f}] - \frac{n(n-1)}{n^2}\mathbb{E}[f_jy_k\bar{f}]\right)$$

$$= \frac{n}{n-1}\left(\frac{n(n-1)}{n^2}\mathbb{E}[y_if_i\bar{f}] - \frac{n(n-1)}{n^2}\mathbb{E}[f_jy_k\bar{f}]\right)$$

$$= \left(\mathbb{E}[y_if_i\bar{f}] - \mathbb{E}[f_jy_k\bar{f}]\right) \tag{6}$$

We then consider each term in Equation (6) separately

$$\mathbb{E}[y_if_i\bar{f}] = \frac{1}{n}\left(\mathbb{E}[y_if_if_i] + \sum_{j\neq i}\mathbb{E}[y_if_if_j]\right) \qquad = \frac{1}{n}\mathbb{E}[y_if_if_i] + \frac{n-1}{n}\mathbb{E}[y_if_if_j]$$

$$\mathbb{E}[f_jy_k\bar{f}] = \frac{1}{n}\left(\mathbb{E}[f_jy_kf_j] + \mathbb{E}[f_jy_kf_k] + \sum_{i\neq j\neq k}\mathbb{E}[f_jy_kf_i]\right) \qquad = \frac{1}{n}\left(\mathbb{E}[f_jy_kf_j] + \mathbb{E}[f_jy_kf_k] + (n-2)\mathbb{E}[f_jy_kf_i]\right)$$

where we use the subscripts $i, j, k$ as described in Remark A.1. We then rewrite Equation (6) as

$$\mathbb{E}[y_if_i\bar{f}] - \mathbb{E}[f_jy_k\bar{f}] = \frac{1}{n}\left(\mathbb{E}[yf^2] + (n-1)\mathbb{E}[yf]\mathbb{E}[f] - \mathbb{E}[y]\mathbb{E}[f^2] - \mathbb{E}[f]\mathbb{E}[yf] - (n-2)\mathbb{E}[f]^2\mathbb{E}[y]\right)$$

$$= \frac{1}{n}\left(\sigma_{yf^2} + (n-1)\mathbb{E}[yf]\mathbb{E}[f] - \mathbb{E}[f]\mathbb{E}[yf] - (n-2)\mathbb{E}[f]^2\mathbb{E}[y]\right)$$

$$= \frac{1}{n}\left(\sigma_{yf^2} + (n-2)\mathbb{E}[yf]\mathbb{E}[f] - (n-2)\mathbb{E}[f]^2\mathbb{E}[y]\right)$$

$$= \frac{1}{n}\left(\sigma_{yf^2} + (n-2)\sigma_{yf}\mu_f\right)$$

which gives the desired result. $\qquad\square$

While the following result is of interest in it's own right (and makes up substantially the entire proof of Proposition 4.5), we give it here as a Lemma, as we also make use of it in the proof of Theorem 4.1.

**Lemma A.4.** *Given $\hat\lambda$ and $\bar f$ estimated on the same sample, we have it that*

$$\mathbb{E}[(\hat\lambda(\bar f - \mu_f))] = \frac{1}{n\sigma_f^2}\left(\sigma_{yf^2} - 2\sigma_{yf}\mu_f\right)$$

*Proof.*

$$
\begin{aligned}
\mathbb{E}[(\hat\lambda(\bar f - \mu_f))] &= \frac{1}{\sigma_f^2}\mathbb{E}[\widehat{\mathrm{Cov}}(y,f)(\bar f - \mu_f)] \\
&= \frac{1}{\sigma_f^2}\mathbb{E}[\widehat{\mathrm{Cov}}(y,f)\bar f] - \frac{\sigma_{yf}\mu_f}{\sigma_f^2} \\
&= \frac{1}{n\sigma_f^2}\left(\sigma_{yf^2} + (n-2)\sigma_{yf}\mu_f\right) - \frac{\sigma_{yf}\mu_f}{\sigma_f^2} \\
&= \frac{1}{n\sigma_f^2}\left(\sigma_{yf^2} - 2\sigma_{yf}\mu_f\right)
\end{aligned}
\tag{7}
$$

Where Equation (7) follows from Lemma A.3. $\qquad\square$

A well-known result in statistics is that for Gaussian data, the sample mean and the sample covariance matrix are independent. Here we give the formal statement and include the proof.

**Lemma A.5** (Independence of sample mean and covariance, Gaussian)**.** *Given I.I.D. random sample $X_1, ..., X_n$ from a multivariate Gaussian distribution $\mathcal{N}(\mu, \Sigma)$, then the sample mean $\bar X = \frac{\sum_{i=1}^n X_i}{n}$ and the sample covariance matrix $S = \frac{1}{n-1}\sum_{i=1}^n (X_i - \bar X)(X_i - \bar X)^T$ are independent.*

*Proof.* Let $D_i := X_i - \bar X$. Since $D_i$ is a linear combination of independent random normal vectors, $D_i$ also follows a multivariate normal distribution.

Now let's consider the covariance between $D_i$ and $\bar X$.

$$
\begin{aligned}
\mathrm{Cov}(D_i, \bar X) &= \mathrm{Cov}(X_i - \bar X, \bar X) \\
&= \mathrm{Cov}(X_i, \bar X) - \mathrm{Cov}(\bar X, \bar X) \\
&= \mathrm{Cov}(X_i, \frac{1}{n}\sum_{j=1}^n X_j) - \mathrm{Cov}(\frac{1}{n}\sum_{j=1}^n X_j, \frac{1}{n}\sum_{j=1}^n X_j) \\
&= \mathrm{Cov}(X_i, \frac{1}{n}X_i) - \frac{1}{n^2}\sum_{j=1}^n \mathrm{Cov}(X_j, X_j) \qquad X_j \perp X_i, \forall i \neq j \\
&= \frac{1}{n}\Sigma - \frac{1}{n}\Sigma \\
&= 0
\end{aligned}
$$

Since $D_i, \bar X$ are Gaussian, $D_i, \bar X$ are independent, $\forall i$. Therefore $S = \sum_{i=1}^n D_i D_i^T$ and $\bar X$ are independent. $\qquad\square$

### A.3. Cross-fit PPI++ is unbiased

While fairly simple to observe, we provide a proof for the unbiasedness of the Cross-fit PPI++ estimator. This proof is intended to provide an intuition for the next section where we consider the bias of Single Sample PPI++. We include this unbiasedness as a proposition

**Proposition A.1.** *Consider the Cross-fit PPI++ as defined in Definition 3.5. The expected value of this estimator is equal to the mean $E[Y]$. That is,*

$$\mathbb{E}[\hat{\theta}_{CF\text{-}PPI++}] = \theta \tag{8}$$

*where $\theta = \mathbb{E}[Y]$*

*Proof.*

$$\hat{\theta}_{\text{CF-PPI++}} = \frac{\hat{\theta}_1 + \hat{\theta}_2}{2}$$

where $\hat{\theta}_k$, as defined in Definition 3.5, are the estimates formed on each fold of the data using the other fold for computing $\hat{\lambda}$.

Therefore,

$$\mathbb{E}[\hat{\theta}_{\text{CF-PPI++}}] = \mathbb{E}\left[\frac{\hat{\theta}_1 + \hat{\theta}_2}{2}\right] = \frac{\mathbb{E}\left[\hat{\theta}_1 + \hat{\theta}_2\right]}{2} = \mathbb{E}[\hat{\theta}_k]$$

since $\mathbb{E}[\hat{\theta}_1] = \mathbb{E}[\hat{\theta}_2]$ which we denote $\mathbb{E}[\hat{\theta}_k]$.

Now,

$$\hat{\theta}_k = \bar{y}_n + \hat{\lambda}(\mu_f - \bar{f}_n)$$

Then,

$$\mathbb{E}[\hat{\theta}_k] = \mathbb{E}[\bar{y}_n] + \mathbb{E}[\hat{\lambda}] \times (\mu_f - \mathbb{E}[\bar{f}_n])$$

where we have used that $\hat{\lambda}$ is independent of $\bar{f}_n$, combined with the linearity of expectations.

Now, since $\bar{y}_n = \mathbb{E}[Y] = \theta$ and $\bar{f}_n = \mathbb{E}[F] = \mu_f$, we have

$$\mathbb{E}[\hat{\theta}_k] = \theta + \mathbb{E}[\hat{\lambda}] \times (\mu_f - \mu_f) = \theta$$

$\square$

The key to the unbiasedness of Cross-fit PPI++ is the independence of $\hat{\lambda}$ from the data used to compute sample means $\bar{f}_n$. Since this independence is not applicable to Single Sample PPI++, we cannot factor $\hat{\lambda}$ into its own expectation leading to a finite-sample bias as we shall prove next.

### A.4. Proof of Proposition 4.5: Bias of Single Sample PPI++

**Proposition 4.5** (Bias of Single Sample). *The bias of the Single Sample PPI++ defined in Definition 3.4 is given by $\mathbb{E}[\hat{\theta}_{Single\text{-}PPI++} - \theta] = \left(2\sigma_{yf}\mu_f - \sigma_{yf^2}\right)/(n\sigma_f^2)$*

*Proof.* The Single Sample PPI++ as defined in Definition 3.4 is

$$\hat{\theta}_{\text{Single-PPI++}} = \frac{1}{n} \sum_{i \in \mathcal{D}_n} y_i + \hat{\lambda} \left( \mu_f - \frac{1}{n} \sum_{i \in \mathcal{D}_n} f_i \right)$$

$$\hat{\lambda} = \frac{\widehat{\text{Cov}}(Y, F; \mathcal{D}_n)}{\sigma_f^2}$$

We have,

$$\mathbb{E}[\hat{\theta}_{\text{Single-PPI++}}] = \frac{1}{n} \mathbb{E} \left[ \sum_{i \in \mathcal{D}_n} y_i \right] + \mathbb{E} \left[ \hat{\lambda} \left( \mu_f - \frac{1}{n} \sum_{i \in \mathcal{D}_n} f_i \right) \right]$$

where we have used the linearity of expectations. Applying this linearity into the sum over $y_i$, we obtain

$$\mathbb{E}[\hat{\theta}_{\text{Single-PPI++}}] = \frac{1}{n} \sum_{i \in \mathcal{D}_n} \mathbb{E}[y_i] + \mathbb{E} \left[ \hat{\lambda} \left( \mu_f - \frac{1}{n} \sum_{i \in \mathcal{D}_n} f_i \right) \right]$$

Since, $E[Y_i] = E[Y] = \theta$, we get

$$\mathbb{E}[\hat{\theta}_{\text{Single-PPI++}}] = \theta + \mathbb{E} \left[ \hat{\lambda} \left( \mu_f - \frac{1}{n} \sum_{i \in \mathcal{D}_n} f_i \right) \right]$$

Thus the bias, becomes,

$$\mathbb{E}[\hat{\theta}_{\text{Single-PPI++}} - \theta] = \mathbb{E} \left[ \hat{\lambda} \left( \mu_f - \bar{f}_n \right) \right]$$

In the case of Cross-fit PPI++, at this stage, we were able to separate $\hat{\lambda}$ from the difference in expectation of mean and sample mean, reducing the bias to zero. Here, $\hat{\lambda}$ and $\bar{f}_n$ are estimated on the sample making them dependent. However, from Lemma A.4 we have,

$$\mathbb{E} \left[ \hat{\lambda} \left( \mu_f - \frac{1}{n} \sum_{i \in \mathcal{D}_n} f_i \right) \right] = \frac{1}{n\sigma_f^2} \left( 2\sigma_{yf}\mu_f - \sigma_{yf^2} \right)$$

which completes the proof.

$\square$

## A.5. Theorem 4.1: Finite-Sample Variance of Cross-fit PPI++ Estimator

**Theorem 4.1** (Variance of Crossfit-PPI++). *Given a sample size of $n$ labels, the variance of the Cross-fit PPI++ estimator (Definition 3.5) is given by*

$$\text{Var}(\hat{\theta}_{CF\text{-}PPI++}) = \underbrace{\frac{\sigma_y^2}{n}}_{\text{Var}\hat{\theta}_c} - \underbrace{\frac{\sigma_{fy}^2}{n\sigma_f^2}}_{\text{(Eff. Gain)}} + \underbrace{\frac{1}{n\sigma_f^2}\mathbb{E}[(\hat{\sigma}_{fy} - \sigma_{fy})^2]}_{\text{(Eff. Loss)}}$$

$$+ \underbrace{\frac{2}{n^2\sigma_f^4}(\sigma_{yf^2} - 2\sigma_{fy}\mu_f)^2}_{\text{Cross-term b/w folds}} \tag{2}$$

*Proof.* The structure of the proof is as follows. We first derive the variance in terms of the variance of the PPI++ estimate formed from a single-fold. Then, we derive the variance of a single-fold. Putting the two together completes the proof of Theorem 4.1.

A.5.1. VARIANCE IN TERMS OF VARIANCE OF SINGLE-FOLD ESTIMATE $\hat{\theta}_k$

Let $\mathcal{D}_1, \mathcal{D}_2$ denote our two dataset splits. Then we have that $\hat{\theta}_1$ is derived using $\mathcal{D}_1$ for $\hat{\lambda}$ and $\mathcal{D}_2$ for the point estimate, and vice versa for $\hat{\theta}_2$. The variance of the estimate is given by,

$$\frac{\hat{\theta}_1 + \hat{\theta}_2}{2} - \theta = \frac{\hat{\theta}_1 - \theta + \hat{\theta}_2 - \theta}{2}$$

$$\mathbb{E}\left[\left(\frac{\hat{\theta}_1 + \hat{\theta}_2}{2} - \theta\right)^2\right] = \frac{1}{4}\mathbb{E}[(\hat{\theta}_1 - \theta + \hat{\theta}_2 - \theta)^2]$$

$$= \frac{1}{4}\mathbb{E}\left[(\hat{\theta}_1 - \theta)^2 + 2(\hat{\theta}_1 - \theta)(\hat{\theta}_2 - \theta) + (\hat{\theta}_2 - \theta)^2\right]$$

$$= \frac{1}{4}\left(\mathrm{Var}(\hat{\theta}_1) + \mathrm{Var}(\hat{\theta}_2) + 2E\left[(\hat{\theta}_1 - \theta)(\hat{\theta}_2 - \theta)\right]\right)$$

$$= \frac{1}{2}\mathrm{Var}(\hat{\theta}_1) + \frac{1}{2}\mathbb{E}\left[(\hat{\theta}_1 - \theta)(\hat{\theta}_2 - \theta)\right] \tag{9}$$

where we have used $\mathbb{E}[\hat{\theta}_{\text{CF-PPI++}}] = \theta$ in writing the variance definition on the left hand side.

The first term is intuitive: The cross-fit estimator uses twice the samples of the estimate on just each fold, and so it has half the variance. The second term arises due to correlations in the errors of each cross-fit term. To analyze this term, we use subscripts to denote different datasets that serve as the source of different terms, e.g., $\bar{y}_1$ is the average of $y$ in the first split, and $\bar{y}_2$ is the average in the second split. Then the cross term is given by

$$\mathbb{E}[(\hat{\theta}_1 - \theta)(\hat{\theta}_2 - \theta)] = \mathbb{E}[((\bar{y}_2 - \theta) + \hat{\lambda}_1(\bar{f}_2 - \mu_f))((\bar{y}_1 - \theta) + \hat{\lambda}_2(\bar{f}_1 - \mu_f))]$$

$$= \mathbb{E}[(\bar{y}_2 - \theta)(\bar{y}_1 - \theta)]$$

$$+ \mathbb{E}[(\bar{y}_2 - \theta)(\hat{\lambda}_2(\bar{f}_1 - \mu_f))]$$

$$+ \mathbb{E}[(\bar{y}_1 - \theta)(\hat{\lambda}_1(\bar{f}_2 - \mu_f))]$$

$$+ \mathbb{E}[(\hat{\lambda}_1(\bar{f}_2 - \mu_f))(\hat{\lambda}_2(\bar{f}_1 - \mu_f))]$$

$$= \mathbb{E}[(\bar{y}_2 - \theta)]\mathbb{E}[(\bar{y}_1 - \theta)] \qquad\qquad \bar{y}_1 \perp\!\!\!\perp \bar{y}_2$$

$$+ \mathbb{E}[\hat{\lambda}_2(\bar{y}_2 - \theta)]\mathbb{E}[\bar{f}_1 - \mu_f] \qquad\qquad \bar{y}_2, \hat{\lambda}_2 \perp\!\!\!\perp \bar{f}_1$$

$$+ \mathbb{E}[\hat{\lambda}_1(\bar{y}_1 - \theta)]\mathbb{E}[\bar{f}_2 - \mu_f] \qquad\qquad \bar{y}_1, \hat{\lambda}_1 \perp\!\!\!\perp \bar{f}_2$$

$$+ \mathbb{E}[(\hat{\lambda}_1(\bar{f}_1 - \mu_f))]\mathbb{E}[(\hat{\lambda}_2(\bar{f}_2 - \mu_f))] \tag{10}$$

Note that the first three terms are zero, since they include mean-zero terms (e.g., $\mathbb{E}[\bar{y} - \theta] = 0, \mathbb{E}[\bar{f} - \mu_f] = 0$, etc). The fourth term is given by Lemma A.4, which we recall below:

**Lemma A.4.** *Given $\hat{\lambda}$ and $\bar{f}$ estimated on the same sample, we have it that*

$$\mathbb{E}[(\hat{\lambda}(\bar{f} - \mu_f))] = \frac{1}{n\sigma_f^2}\left(\sigma_{yf^2} - 2\sigma_{yf}\mu_f\right)$$

However, the above lemma assumes the sample means are computed on an $n$-sample. In the case of Cross-fit PPI++, the sample means are computed on $n' = n/2$ samples. Therefore,

$$\mathbb{E}[(\hat{\lambda}(\bar{f} - \mu_f))] = \frac{1}{(n/2)\sigma_f^2}\left(\sigma_{yf^2} - 2\sigma_{yf}\mu_f\right) = \frac{2}{n\sigma_f^2}\left(\sigma_{yf^2} - 2\sigma_{yf}\mu_f\right)$$

Plugging this expression back into Equation (10), we get the value of the cross term $\mathbb{E}[(\hat{\theta}_1 - \theta)(\hat{\theta}_2 - \theta)]$.

$$\mathbb{E}[(\hat{\theta}_1 - \theta)(\hat{\theta}_2 - \theta)] = \mathbb{E}[(\hat{\lambda}_1(\bar{f}_1 - \mu_f))]\mathbb{E}[(\hat{\lambda}_2(\bar{f}_2 - \mu_f))] \qquad\qquad \text{From Equation (10)}$$

$$= \frac{4}{n^2 \sigma_f^4} \left( \sigma_{yf^2} - 2\sigma_{yf}\mu_f \right)^2$$

Plugging this expression back into Equation (9) gives us the overall MSE of the cross-fit estimator.

$$\mathbb{E}\left[ \left( \frac{\hat{\theta}_1 + \hat{\theta}_2}{2} - \theta \right)^2 \right] = \frac{1}{2}\text{Var}(\hat{\theta}_1) + \frac{1}{2}\mathbb{E}\left[ (\hat{\theta}_1 - \theta)(\hat{\theta}_2 - \theta) \right] \qquad \text{From Equation (9)}$$

$$= \frac{1}{2}\left( \text{Var}(\hat{\theta}_1) + \frac{4}{n^2 \sigma_f^4} \left( \sigma_{yf^2} - 2\sigma_{yf}\mu_f \right)^2 \right)$$

$\square$

### A.5.2. VARIANCE OF A SINGLE-FOLD ESTIMATE $\hat{\theta}_1$

Having expressed the variance in terms of the variance of a single-fold estimate, we now characterize the variance of this single-fold estimate.

Consider the estimate on a single-fold from Definition 3.5, from which we have it that the variance can be decomposed as follows, (using $E[\hat{\theta}_k] = \theta$)

$$(\hat{\theta}_k - \theta) = \underbrace{(\bar{y}_{n/2} - \theta)}_{A} + \hat{\lambda}\underbrace{\left( \mu_f - \frac{1}{n/2}\sum_{i \in \mathcal{D}'_{n/2}} f_i \right)}_{D} = A + \hat{\lambda}D \qquad (11)$$

Such that the variance is given by

$$\text{Var}(\hat{\theta}_k) = \mathbb{E}[(\hat{\theta}_k - \theta)^2]$$
$$= \mathbb{E}[(A + \hat{\lambda}D)^2] \qquad \text{by Equation (11)}$$
$$= \mathbb{E}[A^2 + 2A\hat{\lambda}D + \hat{\lambda}^2 D^2]$$
$$= \mathbb{E}[A^2] + 2\mathbb{E}[A\hat{\lambda}D] + \mathbb{E}[\hat{\lambda}^2 D^2] \qquad \text{by Linearity of Expectation}$$

We can observe that

$$\mathbb{E}[A^2] = \mathbb{E}[(\bar{y}_{n/2} - \theta)^2] = \frac{\sigma_y^2}{n/2} = \frac{2\sigma_y^2}{n}$$

Consider the second term:

$$2\mathbb{E}[A\hat{\lambda}D] = 2\mathbb{E}[\hat{\lambda}(\bar{y}_{n/2} - \theta)(\mu_f - \bar{f}_{n/2})] \qquad (12)$$

Recall that in each split-estimate, $\hat{\lambda}$ is estimated on an independent split and is therefore independent of the other terms in the expectation. This allows us to apply Lemma A.2 to obtain

$$2\mathbb{E}[A\hat{\lambda}D] = 2\mathbb{E}[\hat{\lambda}(\bar{y}_{n/2} - \theta)(\mu_f - \bar{f}_{n/2})] = -2\frac{\sigma_{fy}^2}{(n/2)\sigma_f^2} = -4\frac{\sigma_{fy}^2}{n\sigma_f^2} \qquad (13)$$

where we have accordingly applied $n/2$ where the lemma uses $n$. It is also useful to note that while $\hat{\lambda}$ depends on the covariance estimate which also depends on the number of samples it is computed on, the expectation of the covariance estimate is independent of $n$ and is just $\sigma_{fy}$. Since only the expectation of $\hat{\lambda}$ appears in the lemma, there is no need to adjust additionally for this.

Now consider the third term: Since $\hat{\lambda}$ is independent of $D$, we have it that

$$\mathbb{E}[\hat{\lambda}^2 D^2] = \mathbb{E}[\hat{\lambda}^2]\mathbb{E}[(\mu_f - \bar{f}_{n/2})^2]$$

$$= \frac{\mathbb{E}[\hat{\sigma}_{fy}^2]}{\sigma_f^4} \cdot \frac{\sigma_f^2}{n/2}$$

$$\hat{\lambda} := \frac{\hat{\sigma}_{fy}}{\sigma_f^2}$$

$$= \frac{2\mathbb{E}[\hat{\sigma}_{fy}^2]}{n\sigma_f^2}$$

where we use the definition of $\hat{\lambda}$ and the fact that $\mathbb{E}[(\bar{f} - \mu_f)^2] = \sigma_f^2/(n/2)$ by definition of $\bar{f}$. Putting these together, we get

$$\begin{aligned}
\mathrm{Var}(\hat{\theta}_k) &= \frac{2\sigma_y^2}{n} - 4\frac{\sigma_{fy}^2}{n\sigma_f^2} + \frac{2\mathbb{E}[\hat{\sigma}_{fy}^2]}{n\sigma_f^2} \\
&= \frac{2\sigma_y^2}{n} - \frac{2\sigma_{fy}^2}{n\sigma_f^2} + \frac{2\mathbb{E}[\hat{\sigma}_{fy}^2]}{n\sigma_f^2} - \frac{2\sigma_{fy}^2}{n\sigma_f^2} \\
&= \frac{2\sigma_y^2}{n} - \frac{2\sigma_{fy}^2}{n\sigma_f^2} + \frac{2\mathbb{E}[\hat{\sigma}_{fy}^2 - \sigma_{fy}^2]}{n\sigma_f^2} \\
&= \frac{2\sigma_y^2}{n} - \frac{2\sigma_{fy}^2}{n\sigma_f^2} + \frac{2\mathbb{E}[(\hat{\sigma}_{fy} - \sigma_{fy})^2]}{n\sigma_f^2}
\end{aligned}$$

where the last line follows from the fact that $\mathbb{E}[(\hat{\sigma}_{fy} - \sigma_{fy})^2] = \mathbb{E}[\hat{\sigma}_{fy}^2 - 2\hat{\sigma}_{fy}\sigma_{fy} + \sigma_{fy}^2] = \mathbb{E}[\hat{\sigma}_{fy}^2 - \sigma_{fy}^2]$, from the fact that $\mathbb{E}[\hat{\sigma}_{fy}] = \sigma_{fy}$.

### A.5.3. FINITE-SAMPLE VARIANCE OF CROSS-FIT PPI++

Having characterized the variance of a single-fold estimate, we can plug this into the overall variance of the Cross-fit PPI++ to complete the proof as follows:

$$\mathrm{Var}(\hat{\theta}_{\text{CF-PPI++}}) = \mathbb{E}\left[\left(\frac{\hat{\theta}_1 + \hat{\theta}_2}{2} - \theta\right)^2\right] = \frac{1}{2}\left(\mathrm{Var}(\hat{\theta}_1) + \frac{4}{n^2\sigma_f^4}\left(\sigma_{yf^2} - 2\sigma_{yf}\mu_f\right)^2\right)$$

$$\begin{aligned}
\mathrm{Var}(\hat{\theta}_{\text{CF-PPI++}}) &= \frac{1}{2}\left(\frac{2\sigma_y^2}{n} - \frac{2\sigma_{fy}^2}{n\sigma_f^2} + \frac{2\mathbb{E}[(\hat{\sigma}_{fy} - \sigma_{fy})^2]}{n\sigma_f^2} + \frac{4}{n^2\sigma_f^4}\left(\sigma_{yf^2} - 2\sigma_{yf}\mu_f\right)^2\right) \\
&= \frac{\sigma_y^2}{n} - \frac{\sigma_{fy}^2}{n\sigma_f^2} + \frac{\mathbb{E}[(\hat{\sigma}_{fy} - \sigma_{fy})^2]}{n\sigma_f^2} + \frac{2}{n^2\sigma_f^4}\left(\sigma_{yf^2} - 2\sigma_{yf}\mu_f\right)^2
\end{aligned}$$

### A.6. Proof of : Condition for Worse Performance of Cross-fit PPI++

With the above theorem, it is easy to prove Corollary 4.2 which we restate here.

**Corollary A.2.** $\mathrm{Var}(\hat{\theta}_{\text{CF-PPI++}}) > \sigma_y^2/n$ if $\mathbb{E}[(\hat{\sigma}_{fy} - \sigma_{fy})^2] > \sigma_{fy}^2$ i.e., if the mean-squared error in estimating the covariance $\sigma_{fy}$ is higher than the squared covariance $\sigma_{yf}^2$.

From Theorem 4.1, we have

$$\mathrm{Var}(\hat{\theta}_{\text{CF-PPI++}}) = \frac{\sigma_y^2}{n} - \frac{\sigma_{fy}^2}{n\sigma_f^2} + \frac{\mathbb{E}[(\hat{\sigma}_{fy} - \sigma_{fy})^2]}{n\sigma_f^2} + \frac{2}{n^2\sigma_f^4}\left(\sigma_{yf^2} - 2\sigma_{yf}\mu_f\right)^2$$

Further, since $\frac{2}{n^2\sigma_f^4}\left(\sigma_{yf^2} - 2\sigma_{yf}\mu_f\right)^2 \geq 0$, we have

$$\mathrm{Var}(\hat{\theta}_{\text{CF-PPI++}}) - \frac{\sigma_y^2}{n} \geq \frac{\mathbb{E}[(\hat{\sigma}_{fy} - \sigma_{fy})^2]}{n\sigma_f^2} - \frac{\sigma_{fy}^2}{n\sigma_f^2}$$

Therefore if $\frac{\mathbb{E}[(\hat{\sigma}_{fy}-\sigma_{fy})^2]}{n\sigma_f^2} > \frac{\sigma_{fy}^2}{n\sigma_f^2}$, we get $\mathrm{Var}(\hat{\theta}_{\text{CF-PPI++}}) > \frac{\sigma_y^2}{n}$

## A.7. Proposition 4.4: Variance of Cross-fit PPI++ with Independent Pseudo-Labels

Motivated by the desire to build intuition, we first consider the case when $Y, F$ are independent, and present a proof in that scenario. Using Lemma A.6 in combination with Theorem 4.1 allows us to prove Proposition 4.4.

**Lemma A.6** (Covariance Estimation Error, Independent Case). *Given $n$ samples $\{y_i, f_i\}_{i=1}^n$ drawn iid from $P_{FY}$, where $Y$ and $F$ are independent with bounded second moments, then*

$$\mathbb{E}\left[(\hat{\sigma}_{yf}-\sigma_{yf})^2\right] = \frac{\sigma_y^2 \sigma_f^2}{n-1},$$

*Proof.* By definition, we can write that

$$\hat{\sigma}_{fy}(Y, F) = \frac{1}{n-1} \sum_{i=1}^n (y_i - \bar{y})(f_i - \bar{f}).$$

where $\bar{y} = n^{-1} \sum_{i=1}^n y_i$ and similar for $\bar{f}$. Taking the square gives us

$$\hat{\sigma}_{fy}^2 = \left( \frac{1}{n-1} \sum_{i=1}^n (y_i - \bar{y})(f_i - \bar{f}) \right)^2$$

$$= \frac{1}{(n-1)^2} \sum_{i=1}^n \sum_{j=1}^n (y_i - \bar{y})(f_i - \bar{f})(y_j - \bar{y})(f_j - \bar{f})$$

and by linearity of expectation, we therefore have it that

$$\mathbb{E}[\hat{\sigma}_{fy}^2] = \frac{1}{(n-1)^2} \sum_{i=1}^n \sum_{j=1}^n \mathbb{E}\left[(y_i - \bar{y})(f_i - \bar{f})(y_j - \bar{y})(f_j - \bar{f})\right]$$

Since we have independence of $F, Y$ (and hence, joint independence $(F_1, \ldots, F_n) \perp (Y_1, \ldots, Y_n)$), we can treat the expectations over terms involving $F, Y$ separately, giving us

$$\mathbb{E}[\hat{\sigma}_{fy}^2] = \frac{1}{(n-1)^2} \sum_{i=1}^n \sum_{j=1}^n \mathbb{E}\left[(y_i - \bar{y})(y_j - \bar{y})\right] \mathbb{E}\left[(f_i - \bar{f})(f_j - \bar{f})\right] \tag{14}$$

where

$$\mathbb{E}[(y_i - \bar{y})(y_j - \bar{y})] = \begin{cases} \sigma_y^2 \left(1 - \frac{1}{n}\right) & \text{if } i = j, \\ -\frac{\sigma_y^2}{n} & \text{if } i \neq j. \end{cases} \tag{15}$$

While Equation (15) is a standard result, it can be proven as follows: First, we distribute terms and use linearity of expectation to write

$$\mathbb{E}[(y_i - \bar{y})(y_j - \bar{y})] = \mathbb{E}[y_i y_j] - \mathbb{E}[y_i \bar{y}] - \mathbb{E}[y_j \bar{y}] + \mathbb{E}[\bar{y}^2]$$

and we can then observe that each term can be written as

$$\mathbb{E}[y_i y_j] = \begin{cases} \sigma_y^2 + \mu_y^2 & \text{if } i = j \\ \mu_y^2 & \text{if } i \neq j \end{cases}$$

$$\mathbb{E}[y_i \bar{y}] = \frac{1}{n}\left(\mathbb{E}[y_i^2] + (n-1)\mu_y^2\right) = \frac{1}{n}(\sigma_y^2 + n\mu_y^2)$$

$$\mathbb{E}[\bar{y}^2] = \frac{1}{n^2}\left(n(\sigma_y^2 + \mu_y^2) + n(n-1)\mu_y^2\right) = \frac{\sigma_y^2}{n} + \mu_y^2$$

We can then consider each case in Equation (15) as follows:

**Case 1:** $i = j$

$$\mathbb{E}[(y_i - \bar{y})^2] = (\sigma_y^2 + \mu_y^2) - 2 \cdot \frac{1}{n}(\sigma_y^2 + n\mu_y^2) + \left(\frac{\sigma_y^2}{n} + \mu_y^2\right) = \sigma_y^2\left(1 - \frac{1}{n}\right)$$

**Case 2:** $i \neq j$

$$\mathbb{E}[(y_i - \bar{y})(y_j - \bar{y})] = \mu_y^2 - 2 \cdot \frac{1}{n}(\sigma_y^2 + n\mu_y^2) + \left(\frac{\sigma_y^2}{n} + \mu_y^2\right) = -\frac{\sigma_y^2}{n}$$

which yields Equation (15), with a similar result holding for $F$. Returning to Equation (14), we can observe that there are $n$ terms where $i = j$ and $n(n-1)$ terms where $i \neq j$. Therefore, we obtain

$$\mathbb{E}[\hat{\sigma}_{fy}^2] = \frac{1}{(n-1)^2}\left[n \cdot \sigma_y^2\left(1 - \frac{1}{n}\right) \cdot \sigma_f^2\left(1 - \frac{1}{n}\right) + n(n-1) \cdot \left(-\frac{\sigma_y^2}{n}\right) \cdot \left(-\frac{\sigma_f^2}{n}\right)\right]$$

$$= \frac{1}{(n-1)^2}\left[n \cdot \sigma_y^2\sigma_f^2\frac{(n-1)^2}{n^2} + \sigma_y^2\sigma_f^2\frac{n(n-1)}{n^2}\right]$$

$$= \frac{n(n-1)}{n^2(n-1)^2}\sigma_y^2\sigma_f^2\left[n - 1 + 1\right]$$

$$= \frac{\sigma_y^2\sigma_f^2}{n-1}$$

which gives the desired result.

Finally, it is worth noting that if we estimate the covariance from $n' = c \cdot n$ samples, then the estimation error becomes,

$$\mathbb{E}[\hat{\sigma}_{fy}^2] = \frac{\sigma_y^2\sigma_f^2}{cn - 1}$$

$\square$

With Lemma A.6 in hand, we now prove Proposition 4.4. Note that we could have also written this proof using the more general result of Lemma 4.1 instead of Lemma A.6.

**Proposition 4.4.** *Given $n$ samples $\{y_i, f_i\}_{i=1}^n$ drawn from $P_{FY}$, where $Y$ and $F$ are independent with bounded second moments, the variance of* Cross-fit PPI++ *is given by*

$$\mathrm{Var}(\hat{\theta}_{CF\text{-}PPI++}) = \frac{\sigma_y^2}{n}\left(1 + \frac{2}{n-2}\right)$$

*Proof.* From Theorem 4.1, we have after applying zero covariance between $Y, F$,

$$\mathrm{Var}(\hat{\theta}_{\text{CF-PPI++}}) = \frac{\sigma_y^2}{n} + \frac{\mathbb{E}[(\hat{\sigma}_{fy} - \sigma_{fy})^2]}{n\sigma_f^2}$$

since any term involving a covariance between powers of $F$ and powers of $Y$ reduces to zero in the independent case.

Now, we have from Lemma A.6 that

$$\frac{\mathbb{E}[(\hat{\sigma}_{fy} - \sigma_{fy})^2]}{n\sigma_f^2} = \frac{1}{n\sigma_f^2} \cdot \frac{\sigma_y^2\sigma_f^2}{(n/2) - 1}$$

where we have adjusted the covariance estimation error to use $n/2$ samples as in the case of a single-fold of cross-fit. This gives us the desired result:

$$
\begin{aligned}
\mathrm{Var}(\hat{\theta}_{\text{CF-PPI++}}) &= \frac{\sigma_y^2}{n} + \frac{1}{n\sigma_f^2} \cdot \frac{\sigma_y^2 \sigma_f^2}{(n/2) - 1} \\
&= \frac{\sigma_y^2}{n} + \frac{2}{n(n-2)} \cdot \sigma_y^2 \\
&= \frac{\sigma_y^2}{n}\left(1 + \frac{2}{n-2}\right)
\end{aligned}
$$

$\square$

### A.8. Theorem 4.1: Covariance Estimation Error General

**Lemma 4.1** (Covariance Estimation Error, General Case). *Let $Y$ and $F$ have bounded fourth-moments. Given $n$ IID samples $\{y_i, f_i\}_{i=1}^n$ drawn from $P_{YF}$, we have*

$$
\begin{aligned}
\mathbb{E}[(\hat{\sigma}_{yf} - \sigma_{yf})^2] &= \frac{1}{n}\sigma_{f^2 y^2} + \frac{1}{n-1}\sigma_f^2 \sigma_y^2 - \frac{(n-2)}{n(n-1)}\sigma_{fy}^2 \\
&\quad - \frac{2}{n}[\sigma_{y^2 f}\mu_f + \sigma_{f^2 y}\mu_y - 2\sigma_{fy}\mu_f\mu_y]
\end{aligned}
$$

*Proof.* Noting that $\mathbb{E}[(\hat{\sigma}_{fy} - \sigma_{fy})^2] = \mathbb{E}[\hat{\sigma}_{fy}^2] - \sigma_{fy}^2$, we focus on first term.

$$
\begin{aligned}
\mathbb{E}[\hat{\sigma}_{fy}^2] &= \frac{1}{(n-1)^2}\sum_i\sum_j \mathbb{E}[(y_i f_i - \bar{y}\bar{f})(y_j f_j - \bar{y}\bar{f})] &&\text{by Lemma A.1} \\
&= \frac{1}{(n-1)^2}\sum_i\sum_j \mathbb{E}[(y_i f_i y_j f_j - y_i f_i \bar{y}\bar{f} - y_j f_j \bar{y}\bar{f} + (\bar{y}\bar{f})^2)] \\
&= \frac{1}{(n-1)^2}\left(\sum_i\sum_j \mathbb{E}[y_i f_i y_j f_j] - 2n^2\mathbb{E}[y_i f_i \bar{y}\bar{f}] + n^2\mathbb{E}[\bar{y}\bar{f}\bar{y}\bar{f}]\right) \\
&= \frac{1}{(n-1)^2}\left(n\mathbb{E}[y_i f_i y_i f_i] + n(n-1)\mathbb{E}[y_i f_i y_j f_j] - 2n^2\mathbb{E}[y_i f_i \bar{y}\bar{f}] + n^2\mathbb{E}[\bar{y}\bar{f}\bar{y}\bar{f}]\right) &&(16)
\end{aligned}
$$

where we use $\{(y_i, f_i), (y_j, f_j)\}$ as shorthand for any pair of samples $(y, f)$ that are independently drawn, as discussed in Remark A.1. We can further note that

$$
\mathbb{E}[\bar{y}\bar{f}\bar{y}\bar{f}] = \mathbb{E}[y_i \bar{f}\bar{y}\bar{f}] = \frac{1}{n}\mathbb{E}[y_i f_i \bar{y}\bar{f}] + \frac{n-1}{n}\mathbb{E}[y_i f_j \bar{y}\bar{f}]
$$

which we can combine with Equation (16) to obtain

$$
\begin{aligned}
&\mathbb{E}[\hat{\sigma}_{fy}^2] \\
&= \frac{1}{(n-1)^2}\left(n\mathbb{E}[y_i f_i y_i f_i] + n(n-1)\mathbb{E}[y_i f_i y_j f_j] - 2n^2\mathbb{E}[y_i f_i \bar{y}\bar{f}] + n^2\left(\frac{1}{n}\mathbb{E}[y_i f_i \bar{y}\bar{f}] + \frac{n-1}{n}\mathbb{E}[y_i f_j \bar{y}\bar{f}]\right)\right) \\
&= \frac{1}{(n-1)^2}\left(n\mathbb{E}[y_i f_i y_i f_i] + n(n-1)\mathbb{E}[y_i f_i y_j f_j] - (2n^2 - n)\mathbb{E}[y_i f_i \bar{y}\bar{f}] + n(n-1)\mathbb{E}[y_i f_j \bar{y}\bar{f}]\right) \\
&= \frac{1}{(n-1)^2}\left(n\mathbb{E}[y^2 f^2] + n(n-1)\mathbb{E}[yf]^2 - (2n^2 - n)\mathbb{E}[y_i f_i \bar{y}\bar{f}] + n(n-1)\mathbb{E}[y_i f_j \bar{y}\bar{f}]\right) \\
&= \frac{1}{(n-1)^2}\left(n\mathbb{E}[y^2 f^2] + n(n-1)\mathbb{E}[yf]^2 - n^2\mathbb{E}[y_i f_i \bar{y}\bar{f}] - n(n-1)(\mathbb{E}[y_i f_i \bar{y}\bar{f}] - \mathbb{E}[y_i f_j \bar{y}\bar{f}])\right) &&(17)
\end{aligned}
$$

We can further note that

$$\mathbb{E}[y_i f_i \bar{y} \bar{f}] = \frac{1}{n}\mathbb{E}[y_i f_i(y_i + \sum_{j \neq i} y_j)\bar{f}]$$

$$= \frac{1}{n}\left(\mathbb{E}[y_i^2 f_i \bar{f}] + (n-1)\mathbb{E}[y_i f_i y_j \bar{f}]\right)$$

$$= \frac{1}{n^2}\left(\mathbb{E}[y_i^2 f_i(f_i + \sum_{j \neq i} f_j)] + (n-1)\mathbb{E}[y_i f_i y_j(f_i + f_j + \sum_{k \neq (i,j)} f_k)]\right)$$

$$= \frac{1}{n^2}\left(\mathbb{E}[y^2 f^2] + (n-1)\mathbb{E}[y^2 f]\mathbb{E}[f] + (n-1)\mathbb{E}[y f^2]\mathbb{E}[y] + (n-1)\mathbb{E}[yf]^2 + (n-1)(n-2)\mathbb{E}[yf]\mathbb{E}[y]\mathbb{E}[f]\right) \quad (18)$$

and similarly,

$$\mathbb{E}[y_i f_j \bar{y} \bar{f}] = \frac{1}{n}\mathbb{E}[y_i f_j(y_i + y_j + \sum_{k \neq (i,j)} y_k)\bar{f}]$$

$$= \frac{1}{n}\left(\mathbb{E}[y_i^2 f_j \bar{f}] + \mathbb{E}[y_i f_j y_j \bar{f}] + (n-2)\mathbb{E}[y_i f_j y_k \bar{f}]\right)$$

$$= \frac{1}{n^2}\left(\mathbb{E}[y_i^2 f_j(f_i + f_j + \sum_{k \neq (i,j)} f_k)] + \mathbb{E}[y_i f_j y_j(f_i + f_j + \sum_{k \neq (i,j)} f_k)] + (n-2)\mathbb{E}[y_i f_j y_k(f_i + f_j + f_k + \sum_{l \neq (i,j,k)} f_l)]\right)$$

$$= \frac{1}{n^2}\left(\mathbb{E}[y^2 f]\mathbb{E}[f] + \mathbb{E}[y^2]\mathbb{E}[f^2] + (n-2)\mathbb{E}[y^2]\mathbb{E}[f]^2\right.$$

$$+ \mathbb{E}[yf]^2 + \mathbb{E}[yf^2]\mathbb{E}[y] + (n-2)\mathbb{E}[yf]\mathbb{E}[y]\mathbb{E}[f]$$

$$\left. + (n-2)\left(\mathbb{E}[yf]\mathbb{E}[y]\mathbb{E}[f] + \mathbb{E}[f^2]\mathbb{E}[y]^2 + \mathbb{E}[yf]\mathbb{E}[y]\mathbb{E}[f] + (n-3)\mathbb{E}[y]^2\mathbb{E}[f]^2\right)\right)$$

which gives us that the last term of Equation (17) is given by

$$n(n-1)(\mathbb{E}[y_i f_i \bar{y} \bar{f}] - \mathbb{E}[y_i f_j \bar{y} \bar{f}])$$

$$= \frac{n(n-1)}{n^2}\left(\mathbb{E}[y^2 f^2] + (n-1)\mathbb{E}[y^2 f]\mathbb{E}[f] + (n-1)\mathbb{E}[yf^2]\mathbb{E}[y] + (n-1)\mathbb{E}[yf]^2 + (n-1)(n-2)\mathbb{E}[yf]\mathbb{E}[y]\mathbb{E}[f]\right.$$

$$- \left(\mathbb{E}[y^2 f]\mathbb{E}[f] + \mathbb{E}[y^2]\mathbb{E}[f^2] + (n-2)\mathbb{E}[y^2]\mathbb{E}[f]^2\right.$$

$$+ \mathbb{E}[yf]^2 + \mathbb{E}[yf^2]\mathbb{E}[y] + (n-2)\mathbb{E}[yf]\mathbb{E}[y]\mathbb{E}[f]$$

$$\left.\left. + 2(n-2)\mathbb{E}[yf]\mathbb{E}[y]\mathbb{E}[f] + (n-2)\mathbb{E}[f^2]\mathbb{E}[y]^2 + (n-2)(n-3)\mathbb{E}[y]^2\mathbb{E}[f]^2\right)\right)$$

$$= \frac{n(n-1)}{n^2}\left(\left[\mathbb{E}[y^2 f^2] - \mathbb{E}[y^2]\mathbb{E}[f^2]\right] + (n-2)[\mathbb{E}[y^2 f]\mathbb{E}[f] - \mathbb{E}[y^2]\mathbb{E}[f]\mathbb{E}[f]] + (n-2)[\mathbb{E}[yf^2]\mathbb{E}[y] - \mathbb{E}[y]\mathbb{E}[f^2]\mathbb{E}[y]]\right.$$

$$\left. + (n-2)(\mathbb{E}[yf]^2 - \mathbb{E}[yf]\mathbb{E}[y]\mathbb{E}[f]) + (n-2)(n-3)[\mathbb{E}[yf]\mathbb{E}[y]\mathbb{E}[f] - \mathbb{E}[y]\mathbb{E}[f]\mathbb{E}[y]\mathbb{E}[f]]\right)$$

Noting that $\sigma_{ab} = \mathbb{E}[ab] - \mathbb{E}[a]\mathbb{E}[b]$, we can then rewrite the above as

$$n(n-1)(\mathbb{E}[y_i f_i \bar{y} \bar{f}] - \mathbb{E}[y_i f_j \bar{y} \bar{f}])$$

$$= \frac{n(n-1)}{n^2}\left(\sigma_{y^2 f^2} + (n-2)\sigma_{y^2 f}\mu_f + (n-2)\sigma_{yf^2}\mu_y + (n-2)\mathbb{E}[yf]\sigma_{yf} + (n-2)(n-3)\sigma_{yf}\mu_y\mu_f\right) \quad (19)$$

and we can combine Equations (18) and (19) with Equation (17) to obtain

$$\mathbb{E}[\hat{\sigma}_{fy}^2]$$

$$= \frac{1}{(n-1)^2}\left( n\mathbb{E}[y^2f^2] + n(n-1)\mathbb{E}[yf]^2 \right. \tag{From Eq. 17}$$

$$\left. - n^2\mathbb{E}[y_if_i\bar{y}\bar{f}] - n(n-1)(\mathbb{E}[y_if_i\bar{y}\bar{f}] - \mathbb{E}[y_if_j\bar{y}\bar{f}]) \right)$$

$$= \frac{1}{(n-1)^2}\left( n\mathbb{E}[y^2f^2] + n(n-1)\mathbb{E}[yf]^2 \right.$$

$$- \left( \mathbb{E}[y^2f^2] + (n-1)\mathbb{E}[y^2f]\mathbb{E}[f] + (n-1)\mathbb{E}[yf^2]\mathbb{E}[y] \right. \tag{From Eq. 18}$$

$$\left. + (n-1)\mathbb{E}[yf]^2 + (n-1)(n-2)\mathbb{E}[yf]\mathbb{E}[y]\mathbb{E}[f] \right)$$

$$- \frac{n(n-1)}{n^2}\left( \sigma_{y^2f^2} + (n-2)\sigma_{y^2f}\mu_f \right. \tag{From Eq. 19}$$

$$\left.\left. + (n-2)\sigma_{yf^2}\mu_y + (n-2)\mathbb{E}[yf]\sigma_{yf} + (n-2)(n-3)\sigma_{yf}\mu_y\mu_f \right) \right)$$

$$= \frac{1}{(n-1)^2}\left( (n-1)\mathbb{E}[y^2f^2] + (n-1)(n-1)\mathbb{E}[yf]^2 \right. \tag{Collect terms}$$

$$- \left( (n-1)\mathbb{E}[y^2f]\mathbb{E}[f] + (n-1)\mathbb{E}[yf^2]\mathbb{E}[y] + (n-1)(n-2)\mathbb{E}[yf]\mathbb{E}[y]\mathbb{E}[f] \right)$$

$$- \frac{n(n-1)}{n^2}\left( \sigma_{y^2f^2} + (n-2)\sigma_{y^2f}\mu_f \right.$$

$$\left.\left. + (n-2)\sigma_{yf^2}\mu_y + (n-2)\mathbb{E}[yf]\sigma_{yf} + (n-2)(n-3)\sigma_{yf}\mu_y\mu_f \right) \right)$$

$$= \frac{1}{n(n-1)}\left( n\mathbb{E}[y^2f^2] + n(n-1)\mathbb{E}[yf]^2 \right. \tag{Multiply by $\frac{n}{n}$}$$

$$- \left( n\mathbb{E}[y^2f]\mathbb{E}[f] + n\mathbb{E}[yf^2]\mathbb{E}[y] + n(n-2)\mathbb{E}[yf]\mathbb{E}[y]\mathbb{E}[f] \right)$$

$$- \left( \sigma_{y^2f^2} + (n-2)\sigma_{y^2f}\mu_f \right.$$

$$\left.\left. + (n-2)\sigma_{yf^2}\mu_y + (n-2)\mathbb{E}[yf]\sigma_{yf} + (n-2)(n-3)\sigma_{yf}\mu_y\mu_f \right) \right)$$

To further simplify, we expand terms like $\mathbb{E}[y^2f] = \sigma_{y^2f} + \mathbb{E}[y^2]\mathbb{E}[f] = \sigma_{y^2f} + (\sigma_y^2 + \mu_y^2)\mu_f$ and so on.

$$= \frac{1}{n(n-1)}\left( n\sigma_{y^2f^2} + n(\sigma_y^2 + \mu_y^2)(\sigma_f^2 + \mu_f^2) + n(n-1)(\sigma_{yf} + \mu_f\mu_y)(\sigma_{yf} + \mu_f\mu_y) \right.$$

$$- \left( n\sigma_{y^2f}\mu_f + n(\sigma_y^2 + \mu_y^2)\mu_f^2 + n\sigma_{yf^2}\mu_y + n(\sigma_f^2 + \mu_f^2)\mu_y^2 + n(n-2)(\sigma_{yf} + \mu_f\mu_y)\mu_f\mu_y \right)$$

$$- \left.\left( \sigma_{y^2f^2} + (n-2)\sigma_{y^2f}\mu_f + (n-2)\sigma_{yf^2}\mu_y + (n-2)(\sigma_{yf} + \mu_f\mu_y)\sigma_{yf} + (n-2)(n-3)\sigma_{yf}\mu_y\mu_f \right) \right)$$

We then expand and collect similar terms

$$= \frac{1}{n(n-1)}\left( \textcolor{blue}{n\sigma_{y^2f^2}} + n(\sigma_y^2\sigma_f^2 + \sigma_y^2\mu_f^2 + \sigma_f^2\mu_y^2 + \mu_y^2\mu_f^2) + n(n-1)(\sigma_{yf}^2 + 2\sigma_{yf}\mu_f\mu_y + \mu_f^2\mu_y^2) \right.$$

$$- \left( n\sigma_{y^2f}\mu_f + n(\sigma_y^2\mu_f^2 + \mu_f^2\mu_y^2) + n\sigma_{yf^2}\mu_y + n(\sigma_f^2\mu_y^2 + \mu_f^2\mu_y^2) + n(n-2)(\sigma_{yf}\mu_f\mu_y + \mu_f^2\mu_y^2) \right)$$

$$- \left.\left( \textcolor{blue}{\sigma_{y^2f^2}} + (n-2)\sigma_{y^2f}\mu_f + (n-2)\sigma_{yf^2}\mu_y + (n-2)(\sigma_{yf}^2 + \sigma_{yf}\mu_f\mu_y) + (n-2)(n-3)\sigma_{yf}\mu_y\mu_f \right) \right)$$

$$= \frac{\sigma_{y^2f^2}}{n} + \frac{\sigma_y^2\sigma_f^2}{n-1} + \frac{1}{n(n-1)}\left( n(\sigma_y^2\mu_f^2 + \sigma_f^2\mu_y^2 + \mu_y^2\mu_f^2) + n(n-1)(\textcolor{blue}{\sigma_{yf}^2} + 2\sigma_{yf}\mu_f\mu_y + \mu_f^2\mu_y^2) \right.$$

$$- \left( n\sigma_{y^2f}\mu_f + n(\sigma_y^2\mu_f^2 + \mu_f^2\mu_y^2) + n\sigma_{yf^2}\mu_y + n(\sigma_f^2\mu_y^2 + \mu_f^2\mu_y^2) + n(n-2)(\sigma_{yf}\mu_f\mu_y + \mu_f^2\mu_y^2) \right)$$

$$- \left( (n-2)\sigma_{y^2f}\mu_f + (n-2)\sigma_{yf^2}\mu_y + (n-2)(\sigma_{yf}^2 + \sigma_{yf}\mu_f\mu_y) + (n-2)(n-3)\sigma_{yf}\mu_y\mu_f \right) \Big)$$

$$= \frac{\sigma_{y^2f^2}}{n} + \frac{\sigma_y^2\sigma_f^2}{n-1} + \left( 1 - \frac{n-2}{n(n-1)} \right)\sigma_{yf}^2 + \frac{1}{n(n-1)} \left( n(\sigma_y^2\mu_f^2 + \sigma_f^2\mu_y^2 + \mu_y^2\mu_f^2) + n(n-1)(2\sigma_{yf}\mu_f\mu_y + \mu_f^2\mu_y^2) \right.$$

$$- \left( n\sigma_{y^2f}\mu_f + n(\sigma_y^2\mu_f^2 + \mu_f^2\mu_y^2) + n\sigma_{yf^2}\mu_y + n(\sigma_f^2\mu_y^2 + \mu_f^2\mu_y^2) + n(n-2)(\sigma_{yf}\mu_f\mu_y + \mu_f^2\mu_y^2) \right)$$

$$- \left( (n-2)\sigma_{y^2f}\mu_f + (n-2)\sigma_{yf^2}\mu_y + (n-2)(\sigma_{yf}\mu_f\mu_y) + (n-2)(n-3)\sigma_{yf}\mu_y\mu_f \right) \Big)$$

$$= \frac{\sigma_{y^2f^2}}{n} + \frac{\sigma_y^2\sigma_f^2}{n-1} + \left( 1 - \frac{n-2}{n(n-1)} \right)\sigma_{yf}^2 + \frac{1}{n(n-1)} \left( n(\sigma_y^2\mu_f^2 + \sigma_f^2\mu_y^2) + n(n-1)(2\sigma_{yf}\mu_f\mu_y) \right.$$

$$- \left( n\sigma_{y^2f}\mu_f + n\sigma_y^2\mu_f^2 + n\sigma_{yf^2}\mu_y + n\sigma_f^2\mu_y^2 + n(n-2)(\sigma_{yf}\mu_f\mu_y) \right)$$

$$- \left( (n-2)\sigma_{y^2f}\mu_f + (n-2)\sigma_{yf^2}\mu_y + (n-2)(\sigma_{yf}\mu_f\mu_y) + (n-2)(n-3)\sigma_{yf}\mu_y\mu_f \right) \Big)$$

$$= \frac{\sigma_{y^2f^2}}{n} + \frac{\sigma_y^2\sigma_f^2}{n-1} + \left( 1 - \frac{n-2}{n(n-1)} \right)\sigma_{yf}^2 + \frac{1}{n(n-1)} \left( n(n-1)(2\sigma_{yf}\mu_f\mu_y) \right.$$

$$- \left( n\sigma_{y^2f}\mu_f + n\sigma_{yf^2}\mu_y + n(n-2)(\sigma_{yf}\mu_f\mu_y) \right)$$

$$- \left( (n-2)\sigma_{y^2f}\mu_f + (n-2)\sigma_{yf^2}\mu_y + (n-2)(\sigma_{yf}\mu_f\mu_y) + (n-2)(n-3)\sigma_{yf}\mu_y\mu_f \right) \Big)$$

$$= \frac{\sigma_{y^2f^2}}{n} + \frac{\sigma_y^2\sigma_f^2}{n-1} + \left( 1 - \frac{n-2}{n(n-1)} \right)\sigma_{yf}^2 - \frac{1}{n}(2\sigma_{y^2f}\mu_f + 2\sigma_{yf^2}\mu_y)$$

$$+ \frac{1}{n(n-1)} \left( [2n(n-1) - n(n-2) - (n-2) - (n-2)(n-3)](\sigma_{yf}\mu_f\mu_y) \right)$$

$$= \frac{\sigma_{y^2f^2}}{n} + \frac{\sigma_y^2\sigma_f^2}{n-1} + \left( 1 - \frac{n-2}{n(n-1)} \right)\sigma_{yf}^2 - \frac{2}{n}(\sigma_{y^2f}\mu_f + \sigma_{yf^2}\mu_y - 2\sigma_{yf}\mu_f\mu_y) \tag{20}$$

where in the last line we use the fact that

$$\frac{1}{n(n-1)}[2n(n-1) - n(n-2) - (n-2) - (n-2)(n-3)]$$

$$= \frac{1}{n(n-1)}[2n^2 - 2n - n^2 + 2n - n + 2 - (n^2 - 5n + 6)]$$

$$= \frac{1}{n(n-1)}[-2n + 2n - n + 2 + 5n - 6]$$

$$= \frac{1}{n(n-1)}[-n + 2 + 5n - 6]$$

$$= \frac{1}{n(n-1)}[4(n-1)]$$

$$= \frac{4}{n}$$

Subtracting $\sigma_{yf}^2$ from the final expression for $\mathbb{E}[\hat{\sigma}_{yf}^2]$ in Equation (20), to obtain $\mathbb{E}[\hat{\sigma}_{yf}^2] - \sigma_{yf}^2 = \mathbb{E}[(\hat{\sigma}_{yf} - \sigma_{yf})^2]$, gives the desired result.

$\square$

### A.9. Section 4.1 Proofs for the Gaussian Setting

We begin with some notes on special properties of Gaussians. We then prove the results in Section 4.1 for Cross-fit PPI++ and then follow it up with the proofs for the Single Sample PPI++ portion of the statements.

For jointly normal random variables $X$ and $Y$ with means $\mu_x$, $\mu_y$, variances $\sigma_x^2$, $\sigma_y^2$, and covariance $\sigma_{xy}$, we have the following identities:

$$\mathbb{E}[X^2Y^2] = \mu_x^2\mu_y^2 + \sigma_x^2\mu_y^2 + \sigma_y^2\mu_x^2 + \sigma_x^2\sigma_y^2 + 2\sigma_{xy}^2 + 4\sigma_{xy}\mu_x\mu_y$$
$$\mathbb{E}[X^2Y] = \mu_x^2\mu_y + \sigma_x^2\mu_y + 2\sigma_{xy}\mu_x$$
$$\mathbb{E}[XY^2] = \mu_x\mu_y^2 + \sigma_y^2\mu_x + 2\sigma_{xy}\mu_y$$

Now consider the following expression for the expected value of the squared sample covariance obtained from Lemma 4.1:

$$\mathbb{E}\left[\hat{\sigma}_{fy}^2\right] = \frac{1}{n}\sigma_{f^2y^2} + \frac{1}{n-1}\sigma_f^2\sigma_y^2 + \left[\frac{n-1}{n} + \frac{1}{n(n-1)}\right]\sigma_{fy}^2$$
$$+ \frac{1}{n}[-2\sigma_{y^2f}\mu_f - 2\sigma_{f^2y}\mu_y + 4\sigma_{fy}\mu_f\mu_y]$$

Using these identities, we can simplify the covariance terms in our expression:

$$\begin{aligned}
\sigma_{f^2y^2} &= \mathbb{E}[F^2Y^2] - \mathbb{E}[F^2]\mathbb{E}[Y^2] \\
&= (\mu_f^2\mu_y^2 + \sigma_f^2\mu_y^2 + \sigma_y^2\mu_f^2 + \sigma_f^2\sigma_y^2 + 2\sigma_{fy}^2 + 4\sigma_{fy}\mu_f\mu_y) \\
&\quad - (\mu_f^2 + \sigma_f^2)(\mu_y^2 + \sigma_y^2) \\
&= \mu_f^2\mu_y^2 + \sigma_f^2\mu_y^2 + \sigma_y^2\mu_f^2 + \sigma_f^2\sigma_y^2 + 2\sigma_{fy}^2 + 4\sigma_{fy}\mu_f\mu_y \\
&\quad - \mu_f^2\mu_y^2 - \mu_f^2\sigma_y^2 - \sigma_f^2\mu_y^2 - \sigma_f^2\sigma_y^2 \\
&= 2\sigma_{fy}^2 + 4\sigma_{fy}\mu_f\mu_y
\end{aligned}$$

Similarly for the other covariance terms:

$$\begin{aligned}
\sigma_{f^2y} &= \mathbb{E}[F^2Y] - \mathbb{E}[F^2]\mathbb{E}[Y] \\
&= (\mu_f^2\mu_y + \sigma_f^2\mu_y + 2\sigma_{fy}\mu_f) - (\mu_f^2 + \sigma_f^2)\mu_y \\
&= \mu_f^2\mu_y + \sigma_f^2\mu_y + 2\sigma_{fy}\mu_f - \mu_f^2\mu_y - \sigma_f^2\mu_y \\
&= 2\sigma_{fy}\mu_f
\end{aligned}$$

and

$$\begin{aligned}
\sigma_{y^2f} &= \mathbb{E}[Y^2F] - \mathbb{E}[Y^2]\mathbb{E}[F] \\
&= (\mu_y^2\mu_f + \sigma_y^2\mu_f + 2\sigma_{fy}\mu_y) - (\mu_y^2 + \sigma_y^2)\mu_f \\
&= \mu_y^2\mu_f + \sigma_y^2\mu_f + 2\sigma_{fy}\mu_y - \mu_y^2\mu_f - \sigma_y^2\mu_f \\
&= 2\sigma_{fy}\mu_y
\end{aligned}$$

Now we substitute these expressions into our original equation:

$$\begin{aligned}
\mathbb{E}[\hat{\sigma}_{fy}^2] &= \frac{1}{n}(2\sigma_{fy}^2 + 4\sigma_{fy}\mu_f\mu_y) + \frac{1}{n-1}\sigma_f^2\sigma_y^2 \\
&\quad + \left[\frac{n-1}{n} + \frac{1}{n(n-1)}\right]\sigma_{fy}^2 \\
&\quad + \frac{1}{n}[-2(2\sigma_{fy}\mu_y)\mu_f - 2(2\sigma_{fy}\mu_f)\mu_y + 4\sigma_{fy}\mu_f\mu_y]
\end{aligned}$$

Simplifying the terms with $\mu_f$ and $\mu_y$:

$$\frac{1}{n}(4\sigma_{fy}\mu_f\mu_y - 4\sigma_{fy}\mu_y\mu_f - 4\sigma_{fy}\mu_f\mu_y + 4\sigma_{fy}\mu_f\mu_y) = 0 \tag{21}$$

So these terms cancel out completely. Now we simplify the terms with $\sigma_{fy}^2$:

$$\frac{2\sigma_{fy}^2}{n} + \left[\frac{n-1}{n} + \frac{1}{n(n-1)}\right]\sigma_{fy}^2 = \sigma_{fy}^2\left[\frac{2}{n} + \frac{n-1}{n} + \frac{1}{n(n-1)}\right]$$

$$= \sigma_{fy}^2\left[\frac{n+1}{n} + \frac{1}{n(n-1)}\right]$$

$$= \sigma_{fy}^2\left[\frac{(n+1)(n-1)+1}{n(n-1)}\right]$$

$$= \sigma_{fy}^2\left[\frac{n^2-1+1}{n(n-1)}\right]$$

$$= \sigma_{fy}^2\left[\frac{n^2}{n(n-1)}\right]$$

$$= \sigma_{fy}^2\left[\frac{n}{n-1}\right]$$

Therefore, for jointly normal random variables $F$ and $Y$, the Covariance Estimation Error becomes:

$$\mathbb{E}[\hat{\sigma}_{fy}^2] - \sigma_{fy}^2 = \frac{n}{n-1}\sigma_{fy}^2 + \frac{1}{n-1}\sigma_f^2\sigma_y^2 - \sigma_{fy}^2 = \frac{1}{n-1}(\sigma_{fy}^2 + \sigma_f^2\sigma_y^2) \tag{22}$$

or

$$\mathbb{E}[(\hat{\sigma}_{fy} - \sigma_{fy})^2] = \frac{n}{n-1}\sigma_{fy}^2 + \frac{1}{n-1}\sigma_f^2\sigma_y^2 - \sigma_{fy}^2 = \frac{1}{n-1}(\sigma_{fy}^2 + \sigma_f^2\sigma_y^2) \tag{23}$$

We are now ready to prove the results in Section 4.1. We begin with a proof for the Cross-fit PPI++ portion of the results. Then we treat the Single-sample case.

### A.9.1. PROOFS FOR THE CROSS-FIT PPI++ ESTIMATOR, GAUSSIAN

We now include an additional lemma for use,

**Lemma A.7** (Variance between the Folds, Gaussian)**.** *In the case of jointly Gaussian variables, $F, Y$ the variance between the folds for Cross-fit PPI++ vanishes.*

*Proof.* We have from Equation (21), that for Gaussians,

$$\sigma_{f^2y} = 2\sigma_{fy}\mu_f \implies \sigma_{f^2y} - 2\sigma_{fy}\mu_f = 0$$

The variance between the folds, from Theorem 4.1, specifically, Equation (2) shows this difference squared is in fact the cross term. This completes the proof of this lemma. $\qquad\square$

That the variance between the folds is zero has the following consequences for Cross-fit PPI++

1. Corollary 4.2 becomes a necessary condition and not just a sufficient condition. It was previously sufficient since the variance between the folds added an additional positive term to the variance of the Cross-fit PPI++ estimator. This term is now zero making the loss $>$ gain for worse performance from Cross-fit PPI++ a necessary condition.

2. This is now a reversible condition. That is, if $\mathbb{E}[(\hat{\sigma}_{fy} - \sigma_{fy})^2] < \sigma_{fy}^2$ then Cross-fit PPI++ performs better than Definition 3.1.

We now prove the condition for improvement using Cross-fit PPI++ in the Gaussian case.

**Proof of Corollary 4.1: Condition for Performant Cross-fit PPI++, Gaussian**

**Corollary 4.1** (Condition for Improvement, Gaussian Case). *Let $Y, F$ be jointly Gaussian random variables, and consider the Single Sample PPI++ and Cross-fit PPI++ estimators which both make use of $n$ labeled samples. Then,*

$$\text{Var}(\hat{\theta}_{PPI++}) < \text{Var}(\hat{\theta}_{Classical}) \iff \frac{1}{\sqrt{c \cdot n - 2}} < \left| \frac{\sigma_{fy}}{\sigma_y \sigma_f} \right|$$

*where $c = 1$ for Single Sample PPI++ and $c = 1/2$ for Cross-fit PPI++.*

*Proof.* While Equation (23) expresses the covariance estimation error when the covariance is estimated using $n$ samples, Cross-fit PPI++ uses only $n/2$ samples to estimate the covariance. In this case, the covariance estimation error becomes,

$$\mathbb{E}[(\hat{\sigma}_{fy} - \sigma_{fy})^2] = \frac{1}{n/2 - 1}(\sigma_{fy}^2 + \sigma_f^2 \sigma_y^2) = \frac{2}{n - 2}(\sigma_{fy}^2 + \sigma_f^2 \sigma_y^2) \tag{24}$$

Now Corollary 4.2 gives us the condition for Var of the Cross-fit PPI++ estimator to be higher than Var of the Classical estimator as

$\text{Var}(\hat{\theta}_{\text{CF-PPI++}}) > \sigma_y^2/n$ when $\mathbb{E}[(\hat{\sigma}_{fy} - \sigma_{fy})^2] > \sigma_{fy}^2$. For the Gaussian case, substituting Equation (24) in this, we get the condition for higher Var of Cross-fit PPI++ as

$$\frac{2}{n - 2}(\sigma_{fy}^2 + \sigma_f^2 \sigma_y^2) > \sigma_{fy}^2$$
$$2\sigma_{fy}^2 + 2\sigma_f^2 \sigma_y^2 > n\sigma_{fy}^2 - 2\sigma_{fy}^2$$
$$2\sigma_f^2 \sigma_y^2 > (n - 4)\sigma_{fy}^2$$

Or, the condition for lower variance is given by,

$$(n - 4)\sigma_{fy}^2 > 2\sigma_f^2 \sigma_y^2$$
$$\frac{\sigma_{fy}^2}{\sigma_f^2 \sigma_y^2} > \frac{2}{(n - 4)}$$
$$\left| \frac{\sigma_{fy}}{\sigma_f \sigma_y} \right| > \frac{\sqrt{2}}{\sqrt{n - 4}}$$
$$\left| \frac{\sigma_{fy}}{\sigma_f \sigma_y} \right| > \frac{1}{\sqrt{n/2 - 2}}$$
$$\left| \frac{\sigma_{fy}}{\sigma_f \sigma_y} \right| > \frac{1}{\sqrt{cn - 2}}$$

which completes the proof of Corollary 4.1 $\qquad\qquad\square$

**Proof of Proposition 4.1: Variance of Cross-fit PPI++, Gaussian**

Our covariance estimation error can be directly used in Theorem 4.1 to obtain the variance in the Gaussian case.

**Proposition 4.1** (Var of PPI++, Gaussian Case). *Let $Y, F$ be jointly Gaussian random variables, and consider the Single Sample PPI++ , Cross-fit PPI++ and Classical estimators which all make use of $n$ labeled samples. Then,*

$$\text{Var}(\hat{\theta}_{PPI++}) \tag{1}$$
$$= \text{Var}(\hat{\theta}_{Classical})\left(1 + \frac{1}{c \cdot n - 1}\right) - \frac{c \cdot n - 2}{n(c \cdot n - 1)} \cdot \frac{\sigma_{fy}^2}{\sigma_f^2}$$

*where $c = 1$ for Single Sample PPI++ and $c = 1/2$ for Cross-fit PPI++. Note that $\text{Var}(\hat{\theta}_{Classical}) = \sigma_y^2/n$*

*Proof.* Consider Equation (2). Now, from Lemma A.7 we know that its cross-term is zero. Further, from Equation (24) we obtain the covariance estimation error for the Cross-fit PPI++ case. Substituting these and simplifying should land the desired result. Thus, from Equation (2), we have

$$\text{Var}(\hat{\theta}_{\text{CF-PPI++}}) = \frac{\sigma_y^2}{n} - \frac{\sigma_{fy}^2}{n\sigma_f^2} + \frac{\mathbb{E}[(\hat{\sigma}_{fy} - \sigma_{fy})^2]}{n\sigma_f^2} + \frac{2}{n^2\sigma_f^4}\left(\sigma_{yf^2} - 2\sigma_{yf}\mu_f\right)^2$$

,

and we know from Lemma A.7, we have $\left(\sigma_{yf^2} - 2\sigma_{yf}\mu_f\right)^2 = 0$.

Further, from Equation (24), we get $\mathbb{E}[(\hat{\sigma}_{fy} - \sigma_{fy})^2] = \frac{2}{n-2}(\sigma_{fy}^2 + \sigma_f^2\sigma_y^2)$

Substituting, we obtain,

$$\begin{aligned}
\text{Var}(\hat{\theta}_{\text{CF-PPI++}}) &= \frac{\sigma_y^2}{n} - \frac{\sigma_{fy}^2}{n\sigma_f^2} + \frac{1}{n\sigma_f^2}\frac{2}{n-2}(\sigma_{fy}^2 + \sigma_f^2\sigma_y^2) \\
&= \frac{\sigma_y^2}{n} - \frac{\sigma_{fy}^2}{n\sigma_f^2}\cdot\frac{n-2}{n-2} + \frac{1}{n\sigma_f^2}\frac{2}{n-2}(\sigma_{fy}^2 + \sigma_f^2\sigma_y^2) \\
&= \frac{\sigma_y^2}{n} - \frac{(n-2-2)\,\sigma_{fy}^2}{n(n-2)\sigma_f^2} + \frac{2\sigma_y^2}{n(n-2)} \\
&= \frac{\sigma_y^2}{n}\left(1 + \frac{2}{n-2}\right) - \frac{n-4}{n(n-2)}\sigma_{fy}^2
\end{aligned}$$

To obtain the form that we want,

$$\begin{aligned}
\text{Var}(\hat{\theta}_{\text{CF-PPI++}}) &= \frac{\sigma_y^2}{n}\left(1 + \frac{2}{n-2}\right) - \frac{n-4}{n(n-2)}\sigma_{fy}^2 \\
&= \frac{\sigma_y^2}{n}\left(1 + \frac{1}{n/2-1}\right) - \frac{2(n/2-2)\sigma_{fy}^2}{2n(n/2-1)} \\
&= \frac{\sigma_y^2}{n}\left(1 + \frac{1}{n/2-1}\right) - \frac{(n/2-2)\sigma_{fy}^2}{n(n/2-1)}
\end{aligned}$$

Setting $c = 1/2$, we obtain

$$\text{Var}(\hat{\theta}_{\text{CF-PPI++}}) = \frac{\sigma_y^2}{n}\left(1 + \frac{1}{n/2-1}\right) - \frac{(n/2-2)\sigma_{fy}^2}{n(n/2-1)} \tag{25}$$

$$= \frac{\sigma_y^2}{n}\left(1 + \frac{1}{cn-1}\right) - \frac{(cn-2)\sigma_{fy}^2}{n(cn-1)} \tag{26}$$

which completes the proof. $\qquad\square$

Finally, the variance when $F, Y$ are independent Gaussians follows trivially from the above as follows.

Those complete the proofs for Cross-fit PPI++ in the Gaussian case. We now turn to the proof of Single Sample PPI++ portion of the statements from Section 4.1.

A.9.2. Proofs for the Single Sample PPI++ estimator, Gaussian

Given that we don't have a generalized expression for the variance of the Single Sample PPI++, our proof involves deriving this variance from first principles. The reason we are able to do so in this Gaussian setting lies in Lemma A.5. The independence of the sample covariance and the sample means results in $\hat{\lambda}$ being independent of $(\mu_f - \bar{f}_n)$ which dramatically simplifies the analysis. The rest of the proof mechanics is quite similar to our proof of Theorem 4.1 in A.5. Then we apply the conditions for improvement using this derived variance.

Consider the variance of the Single Sample PPI++ estimator: We have it that the errors in our estimator can be decomposed as follows

$$(\hat{\theta}_{\text{PPI++}} - \theta) = \underbrace{(\bar{y}_n - \theta)}_{A} + \hat{\lambda}\underbrace{\left(\mu_f - \frac{1}{n}\sum_{i \in \mathcal{D}_n} f_i\right)}_{D} = A + \hat{\lambda}D$$

Such that the mean squared error is given by

$$\begin{aligned}
\text{MSE}(\hat{\theta}_{\text{PPI++}}) &= \mathbb{E}[(\hat{\theta}_{\text{PPI++}} - \theta)^2] \\
&= \mathbb{E}[(A + \hat{\lambda}D)^2] \\
&= \mathbb{E}[A^2 + 2A\hat{\lambda}D + \hat{\lambda}^2 D^2] \\
&= \mathbb{E}[A^2] + 2\mathbb{E}[A\hat{\lambda}D] + \mathbb{E}[\hat{\lambda}^2 D^2] \qquad\qquad \text{by Linearity of Expectation}
\end{aligned}$$

We can observe that

$$\mathbb{E}[A^2] = \mathbb{E}[(\bar{y}_n - \theta)^2] = \frac{\sigma_y^2}{n}$$

is $\text{Var}(\hat{\theta}_{\text{Classical}})$ the classical variance. Now we consider the second term:

$$2\mathbb{E}[A\hat{\lambda}D] = 2\mathbb{E}[\hat{\lambda}(\bar{y}_n - \theta)(\mu_f - \bar{f}_n)]$$

From Lemma A.5 we have that since $Y, F$ are gaussian, $\hat{\lambda}$ is independent of $\bar{y}_n, \bar{f}_n$. Further, since $\theta, \mu_f$ are non-random, we have that $\hat{\lambda}$ is also independent of them. Overall, this gets us that $\hat{\lambda}$ is independent of $(\bar{y}_n - \theta)(\mu_f - \bar{f}_n)$. This allows us to apply Lemma A.2 with $n$ samples to obtain

$$2\mathbb{E}[A\hat{\lambda}D] = 2\mathbb{E}[\hat{\lambda}(\bar{y}_n - \theta)(\mu_f - \bar{f}_n)] = -2\frac{\sigma_{fy}^2}{n\sigma_f^2} = -\frac{2\sigma_{fy}^2}{n\sigma_f^2}$$

Now consider the third term: $\mathbb{E}[\hat{\lambda}^2 D^2]$, once again using Lemma A.5, $\hat{\lambda}$ is independent of $D$, we get $\mathbb{E}[\hat{\lambda}^2 D^2] = \mathbb{E}[\hat{\lambda}^2]\mathbb{E}[(\mu_f - \bar{f}_n)^2]$

$$\hat{\lambda} = \frac{\hat{\sigma}_{fy}}{\sigma_f^2}$$

where $\hat{\sigma}_{fy}$ is an unbiased estimate of the covariance $\sigma_{fy}$. Therefore,

$$\mathbb{E}[\hat{\lambda}^2] = \frac{\mathbb{E}[\hat{\sigma}_{fy}^2]}{\sigma_f^4}$$

Further, consider, $\mathbb{E}[(\bar{f}_n - \mu_f)^2]$. Since $\mu_f$ is the expected value of $\bar{f}_n$, this term reduces to the variance of $\bar{f}_n$ which is $\frac{\sigma_f^2}{n}$. Putting these together, we get

$$\begin{aligned}
\text{MSE}(\hat{\theta}_{\text{PPI++}}) &= \frac{\sigma_y^2}{n} - \frac{2\sigma_{fy}^2}{n\sigma_f^2} + \frac{\mathbb{E}[\hat{\sigma}_{fy}^2]}{\sigma_f^4} \cdot \frac{\sigma_f^2}{n} \\
&= \frac{\sigma_y^2}{n} - \frac{\sigma_{fy}^2}{n\sigma_f^2} + \frac{\mathbb{E}[\hat{\sigma}_{fy}^2]}{n\sigma_f^2} - \frac{\sigma_{fy}^2}{n\sigma_f^2}
\end{aligned}$$

$$= \mathsf{MSE}(\hat{\theta}_{\text{Classical}}) - \frac{\sigma_{fy}^2}{n\sigma_f^2} + \frac{\mathbb{E}[(\hat{\sigma}_{fy} - \sigma_{fy})^2]}{n\sigma_f^2} \tag{27}$$

Therefore, the variance of Single Sample PPI++ in the Gaussian case takes exactly the same form as Cross-fit PPI++. However, one important difference is that Single Sample PPI++ uses all $n$ available samples for estimating the covariance thus reducing $\mathbb{E}[(\hat{\sigma}_{fy} - \sigma_{fy})^2]$.

With this variance derived, we can easily prove the required theorems by simply combining it with the covariance estimation error that we derived.

**Proof of Proposition 4.1: Variance of Single Sample PPI++, Gaussian**

**Proposition 4.1** (Var of PPI++, Gaussian Case). *Let $Y, F$ be jointly Gaussian random variables, and consider the Single Sample PPI++ , Cross-fit PPI++ and Classical estimators which all make use of $n$ labeled samples. Then,*

$$\mathrm{Var}(\hat{\theta}_{PPI++}) \tag{1}$$

$$= \mathrm{Var}(\hat{\theta}_{Classical})\left(1 + \frac{1}{c \cdot n - 1}\right) - \frac{c \cdot n - 2}{n(c \cdot n - 1)} \cdot \frac{\sigma_{fy}^2}{\sigma_f^2}$$

*where $c = 1$ for Single Sample PPI++ and $c = 1/2$ for Cross-fit PPI++. Note that $\mathrm{Var}(\hat{\theta}_{Classical}) = \sigma_y^2/n$*

*Proof.* The covariance estimation error from Equation (23) is

$$\mathbb{E}[(\hat{\sigma}_{fy} - \sigma_{fy})^2] = \frac{1}{n-1}(\sigma_{fy}^2 + \sigma_f^2 \sigma_y^2)$$

Combining with Equation (27), we obtain

$$\begin{aligned}
\mathsf{MSE}(\hat{\theta}_{\text{PPI++}}) &= \frac{\sigma_y^2}{n} - \frac{\sigma_{fy}^2}{n\sigma_f^2} + \frac{\mathbb{E}[(\hat{\sigma}_{fy} - \sigma_{fy})^2]}{n\sigma_f^2} \\
&= \frac{\sigma_y^2}{n} - \frac{\sigma_{fy}^2}{n\sigma_f^2} + \frac{1}{n-1}(\sigma_{fy}^2 + \sigma_f^2 \sigma_y^2)\frac{1}{n\sigma_f^2} \\
&= \frac{\sigma_y^2}{n}\left(1 + \frac{1}{n-1}\right) - \frac{\sigma_{fy}^2}{n\sigma_f^2}\left(1 - \frac{1}{n-1}\right) \\
&= \frac{\sigma_y^2}{n}\left(1 + \frac{1}{n-1}\right) - \frac{\sigma_{fy}^2}{n\sigma_f^2}\left(\frac{n-2}{n-1}\right)
\end{aligned}$$

If we set $c = 1$, we obtain,

$$\mathsf{MSE}(\hat{\theta}_{\text{PPI++}}) = \frac{\sigma_y^2}{n}\left(1 + \frac{1}{cn-1}\right) - \frac{\sigma_{fy}^2}{n\sigma_f^2}\left(\frac{cn-2}{cn-1}\right)$$

which completes the proof. □

Further, setting $\sigma_{fy}$ to 0, we obtain

$$\mathsf{MSE}(\hat{\theta}_{\text{PPI++}}) = \frac{\sigma_y^2}{n}\left(1 + \frac{1}{cn-1}\right)$$

which completes the proof for the Single Sample PPI++ estimator.

Finally, we prove the condition for improvement,

**Proof of Corollary 4.1, Condition for variance improvement Single Sample PPI++ estimator**

**Corollary 4.1** (Condition for Improvement, Gaussian Case). *Let $Y, F$ be jointly Gaussian random variables, and consider the Single Sample PPI++ and Cross-fit PPI++ estimators which both make use of $n$ labeled samples. Then,*

$$\text{Var}(\hat{\theta}_{PPI++}) < \text{Var}(\hat{\theta}_{Classical}) \iff \frac{1}{\sqrt{c \cdot n - 2}} < \left| \frac{\sigma_{fy}}{\sigma_y \sigma_f} \right|$$

*where $c = 1$ for Single Sample PPI++ and $c = 1/2$ for Cross-fit PPI++.*

*Proof.* Now, $\text{MSE}(\hat{\theta}_{\text{PPI++}})$ is lower when

$$\sigma_{fy}^2 > \mathbb{E}[(\hat{\sigma}_{fy} - \sigma_{fy})^2]$$

This comes directly from Equation (27) The covariance estimation error from Equation (23) is

$$\mathbb{E}[(\hat{\sigma}_{fy} - \sigma_{fy})^2] = \frac{1}{n-1}(\sigma_{fy}^2 + \sigma_f^2 \sigma_y^2)$$

Therefore the required condition becomes,

$$\sigma_{fy}^2 > \frac{1}{n-1}(\sigma_{fy}^2 + \sigma_f^2 \sigma_y^2)$$

giving us,

$$(n-2)\sigma_{fy}^2 > \sigma_f^2 \sigma_y^2$$
$$|\frac{\sigma_{fy}}{\sigma_f \sigma_y}| > \frac{1}{\sqrt{n-2}}$$

$\square$

A.9.3. PROOF OF THE FINITE-N CASE

We now consider the setting where $N$ is finite.

**Proposition 4.2** (Condition for Improvement, Gaussian Case, Finite $N$). *Under the same conditions as Corollary 4.1, consider the Single Sample PPI++ where $N$ is finite. Then the condition for improvement becomes*

$$\text{Var}(\hat{\theta}_{Single\text{-}PPI++}) < \text{Var}(\hat{\theta}_{Classical}) \iff$$
$$\frac{1}{\sqrt{n - 2 - 8\frac{n-1}{N-1}}} < \left| \frac{\sigma_{fy}}{\sigma_y \sigma_f} \right|$$

*Proof.* Suppose we have $(y_1, f_1), \ldots, (y_n, f_n) \sim P_{FY}$, where $P_{FY}$ is a jointly Gaussian distribution with correlation $\rho_{fy}$. Let $\bar{y}_n$ denote the sample mean of $Y$ on this n-sample and $\bar{f}_n$ denote the sample mean of $f$ on this n-sample. Suppose similarly that we also observe an independent N-samples of $F$ from $P$, $f_1, \ldots, f_N \sim P_{FY}$. Similarly define the sample mean as $\bar{f}_N$.

The MSE of Single-Sample PPI++ can be written as,

$$\mathbb{E}[(\hat{\theta}_{\text{PPI++}} - \theta)^2] = \mathbb{E}[(\hat{\theta}_{\text{Classical}} - \theta)^2] + 2\underbrace{\mathbb{E}[\hat{\lambda}(\hat{\theta}_{\text{Classical}} - \theta)(\bar{f}_N - \bar{f}_n)]}_{T1} + \underbrace{\mathbb{E}[\hat{\lambda}^2(\bar{f}_N - \bar{f}_n)^2]}_{T2}$$

We use the following helper lemmas:

**Lemma A.8** (Independence Condition Empirical Covariance, Gaussian). *Let $Y, F$ be jointly normal. Suppose we observe $n$ samples $\{y_i, f_i\}_{i=1}^n$. Then the empirical covariance $\hat{\sigma}_{fy}$ is statistically independent of the sample mean $\bar{y}_n$ and $\bar{f}_n$.*

A special case of Lemma A.8 occurs when $Y = F$, we state this here in the context of the $N$ sample.

**Lemma A.9** (Independence Condition Empirical Variance, Gaussian). *Let $F$ be a random variable following a Gaussian distribution. Suppose we observe $N$ samples $\{f_i\}_{i=1}^N$. Then the empirical variance $\hat{\sigma}_f^2$ is statistically independent of the sample mean $\bar{f}_N$.*

Informally, these lemmas help us buy the statistical independence of $\hat{\lambda}$ from the difference in sample means $\bar{f}_N - \bar{f}_n$. Notice that the numerator in $\hat{\lambda}$ is composed of $\hat{\sigma}_{fy}$ which is trivially independent of $\bar{f}_N$ but now also independent of $\bar{f}_n$ using Lemma A.8. Further, the denominator in $\hat{\lambda}$ contains $\hat{\sigma}_f^2$ which from Lemma A.9 is independent of the sample mean difference. Therefore, $\hat{\lambda}$ is completely independent of the sample mean difference. We state this formally,

**Lemma A.10** (Independence Condition $\hat{\lambda}$ and $(\bar{f}_N - \bar{f}_n)$, Gaussian). *If $F, Y$ are jointly gaussian, then the estimate $\hat{\lambda}$ defined in Definition 3.4 is independent of $(\bar{f}_N - \bar{f}_n)$*

Let us now consider the term $T1$ in Equation (28)

$$
\begin{aligned}
T_1 &= \mathbb{E}[\hat{\lambda}]\mathbb{E}[(\bar{y}_n - \theta)(\bar{f}_N - \bar{f}_n)] \\
&= \mathbb{E}[\hat{\lambda}]\mathbb{E}[(\bar{y}_n - \theta)(\bar{f}_N - \mu_f + \mu_f - \bar{f}_n)] \\
&= \mathbb{E}[\hat{\lambda}]\left(\mathbb{E}[(\bar{y}_n - \theta)(\bar{f}_N - \mu_f)] - \mathbb{E}[(\bar{y}_n - \theta)(\bar{f}_n - \mu_f)]\right) \\
&= -\mathbb{E}[\hat{\lambda}]\mathbb{E}[(\bar{y}_n - \theta)(\bar{f}_n - \mu_f)] \\
&= -\mathbb{E}[\hat{\lambda}]\frac{1}{n}\sigma_{fy} \\
&= -\mathbb{E}\left[\frac{\hat{\sigma}_{fy}}{(1 + n/N)\hat{\sigma}_f^2}\right]\frac{1}{n}\sigma_{fy} \\
&= -\mathbb{E}[\hat{\sigma}_{fy}]\mathbb{E}\left[\frac{1}{(1 + n/N)\hat{\sigma}_f^2}\right]\frac{1}{n}\sigma_{fy} \\
&= -\frac{\sigma_{fy}^2}{n(1 + n/N)}\mathbb{E}\left[\frac{1}{\hat{\sigma}_f^2}\right] \\
&= -\frac{\sigma_{fy}^2}{n(1 + n/N)} \cdot \frac{N-1}{(N-3)\sigma_f^2} \\
&= -\frac{N-1}{n(1 + n/N)(N-3)}\rho_{fy}^2\sigma_y^2
\end{aligned}
$$

where we use the fact that $\frac{1}{\hat{\sigma}_f^2}$ follows an inverse chi-square distribution, which implies $\mathbb{E}\left[\frac{1}{\hat{\sigma}_f^2}\right] = \frac{N-1}{(N-3)\sigma_f^2}$, and in the last line we use the fact that $\sigma_{fy}^2/\sigma_f^2 = \rho_{fy}^2\sigma_y^2$

Now we consider $T_2$.

$$
\begin{aligned}
T_2 &= \mathbb{E}[\hat{\lambda}^2]\mathbb{E}[(\bar{f}_N - \bar{f}_n)^2] \\
&= \mathbb{E}[\hat{\lambda}^2]\left(\mathbb{E}[(\bar{f}_N - \mu_f)^2] + \mathbb{E}[(\bar{f}_n - \mu_f)^2]\right) \\
&= \mathbb{E}[\hat{\lambda}^2]\left(\frac{1}{N} + \frac{1}{n}\right)\sigma_f^2 \\
&= \mathbb{E}[\hat{\sigma}_{fy}^2]\mathbb{E}\left[\frac{1}{(1 + n/N)^2\hat{\sigma}_f^4}\right]\left(\frac{1}{N} + \frac{1}{n}\right)\sigma_f^2 \\
&= \mathbb{E}[\hat{\sigma}_{fy}^2]\frac{1}{(1 + n/N)^2} \cdot \frac{(N-1)^2}{(N-3)(N-5)\sigma_f^4}\left(\frac{1}{n} + \frac{1}{N}\right)\sigma_f^2 \\
&= \mathbb{E}[\hat{\sigma}_{fy}^2] \cdot \frac{(N-1)^2}{n(1 + n/N)(N-3)(N-5)\sigma_f^2}
\end{aligned}
$$

$$= \left( \frac{n}{n-1}\sigma_{fy}^2 + \frac{1}{n-1}\sigma_f^2\sigma_y^2 \right) \cdot \frac{(N-1)^2}{n(1+n/N)(N-3)(N-5)\sigma_f^2}$$

$$= \sigma_y^2 \left( \frac{n}{n-1}\rho_{fy}^2 + \frac{1}{n-1} \right) \cdot \frac{(N-1)^2}{n(1+n/N)(N-3)(N-5)}$$

here again we use the fact that $\frac{1}{\hat{\sigma}_f^2}$ follows an inverse chi-square distribution, which implies $\mathbb{E}\left[ \frac{1}{\hat{\sigma}_f^4} \right] = \frac{(N-1)^2}{(N-3)(N-5)\sigma_f^4}$.

Now we have the MSE

$$\mathbb{E}[(\hat{\theta}_{\text{PPI++}} - \theta)^2] = \text{MSE}(\hat{\theta}_{\text{Classical}}) - 2\frac{N-1}{n(1+n/N)(N-3)}\rho_{fy}^2\sigma_y^2 + \sigma_y^2 \left( \frac{n}{n-1}\rho_{fy}^2 + \frac{1}{n-1} \right) \frac{(N-1)^2}{n(1+n/N)(N-3)(N-5)}$$

$$= \text{MSE}(\hat{\theta}_{\text{Classical}}) + \sigma_y^2 \frac{N-1}{n(1+n/N)(N-3)} \left( -2\rho_{fy}^2 + \frac{n(N-1)}{(n-1)(N-5)}\rho_{fy}^2 + \frac{N-1}{(n-1)(N-5)} \right)$$

$$= \text{MSE}(\hat{\theta}_{\text{Classical}}) + \sigma_y^2 \frac{(N-1)^2}{n(n-1)(1+n/N)(N-3)(N-5)} \left( \frac{-2(n-1)(N-5)}{N-1}\rho_{fy}^2 + n\rho_{fy}^2 + 1 \right)$$

$$= \text{MSE}(\hat{\theta}_{\text{Classical}}) + \sigma_y^2 \frac{(N-1)^2}{n(n-1)(1+n/N)(N-3)(N-5)} \left( \frac{-2(n-1)(N-5)+n(N-1)}{N-1}\rho_{fy}^2 + 1 \right)$$

$$= \text{MSE}(\hat{\theta}_{\text{Classical}}) + \sigma_y^2 \frac{(N-1)^2}{n(n-1)(1+n/N)(N-3)(N-5)} \left( \frac{-2(nN-N-5n+5)+nN-n}{N-1}\rho_{fy}^2 + 1 \right)$$

$$= \text{MSE}(\hat{\theta}_{\text{Classical}}) + \sigma_y^2 \frac{(N-1)^2}{n(n-1)(1+n/N)(N-3)(N-5)} \left( 1 - \frac{nN-2N-9n+10}{N-1}\rho_{fy}^2 \right)$$

$$= \text{MSE}(\hat{\theta}_{\text{Classical}}) + \sigma_y^2 \frac{(N-1)^2}{n(n-1)(1+n/N)(N-3)(N-5)} \left[ 1 - \left( n-2-8\frac{n-1}{N-1} \right) \rho_{fy}^2 \right]$$

Looking at the last term, we have that the condition becomes

$$\rho_{fy}^2 \left( n - 2 - 8\frac{n-1}{N-1} \right) \geq 1 \tag{28}$$

$$|\rho_{fy}| \geq \frac{1}{\sqrt{n - 2 - 8\frac{n-1}{N-1}}} \tag{29}$$

$\square$

## A.10. Variance Estimates of Single Sample PPI++

We now consider the variance estimate of the Definition 3.4 estimator in two methods. The first method is based on the idea of $\lambda$ as a minimum variance linear combination, the spirit of PPI++. Here, we compute the theoretical variance of the PPI++ estimator assuming $\lambda$ is independent of the data, and then use all the required plug-ins, including $\hat{\lambda}$. In the second method, we naively compute the direct empirical plug-in in the Single Sample PPI++ estimator.

### A.10.1. PROPOSITION 4.6: DATA INDEPENDENT PLUG-IN

**Proposition 4.6** (Optimistic Variance Estimation of Single Sample PPI++). *The plug-in variance estimator for Single Sample PPI++ given by $\hat{\sigma}_{\text{Single-PPI++}}^2 = (1/n)[\hat{\sigma}_y^2 + \lambda^2\hat{\sigma}_f^2 - 2\lambda\hat{\sigma}_{fy}]$ reduces to $(1/n)[\hat{\sigma}_y^2 - \hat{\sigma}_{fy}^2/\hat{\sigma}_f^2]$, which is always less than the plug-in estimate of the classical variance $\hat{\sigma}_y^2/n$.*

*Proof.* We first calculate the variance of the estimator assuming $\lambda$ is fixed. Then we evaluate the result upon substituting the data-dependent plug-in $\hat{\lambda}$ and plug-ins for other quantities involved.

**Lemma A.11** (Variance of PPI++). *The variance of the PPI++ estimator defined in Definition 3.3 is given by*

$$\sigma_{\hat{\theta}_{PPI++}}^2 = \frac{\sigma_y^2}{n} + \lambda^2 \left( \frac{\sigma_f^2}{N} + \frac{\sigma_f^2}{n} \right) - 2\lambda\frac{\sigma_{fy}}{n}. \tag{30}$$

$$\hat{\theta}_{\text{PPI++}} = \frac{1}{n} \sum_{i=1}^{n} y_i + \lambda \left( \frac{1}{N} \sum_{i=1}^{N} f_i - \frac{1}{n} \sum_{i=1}^{n} f_i \right). \tag{31}$$

To compute its variance:

$$\text{Var}(\hat{\theta}_{\text{PPI++}}) = \text{Var} \left( \frac{1}{n} \sum_{i=1}^{n} y_i \right) + \lambda^2 \text{Var} \left( \frac{1}{N} \sum_{i=1}^{N} f_i \right) + \lambda^2 \text{Var} \left( \frac{1}{n} \sum_{i=1}^{n} f_i \right) \tag{32}$$

$$- 2\lambda \text{Cov} \left( \frac{1}{n} \sum_{i=1}^{n} y_i, \frac{1}{n} \sum_{i=1}^{n} f_i \right). \tag{33}$$

Using the standard variance formulas for sample means:

$$\text{Var} \left( \frac{1}{n} \sum_{i=1}^{n} y_i \right) = \frac{\sigma_y^2}{n}, \tag{34}$$

$$\text{Var} \left( \frac{1}{N} \sum_{i=1}^{N} f_i \right) = \frac{\sigma_f^2}{N}, \tag{35}$$

$$\text{Var} \left( \frac{1}{n} \sum_{i=1}^{n} f_i \right) = \frac{\sigma_f^2}{n}, \tag{36}$$

$$\text{Cov} \left( \frac{1}{n} \sum_{i=1}^{n} Y_i, \frac{1}{n} \sum_{i=1}^{n} f(X_i) \right) = \frac{\sigma_{fy}}{n}. \tag{37}$$

Therefore, putting it all together, we get the variance when $\lambda$ is fixed,

$$\sigma_{\hat{\theta}_{\text{PPI++}}}^2 = \frac{\sigma_y^2}{n} + \lambda^2 \left( \frac{\sigma_f^2}{N} + \frac{\sigma_f^2}{n} \right) - 2\lambda \frac{\sigma_{fy}}{n}. \tag{38}$$

The plug-in estimate of the variance of the Single-Sample estimator considers the variance of the PPI++ estimator as given in Equation (30) with plug-ins for all quantities (including $\lambda$ computed from data).

Therefore, substitute plug-ins and use $\hat{\lambda} = \frac{\hat{\sigma}_{fy}}{(1 + \frac{n}{N}) \hat{\sigma}_f^2}$, gives us,

$$\hat{\sigma}^2_{\hat{\theta}_{\text{PPI++}}} = \frac{\hat{\sigma}_y^2}{n} - \frac{\left( \frac{\hat{\sigma}_{fy}}{n} \right)^2}{\frac{\hat{\sigma}_f^2}{N} + \frac{\hat{\sigma}_f^2}{n}}. \tag{39}$$

$$\hat{\sigma}^2_{\hat{\theta}_{\text{PPI++}}} = \frac{\hat{\sigma}_y^2}{n} - \frac{\left( \frac{\hat{\sigma}_{fy}}{n} \right)^2}{\left( \frac{1}{N} + \frac{1}{n} \right) \hat{\sigma}_f^2}. \tag{40}$$

$$\hat{\sigma}^2_{\hat{\theta}_{\text{PPI++}}} = \frac{\hat{\sigma}_y^2}{n} - \frac{\left( \frac{\hat{\sigma}_{fy}}{n} \right)^2}{\left( \frac{1}{N} + \frac{1}{n} \right) \hat{\sigma}_f^2}.$$

Further, if we let $N \to \infty$, $\hat{\sigma}_f^2 \to \sigma_f^2$ and $1/N \to 0$, giving us,

$$\hat{\sigma}^2_{\hat{\theta}_{\text{PPI++}}} = \frac{\hat{\sigma}_y^2}{n} - \frac{\hat{\sigma}_{fy}^2}{n \sigma_f^2}$$

Since, $\frac{\hat{\sigma}_y^2}{n}$ is the estimate of the variance of the classical estimator, this shows that the estimate of the variance of the Single Sample PPI++ is always lower than its classical counterpart.

$\square$

### A.10.2. NAIVE PLUG-IN

We now consider an alternate method for computing the empirical variance. Starting with the PPI++ definition, assuming again that $\lambda$ is fixed, we rewrite it as a sum of two independent terms.

$$\hat{\theta}_{\text{PPI++}} = (\bar{y} - \lambda \bar{f}_n) + \lambda \bar{f}_N$$

Now, consider the following standard method for estimating variance, by taking the empirical variance of $\bar{y} - \lambda \bar{f}_n$ and the empirical variance of $\lambda \bar{f}_N$, and adding up the empirical variance estimates, adjusted for sample size. The estimate of the variance of the former is given by

$$\hat{\sigma}_{y-\lambda f}^2 = \frac{1}{n-1} \sum_i ((y_i - \lambda f_i) - (\bar{y} - \lambda \bar{f}))^2$$

$$= \frac{1}{n-1} \sum_i ((y_i - \bar{y}) - (\lambda f_i - \lambda \bar{f}))^2$$

$$= \frac{1}{n-1} \sum_i (y_i - \bar{y})^2 - 2\lambda(y_i - \bar{y})(f_i - \bar{f}) + \lambda^2 (f_i - \bar{f})^2$$

$$= \hat{\sigma}_y^2 + \lambda^2 \hat{\sigma}_f^2[n] - 2\lambda \hat{\sigma}_{yf}$$

where $\hat{\sigma}_y^2, \hat{\sigma}_f^2[n], \hat{\sigma}_{yf}$ are the empirical variances of $Y$ and $F$ and the empirical covariance respectively, where we note that $\hat{\sigma}_f^2[n]$ is the empirical variance on $n$ samples. Using the fact that $\lambda := \hat{\sigma}_{yf}/((1 + n/N)\hat{\sigma}_f^2[N])$, we have it that

$$\hat{\sigma}_{y-\lambda f}^2 = \hat{\sigma}_y^2 + \lambda^2 \hat{\sigma}_f^2[n] - 2\lambda \hat{\sigma}_{yf}$$

$$= \hat{\sigma}_y^2 + \left(\frac{\hat{\sigma}_{yf}^2}{(1 + \frac{n}{N})^2 \hat{\sigma}_f^4[N]}\right) \hat{\sigma}_f^2[n] - 2\left(\frac{\hat{\sigma}_{yf}}{(1 + \frac{n}{N})\hat{\sigma}_f^2[N]}\right) \hat{\sigma}_{yf}$$

$$= \hat{\sigma}_y^2 + \left(\frac{\hat{\sigma}_{yf}^2}{(1 + \frac{n}{N})^2 \hat{\sigma}_f^4[N]}\right) \hat{\sigma}_f^2[n] - 2\left(\frac{\hat{\sigma}_{yf}^2}{(1 + \frac{n}{N})\hat{\sigma}_f^2[N]}\right)$$

$$= \hat{\sigma}_y^2 + \left(\frac{\hat{\sigma}_{yf}^2}{(1 + \frac{n}{N})^2 \hat{\sigma}_f^4[N]}\right) \hat{\sigma}_f^2[n] - 2\hat{\sigma}_f^2[N]\left(1 - \frac{n}{N}\right)\left(\frac{\hat{\sigma}_{yf}^2}{(1 + \frac{n}{N})^2 \hat{\sigma}_f^4[N]}\right)$$

$$= \hat{\sigma}_y^2 - \left(2\hat{\sigma}_f^2[N]\left(1 - \frac{n}{N}\right) - \hat{\sigma}_f^2[n]\right)\left(\frac{\hat{\sigma}_{yf}^2}{(1 + \frac{n}{N})^2 \hat{\sigma}_f^4[N]}\right)$$

$$= \hat{\sigma}_y^2 - \left(2\hat{\sigma}_f^2[N]\left(1 - \frac{n}{N}\right) - \hat{\sigma}_f^2[n]\right)\hat{\lambda}^2$$

Then the empirical variance of $\hat{\sigma}_{\lambda f}^2$, computed on $N$ samples, is given by $\lambda^2 \hat{\sigma}_f^2[N]$, so that the total variance (dividing by $n$ and $N$ respectively, and then taking the sum), is given by

$$\frac{\hat{\sigma}_{y-\lambda f}^2}{n} + \frac{\hat{\sigma}_{\lambda f}^2}{N} = \frac{\hat{\sigma}_y^2}{n} - \frac{\left(2\hat{\sigma}_f^2[N]\left(1 - \frac{n}{N}\right)\right)\hat{\lambda}^2}{n} + \frac{\hat{\lambda}^2 \hat{\sigma}_f^2[n]}{n} + \frac{\hat{\lambda}^2 \hat{\sigma}_f^2[N]}{N}$$

If $N \to \infty$, then $\frac{\hat{\sigma}_{\lambda f}^2}{N} \to 0$, $1 - \frac{n}{N} \to 1$, $\frac{\hat{\lambda}^2 \hat{\sigma}_f^2[N]}{N} \to 0$, and $\sigma_f^2[N] \to \sigma_f^2$, and hence the empirical variance estimate becomes,

$$\frac{\hat{\sigma}_y^2}{n} - \frac{2\hat{\lambda}^2 \sigma_f^2}{n} + \frac{\hat{\lambda}^2 \sigma_f^2[n]}{n}$$

This is always less than the empirical estimate of the classical variance as long as the empirical variance of $F$ on $n$ samples is no greater than double the true variance.

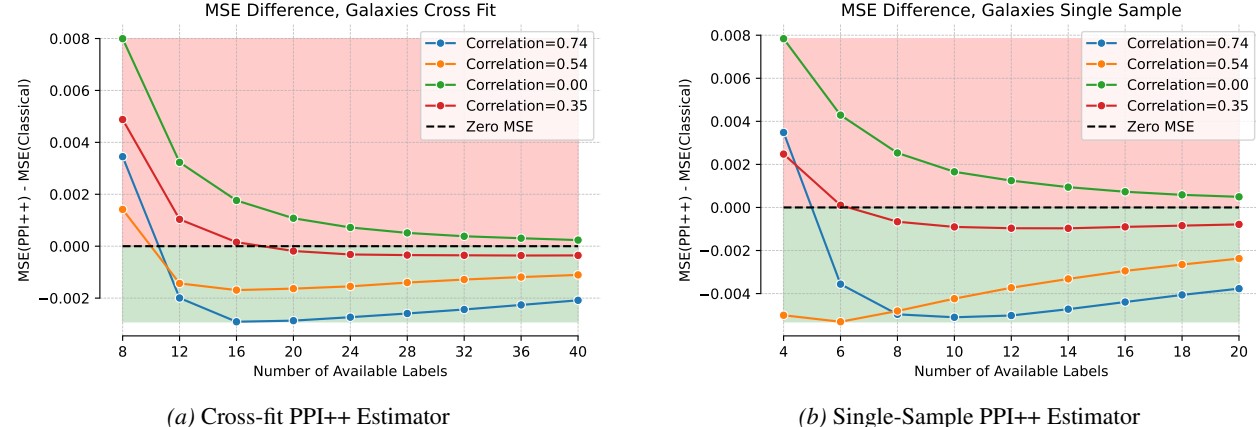

*(a)* Cross-fit PPI++ Estimator        *(b)* Single-Sample PPI++ Estimator

*Figure 4.* Relative MSE vs. Sample Size on the Galaxies Dataset for PPI++ estimators with black-box models $f$ of varying quality. Y-axis gives the difference $\mathsf{MSE}(\hat{\theta}_{\mathrm{PPI++}}) - \mathsf{MSE}(\hat{\theta}_{\mathrm{Classical}})$, such that lower (negative) values imply improvement over the classical estimator. Each line represents PPI++ with a different black-box model $f$. The blue line uses the original model, which has strong predictive performance, but only improves estimation error at $n \geq 12$ (left) and $n \geq 6$ (right).

## B. Experiment Setup Details

In this section, we provide details of the experiment setup on Alphafold (Jumper et al., 2021). For both the MSE as well as the coverage experiments, our results are bootstrapped over 50,000 draws. To closely match the theoretical intuition, we aim to study the setting where $n$ is small and $N$ is large. Therefore, after randomly sampling $n$ (or $2n$, for sample-splitting), we treat all the remaining data as unlabeled. Since the true labels are binary, we also use a binary noise model for the configurations that do not use the original predictor. This noise model is implemented by specifying $\mathbb{P}(F = 1|Y = 1)$ and $\mathbb{P}(F = 1|Y = 0)$. Alternatively, this can be viewed as specifying the True Positive Rate and the False Positive Rate of the pseudo-labels. From a noisy label lens, this can be viewed as specifying the complete noise transition matrix. For each configuration, for each $n$, we reset random states, ensuring full reproducibility. For all plug-in statistical quantities computed, we use *ddof = 1* for unbiasedness.

## C. Additional Experiments: Galaxies Dataset

In this section, we extend our experiments to the galaxies dataset used in Angelopoulos et al. (2023b). Similar insights hold as in Section 5. Our experiment apparatus is the same as described in Section B. We run $50,000$ draws using small-regime of $n$ or $2n$ and all the remaining data as unlabeled. We use a similar binary noise model. We show our results in Figures 4 and 5.

## D. Additional Experiments: Coverage and Interval Width, Alphafold Noised

In this section, we present the coverage and interval width results on Alphafold, similar to Figure 3. We note that in this lowered correlation setting, while Cross-fit PPI++ and Single Sample PPI++ widen their interval width to improve coverage, Cross-fit PPI++ matches the coverage of Classical but Single Sample PPI++ still falls short. The experiment details are as presented in Section 5 and Section B.

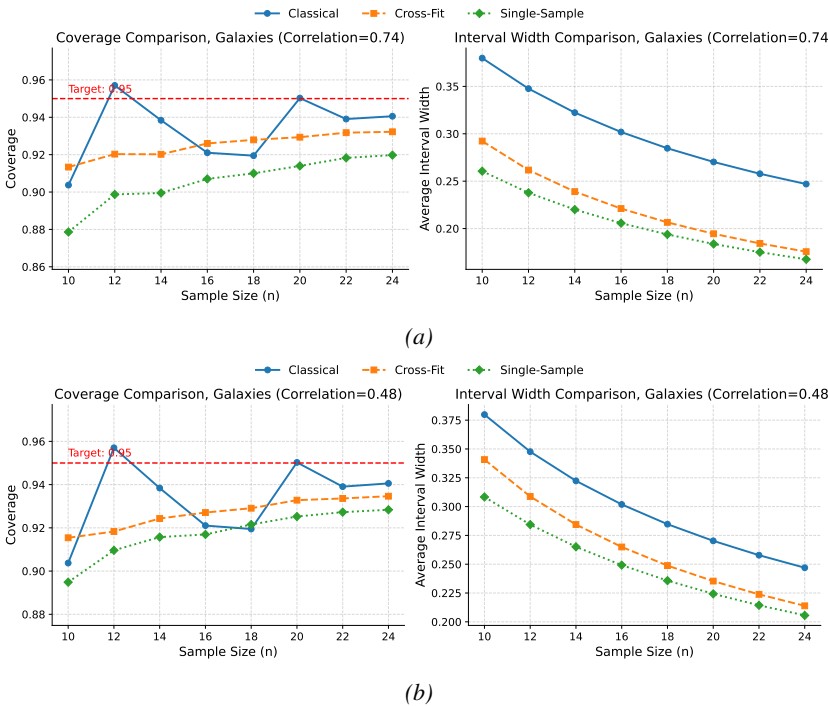

*Figure 5.* Comparison of Coverage and Interval Width for Cross-fit PPI++ and Single Sample PPI++, using different models $f$ on the Galaxies dataset. (a) Original pseudo-label model $f$, where Single-Sample PPI++ has lower coverage than either Cross-fit PPI++ or the classical estimator. (b) A modified model with lower correlation, where similar trends hold.

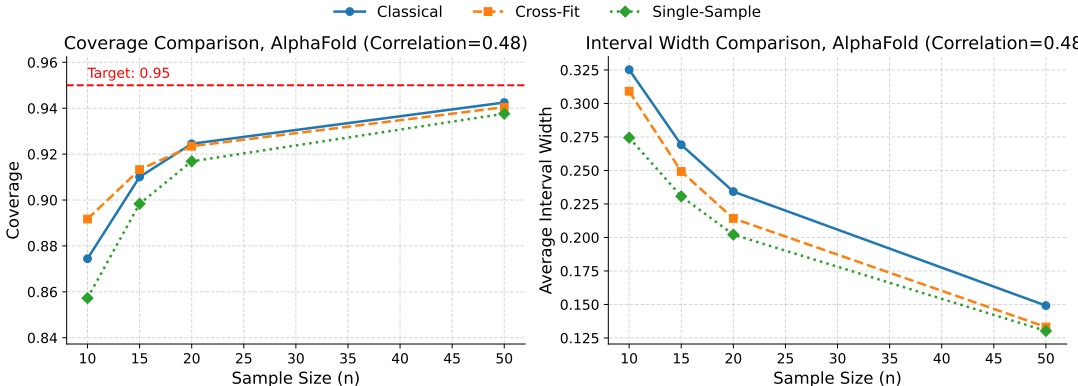

*Figure 6.* **AlphaFold.** Coverage (left) and interval width (right) for Cross-fit PPI++ vs. Single Sample PPI++ using noised pseudo-labels from Alphafold (corr = 0.48). Single Sample PPI++ under-covers relative to Cross-fit PPI++ and Classical.

