# OpenReview forum: "No Free Lunch: Non-Asymptotic Analysis of Prediction-Powered Inference"
_ICML.cc/2026/Conference — ICML 2026 regular_

### Official Review · Reviewer_wXiH · 2026-03-09

**Soundness:** 3
**Presentation:** 3
**Significance:** 2
**Originality:** 3
**Overall Recommendation:** 3
**Confidence:** 3

**Summary:**

Estimating the mean of a random variable is an important task across various fields. The recently proposed PPI is a framework of performing such estimation in a modern setting with both labeled and unlabeled data. PPI++ improves over PPI by deriving the optimal weights for control variates. However, PPI++ is not a free lunch. The weights are also estimated from data and could be detrimental to the estimation when the sample size of labeled data is small. This work verifies this known intuition by deriving the finite-sample variances for PPI++. For a Gaussian setting, the variance of single-sample PPI++ (estimate weights from all data) and cross-fit PPI++ (estimate weights from held-out data) are derived. For the general case, the variance of cross-fit PPI++ and the bias of single-sample PPI++ are derived. In general, the correlation between the labels and the groundtruth determines how many samples are needed to see improvements from PPI++. And single-sample PPI++ gives more narrow confidence interval because of the over-confident variance estimate. The experiments focus on low sample regime where $n<50$. The results align with the theories and PPI++ works better than PPI after having about 20~30 samples.

**Compliance With Llm Reviewing Policy:**

Affirmed.

**Final Justification:**

The rebuttal has addressed my questions about significance. However, I still feel the scope is limited.

**Key Questions For Authors:**

- Are there real applications (with reasonable amount of labeled data) where PPI++ makes the estimation worse?
- The cross-fit PPI++ splits the data evenly. It will also be interesting to explore other splitting ratios, or even adaptive ones. How will this modification affect the results?
- How is this work connected to the control variate literature in machine learning?

**Limitations:**

yes

**Strengths And Weaknesses:**

Strengths:
- This work fills a gap in understanding the finite-sample performance of a popular algorithm widely applied in sciences.
- The presentation of the work is clear. The progression from Gaussian case to non-Gaussian case is natural for understanding.

Weaknesses:
- I have concerns about the significance of the work. It is already well-known that the estimated control variate weights only work with enough data. The work quantifies the finite-sample variance of the PPI++ estimator, but it is unclear how anyone can use the derived variance. As the authors mentioned, using the derived expressions to make decisions also violates the "no free lunch" principle.
- Although control variates are widely used in many fields of machine learning, including optimization, reinforcement learning, and federated learning, the work focuses exclusively on the specialized estimator PPI++. I do not see the reason to restrict the analysis to a specific algorithm.
- The experiments verify the theoretical results, however, I do not think the small sample setting is meaningful. Existing works in PPI aim at making a scientific discovery and often need hundreds of labeled data. This work does not show an application where small sample setting is relevant.

---

> ### Author Rebuttal · Authors · 2026-03-31
>
> Thank you for the review.  We address your questions and points below:
>
> **(W1) …it is unclear how anyone can use the derived variance. As the authors mentioned, using the derived expressions to make decisions also violates the "no free lunch" principle.**
>
> Please see our response to jRz6 on this point (their Q2/W2).  In short, we see our approach as akin to "power analysis" as commonly practiced in the design of statistical studies, where related datasets can be used to inform beliefs about as-of-yet-unobserved quantities (like psuedo-label quality in our case).
>
> For instance, suppose $F$ is a hallucination detector, and we want to estimate the hallucination rate of a LLM-based system.  We may have prior labeled data from evaluating an older system (call it System 1), where $F$ had strong sensitivity/specificity.  We can then use that data to inform our assumptions on the sensitivity/specificity of $F$ for evaluating System 2 (a newer system), even in the absence of new labeled data, as discussed in Figure 1 and the surrounding part of the introduction.
>
> **(W1/W2/Q3) It is already well-known that the estimated control variate weights only work with enough data…the work focuses exclusively on the specialized estimator PPI++. I do not see the reason to restrict the analysis to a specific algorithm…How is this work connected to the control variate literature in machine learning?**
>
> You are correct that PPI++ is not a new idea, as we make clear in the related work section (citing papers going back to 1969).  We already note the connection to estimators in causality, and we will add the clarification that PPI++ can also be viewed as a control variate method, essentially the regression sampling method featured on the wikipedia page for control variates (https://en.wikipedia.org/wiki/Control_variates).  See our response to YTHP for additional comments on connections to other estimators in survey sampling.
>
> However, our work is original and significant in this context: Despite the fact that this estimator arises in several contexts, the non-asymptotic behavior of this estimator has not been studied to our knowledge, when accounting for estimation error in the value of lambda itself.
>
> Moreover, as we discuss in our response to jRz6 (their W1), our conclusions also apply (albeit in a more limited form) to general machine-learning applications of control variates / PPI++, where it is used to estimate the loss for training models via maximum likelihood / M-estimation.
>
> **(Q1/W3) “…I do not think the small sample setting is meaningful. Existing works in PPI aim at making a scientific discovery and often need hundreds of labeled data. This work does not show an application where small sample setting is relevant…”**
>
> We obviously have a difference of opinion - we think the small sample regime is worthy of study, and believe there are plenty of additional applications where PPI++ could be applied thoughtfully, even with limited samples.  Note that our results are not inherently negative for PPI++, but also enable practitioners to decide that PPI++ is warranted in a very small sample (e.g., ~10 samples) regime so long as they have faith in the quality of psuedo-labels.  Our results make precise not just the intuition that "estimated control variate weights only work with enough data", but gives tools for determining how much data is enough.
>
> We used the same scientific discovery datasets to make our experiments comparable to prior work, not because these are the only domains where techniques like PPI++ could be applied.  For an illustration of a setting with small sample sizes that motivates our interest (though not one where PPI++ was used directly), see e.g., https://www.nature.com/articles/s41746-025-02005-2 for a study that attempts to validate LLM-judges of AI-generated discharge summaries in a healthcare context, using ~40 samples.
>
> **(Q2) The cross-fit PPI++ splits the data evenly. It will also be interesting to explore other splitting ratios, or even adaptive ones. How will this modification affect the results?**
>
> That is a good question, and we can give some high-level intuition for how we would expect other splitting procedures to change the results.  First, our current 2-fold split uses the same amount of data in each iteration for (a) estimating lambda, and (b) estimating the mean.  The dependency between folds (the last term of Equation (2) in Theorem 4.2) arises due to the fact that the data used to estimate the mean in one fold is used to estimate lambda in the other.  There is a trade-off here, which we do not precisely characterize, but could be of interest for future work: When using (k-1) folds to estimate lambda, and the kth fold to perform mean estimation, we would expect this to decrease the error in estimating the optimal weight (decreasing the third “eff loss” term in Theorem 4.2), but increase the dependence between folds (since multiple folds are using overlapping data to estimate lambda).

---

> > ### Author Rebuttal · Reviewer_wXiH · 2026-04-01
> >
> > I would like to thank the authors for their detailed responses.
> > - The focus of using the methods as a "power analysis" tool is interesting. I am more convinced about the practicability now.
> > - Regarding the generality, it always feels limited to perform analysis on one specific variant of one specific algorithm in one specific subarea. For this work to be more impactful, I would expect a full literature review on control variates in machine learning, with insights how the analysis may help different areas.
> > - The example that has only 40 samples is eye-opening. I imagine scientists should be convinced to say "according to this research, and our limited resources, let's use PPI instead of PPI++ to draw our conclusions". Nevertheless, it will be more helpful to have one real example that is constrained by a small dataset in the experiments.
> >
> > I increase my score to 3 and will discuss the scope with the other reviewers.

---

> > > ### Author Response · Authors · 2026-04-07
> > >
> > > Thank you for the thoughtful questions and engagement in the rebuttal process, and of course we appreciate you raising your score.
> > >
> > > We will just make one last comment / observation on the third point you raised:
> > > >  Nevertheless, it will be more helpful to have one real example that is constrained by a small dataset in the experiments.
> > >
> > > In our experiments (and those of similar papers), there is a catch-22.  Even when a method is *motivated* by data-scarce settings, a *label-scarce real-world dataset* (e.g., with only 10-40 labels) would make a poor *experimental test-bed* for such a method.
> > >
> > > In short, these type of experiments require a "**ground truth**" value to estimate, in order to measure the quality of any estimation procedure, to measure coverage of confidence intervals, etc.  In practice, experiments often use the mean of a larger labeled dataset as this "ground truth" value, and then see how close one can get using a smaller amount of labeled data.  As a result, while we can point to real-world examples (like the one above) as *motivation* for our work, they do not make sense as datasets for *experimental results*.

---

### Official Review · Reviewer_jRz6 · 2026-03-10

**Soundness:** 3
**Presentation:** 3
**Significance:** 3
**Originality:** 3
**Overall Recommendation:** 4
**Confidence:** 4

**Summary:**

Prediction-powered inference (PPI) and its adaptive variant PPI++ have recently gained popularity for their ability to combine a small set of gold-standard labels with a large set of pseudo-labels to improve statistical estimation. Prior works claimed an asymptotic free lunch for PPI++, suggesting it always performs as well as or better than classical estimation regardless of pseudo-label quality. However, this guarantee is asymptotic, and PPI is most valuable when labeled data is scarce (small $n$), where finite-sample behavior matters most. This paper provides an exact finite-sample analysis of the variance and estimation error of PPI++ in the mean estimation setting.

**Compliance With Llm Reviewing Policy:**

Affirmed.

**Final Justification:**

I will decrease my score.

**Key Questions For Authors:**

# Questions

- **Q1:** How does the required correlation threshold scale if the unlabeled set size $N$ is finite but comparable to $n$? Does the efficiency loss term in Theorem 4.2 increase significantly?
- **Q2:** Based on your analysis, can you propose a practical diagnostic test (e.g., a minimum correlation estimate) that a practitioner should run on their pilot data before deciding to deploy PPI++?

**Limitations:**

yes

**Strengths And Weaknesses:**

# Strengths

- **S1:** The paper addresses a significant gap in the recent PPI literature. By challenging the always improves narrative with a rigorous non-asymptotic analysis, it provides a much-needed correction to the community's understanding of these tools.

- **S2:** The theoretical results are elegant and interpretable. The $1/\sqrt{n-2}$ rule for Gaussian data provides a clear "rule-of-thumb" that is immediately useful for practitioners.

- **S3:** The paper is exceptionally well-written. The winning method heatmap (Figure 1) is an excellent way to operationalize theoretical findings into practical guidance.

- **S4:** The experiments on AlphaFold and Galaxies datasets directly support the theoretical claims, showing real-world instances where PPI++ underperforms or undercovers.

# Weaknesses

- **W1:** The analysis is limited to the mean estimation problem. While PPI is used for more complex tasks (quantiles, GLMs), it remains unclear if the exact same no free lunch thresholds can be derived for these estimators, or if the "overhead" cost behaves differently there.

- **W2:** The general variance expression (Theorem 4.2) depends on higher-order moments. In very low-sample regimes (small $n$), estimating these moments to decide whether to use PPI++ might be just as noisy as the original mean estimation problem, potentially leading to a meta-estimation issue.

- **W3:** The paper largely assumes the unlabeled set size $N \to \infty$. While common, the trade-off between $n$ and finite $N$ where pseudo-label mean/variance are also noisy is less explored in the core no free lunch discussion.

---

> ### Author Rebuttal · Authors · 2026-03-31
>
> Thank you for the kind words and insightful questions, and we are glad that you enjoyed the “winning method heatmap” in Figure 1!   Regarding the perceived weaknesses and questions raised, we offer a few thoughts and answers below.
>
> **(W1) The analysis is limited to the mean estimation problem. While PPI is used for more complex tasks (quantiles, GLMs), it remains unclear if the exact same no free lunch thresholds can be derived for these estimators**
>
> An excellent observation - **for general convex M-estimation problems, the asymptotic claims of PPI / PPI++ still proceed by a “mean estimation variance reduction” argument**.  In particular, their main result (Theorem 1) proceeds in two steps:
> First, they argue similarly that their approach leads to reduced variance in estimation of the expected gradient (instead of considering Y and F, they use the per-sample gradients calculated using Y and F).
> Second, they link reduced variance in the estimate of the expected gradient to variance in the estimated parameter.
>
> The steps involved are essentially the same as those in standard proofs of asymptotic normality for M-estimators.  **As a result, our results would apply directly to a “no-free lunch” in that first stage of the argument**, with the obvious caveat that the linkage between “variance in mean estimation” and “variance in parameter estimation” is harder to characterize exactly in a non-asymptotic framework.  Mean estimation is unique in that the final estimator that solves the M-estimation problem has a closed form, but for more general M-estimation problems, the connection between variance of gradient estimation & variance of parameter estimates is much harder to characterize non-asymptotically.
>
> **(Q2/W2) The general variance expression (Theorem 4.2) depends on higher-order moments. In very low-sample regimes…estimating these moments to decide whether to use PPI++ might be just as noisy as the original mean estimation problem…can you propose a practical diagnostic test (e.g., a minimum correlation estimate) that a practitioner should run on their pilot data before deciding to deploy PPI++?**
>
> We generally do not recommend a procedure of “use pilot data to estimate correlations, and then decide whether or not to use PPI++,” since that inherits the same “no free lunch” challenge of any adaptive procedure, and would need to be compared to a procedure which used the pilot data for estimation and not just selection of PPI++ vs another method.
>
> In practice, we recommend sensitivity analysis, as discussed in the introduction, which in our setting is analogous to "power analysis" in many statistical studies, where future statistical performance (e.g., power) is determined on the basis of reasonable assumptions, and used to inform decision-making (e.g., around sample size to collect).
>
> In many cases, all of the higher-order moments are fully specified by a smaller set of interpretable parameters.  For instance, in the case of binary labels and psuedo-labels, they can all be specified by three parameters, including $P(F = 1)$ which can be estimated from abundant unlabeled data, leaving two remaining parameters to specify for sensitivity analysis.
>
> Sensitivity analysis can be informed on the basis of related datasets.  For instance, suppose $F$ is a hallucination detector, and we want to estimate the hallucination rate of a LLM-based system.  We may have prior labeled data from evaluating an older system (call it System 1), where $F$ had strong sensitivity/specificity.  We can then use that data to inform our assumptions on the sensitivity/specificity of $F$ for evaluating System 2 (a newer system), even in the absence of new labeled data.
>
> **(W3/Q1) Unlabeled sample size**
>
> We do have results on this point in the Gaussian setting, which we can add to the appendix in a revision, but do not have space to include a full proof in the rebuttal here.  When $N$ is finite, the condition for improvement becomes $|\rho_{fy}| \geq 1 / \sqrt{n - 2 - 8(n-1)/(N-1)}$, which simplifies (when $N$ is large) to the condition given in the main text for the Gaussian case.  Otherwise, the threshold for improvement is (intuitively) higher, since the benefit of the unlabeled data is diminished.
>
> Since we cannot provide a full proof here, we provide some intuition as to how that result can be derived, and will include all details in a revised version: As we mention in Footnote 2, the Gaussian setting is unique, where empirical covariances are independent of sample means.  In the Gaussian case, the quantity $\hat{\lambda}$ (based on the estimated covariance and the estimated variance of $F$) is independent of $\bar{f}_n - \bar{f}_N$, the difference in sample means, and we can compute terms like the expected value of $1 / \hat{\sigma\_f\}^2$ (the inverse empirical variance) using the fact that it follows an inverse chi-squared distribution.

---

> > ### Author Rebuttal · Reviewer_jRz6 · 2026-04-03
> >
> > I will keep my score.

---

### Official Review · Reviewer_YTHP · 2026-03-12

**Soundness:** 2
**Presentation:** 3
**Significance:** 3
**Originality:** 2
**Overall Recommendation:** 3
**Confidence:** 4

**Summary:**

This paper studies **PPI procedures** for estimating the expectation of Y. This family of methods aims to **reduce variance** by introducing a control variable (with zero expectation) constructed from pseudo-labels of Y on a larger sample (for example, generated using an LLM).

The paper first presents the PPI procedures, and then aims to analyze the quadratic risk by decomposing it into bias + variance. The variance is derived for the Gaussian case, and then extended to more general distributions.

In both settings, the goal is to identify a condition under which PPI actually improves the estimation (a *“free lunch” condition*). Finally, numerical experiments are conducted.

**Compliance With Llm Reviewing Policy:**

Affirmed.

**Final Justification:**

After reading the other rebuttals, I agree with the remarks of reviewer wXiH. In particular, the literature related to control variables is not sufficiently presented (it also exists in the non-asymptotic setting), which makes it difficult to assess the originality of the results.

**Key Questions For Authors:**

(W1) + ( W2)  and
(bonus question) What happens if $f$ is learned using the same labeled data?
I would be willing to increase my score depending on your response.

**Limitations:**

yes

**Strengths And Weaknesses:**

**Strengths**

(S1) Significance: The problem is very useful and interesting. Asking whether a “trendy” procedure is actually relevant reflects a genuine scientific mindset.

(S2) Presentation: The paper is well written and easy to follow,   particularly with the recurring focus on the "free lunch" question.


**Weaknesses**

(W1) Originality:  In my opinion, the connection with control variates in computational statistics is not sufficiently established (see, for example, the book by C. Robert or https://en.wikipedia.org/wiki/Control_variates). It is not clear to me that this type of estimator has not already been studied (for instance in **causality** or **computational statistics**).

(W2) Soundness: In my view, there are a few issues that prevent the paper from delivering a clear message:

1. Does PPI++  have a bias? Why does Section 4 focus only on the variance?
2. I find that the contribution is somewhat stretched Section 4.4.1 does not add much, and Lemma 4.1 is not very useful. Proposition 4.3 gives the bias; it would be helpful to present the result in terms of bias + variance to make the insight of this part clearer.

(W3) Code not available.

---

> ### Author Rebuttal · Authors · 2026-03-31
>
> We thank the reviewer for recognizing the importance of contextualizing the strengths and weaknesses of “trendy” methods. We are also glad the reviewer found the paper well-written and easy to follow.
>
> We now address the weaknesses and questions posed.
>
> **(W1) Originality: In my opinion, the connection with control variates in computational statistics is not sufficiently established (see, for example, the book by C. Robert or https://en.wikipedia.org/wiki/Control_variates). It is not clear to me that this type of estimator has not already been studied (for instance in causality or computational statistics).**
>
> You are correct that PPI++ is not a new idea, as we make clear in the related work section (citing papers going back to 1969).  We already note the connection to estimators in causality, and we will add the clarification that PPI++ can also be viewed as a control variate method.  In addition, PPI is equivalent to the difference estimator in survey sampling, and PPI++ can be viewed as an example of the GREG estimator (see https://arxiv.org/abs/2603.19160v1, published after we submitted this paper).  We will add these references to the revised version of the paper.
>
> However, our work remains original and significant in this context: Despite the fact that this estimator arises in several contexts, the non-asymptotic behavior of this estimator has **not** been studied to our knowledge, when accounting for estimation error in the value of lambda itself.
>
> **(W2) Soundness…Does PPI++ have a bias? Why does Section 4 focus only on the variance?**
>
> In Section 4.2, we focus only on variance because cross-fit PPI++ has zero bias due to cross-fitting (we will revise to make this point clearer, we show it in Appendix A.2 but it is a trivial consequence of cross-fitting).  In Section 4.3 we consider bias and variance because single-sample PPI++ *does* have bias in finite samples, though it vanishes asymptotically.  The Gaussian setting in Section 4.1 is a special case where neither estimator is biased.  We will revise to make all the above clearer.
>
> **(continued) Proposition 4.3 gives the bias; it would be helpful to present the result in terms of bias + variance to make the insight of this part clearer.**
>
> As we state above Theorem 4.2, the variance of single-sample PPI++ is far less tractable to analyze, and so it is not included.
>
> **(continued) Lemma 4.1 is not very useful:**
>
> On the contrary - Lemma 4.1, when combined with Theorem 4.2, makes up our main result!  As we state above Lemma 4.1
> > we characterize the error in covariance estimation in Lemma 4.1, which, when plugged into Theorem 4.2, yields an analytical expression for the variance of Cross-fit PPI++ in terms of terms involving the mean and covariance of Y, F and their higher-order moments
>
> See our discussion with xCfr on this point - we will revise the presentation to make this connection clearer.
>
> **(continued) Section 4.4.1 does not add much**
> There is no section in our paper that matches that numbering.  Could you clarify which section you are referring to?
>
> **(W3) code not available**
> We will make our experimental code available, though our contribution is primarily theoretical, and hence our main contribution can be checked given the existing materials.
>
> **(bonus question) What happens if is learned using the same labeled data?**
>
> An excellent question - Cross-Fit PPI++ itself can be viewed as PPI where a simple linear predictor function $g$ is learned from data with cross-fitting (i.e., where $g(x; \lambda) = \lambda \cdot F$, treating $F$ as a single scalar "feature"), so our results can be seen as giving intuition for the statistical trade-offs here between (a) the cost incurred to learn the function itself, and (b) the benefit of learning it well.  For more general predictors, the analysis would obviously be more complex - it may be possible (and an interesting direction for future work) to characterize the non-asymptotic behavior of e.g., linear predictors with more features, but it would be difficult to characterize for more general predictors.

---

> > ### Author Rebuttal · Reviewer_YTHP · 2026-04-02
> >
> > Thanks for precisions ! I maintain my score

---

> > > ### Author Response · Authors · 2026-04-07
> > >
> > > Thank you for your positive rebuttal acknowledgement.  We hope you'll consider increasing your score (if we have indeed addressed your concerns), given that you noted in your original review that *"I would be willing to increase my score depending on your response"*, and given that you seem to have selected option *"(a) Fully resolved - my concerns have been adequately addressed"* in the rebuttal acknowledgement.  Insofar as there are lingering concerns, we would appreciate any feedback you might be able to convey as part of the meta-review process, and we thank you for the feedback provided thus far.

---

### Official Review · Reviewer_xCfr · 2026-03-13

**Soundness:** 3
**Presentation:** 3
**Significance:** 2
**Originality:** 2
**Overall Recommendation:** 4
**Confidence:** 4

**Summary:**

This paper analyzed the variance of a well-known estimator, namely PPI++, in prediction-powered inference. Their results imply that there's no-free lunch in applying PPI++: there are regimes where the variance of PPI++ is larger than classical ones.

**Compliance With Llm Reviewing Policy:**

Affirmed.

**Final Justification:**

I appreciate authors' explanation on the issue of unlabeled dataset being infinite. The finite-sample result for the Gaussian setting seems nice, but I am still suspicious on its scope if nothing could say beyond the Gaussian setting. However, I would like to raise my score temporarily to 4 given the partial answer to infinite unlabelled dataset,

**Key Questions For Authors:**

1. What's the role of the covariate X if the goal is to only estimate the marginal (instead of conditional) expectation of Y? Can everything be stated without introducing X at all?
2. Is if PPI++ itself SOTA today? I am not sure if it is still worth studying properties of PPI++ alone if it is not SOTA.
3. PPI++ as in definition 3.5 uses a two-fold splitting. What about other splittings, like K-fold?
4. What's the definition of $\hat\sigma_{fy}$ in Theorem 4.2? Why is equation (2) correct when there's only one term with $\hat\sigma_{fy}$?

**Limitations:**

Would be better to include limitations of the current simplification.

**Strengths And Weaknesses:**

**Strengths:**
1. The perspective of studying PPI through non-asymptotic analysis seems novel.

**Weaknesses:**
1. Assuming the unlabeled dataset to be infinite size greatly simplifies the model and the analysis. Not sure if such simplification is necessary or practical. The asymptotic nature of it actually violates the claimed "non-asymptoticity" of the results.
2. The main result only argues the finite-sample variance for Crossfit-PPI++. Is this the most common variant of PPI++ in the literature?
3. I am not sure if the derived expressions of variances are useful in practice under this perhaps overly simplified model.
4. The results are mostly bias-variance decomposition (in particular, only equations are involved) and their analysis looks a bit too trivial.

---

> ### Author Rebuttal · Authors · 2026-03-31
>
> Thank you for the response, we will focus on your key questions below:
>
> Q1: **Can everything be stated without introducing X at all?**
>
> Yes, the results can be stated without $X$, just by reference to the random proxy-labels $f(X)$ (which we also denote $F$). The key (fixed) quantity of interest is the predictor / proxy label function $f$, and we only introduce $X$ insofar as these proxy-labels are functions $f(X)$, but generally treat $f$ as a random variable of its own.
>
> Q2:  **Is if PPI++ itself SOTA today? I am not sure if it is still worth studying properties of PPI++ alone if it is not SOTA.**
>
> We would argue that PPI++ and the ideas behind it are still extremely relevant.  As we note in our response to YTHP, the basic form of the PPI++ estimator shows up in a variety of domains, but the non-asymptotic behavior has not previously been studied.
>
> As regards the question of whether or not PPI++ is ‘SOTA’:  Our view would be that, in the domain of statistical methods, determining if something is ‘SOTA’ is not purely an empirical question, but also a theoretical one (in terms of what kind of guarantees can be achieved by different approaches).  PPI++ presents itself as having a benefit that might lead some to consider it the ‘state of the art’ approach to combining pseudo-labels with ground-truth labels (namely, that it ‘always improves’).  In that sense, our work helps inform consensus in the broader community regarding the benefits / trade-offs of PPI++.
>
> Q3:  **PPI++ as in definition 3.5 uses a two-fold splitting. What about other splittings, like K-fold?**
>
> That is a good question, and we can give some high-level intuition for how we would expect other K-fold splitting procedures to change the results.  First, our current 2-fold split uses the same amount of data in each iteration for (a) estimating $\lambda$, and (b) estimating the mean.  The dependency between folds (the last term of Equation (2) in Theorem 4.2) arises due to the fact that the data used to estimate the mean in one fold is used to estimate $\lambda$ in the other.
>
> Hence, there is a trade-off here, which we do not precisely characterize, but could be of interest for future work: When using (k-1) folds to estimate lambda, and the kth fold to perform mean estimation, we would expect this to decrease the error in estimating the optimal weight (decreasing the third “eff loss” term in Theorem 4.2), but increase the dependence between folds (since multiple folds are using overlapping data to estimate lambda).
>
> Q4: ***Theorem 4.2 / Equation (2)***
>
> The right way to read Equation (2) in Theorem 4.2 is to plug in Lemma 4.1 (which replaces the term you mention). We mention this (though it may be easy to miss in retrospect) above Lemma 4.1,
> > we characterize the error in covariance estimation in Lemma 4.1, which, when plugged into Theorem 4.2, yields an analytical expression for the variance of Cross-fit PPI++ in terms of terms involving the mean and covariance of Y, F and their higher-order moments
>
> We did not combine them directly in the paper because it felt more natural to introduce the general formulation, highlighting where the error in covariance estimation enters the picture, and then to “complete the picture” with Lemma 4.1.
>
> We now realize this structure (splitting the main result across Theorem 4.2 and Lemma 4.1) may have led to some confusion, and in the revision we will present the unified result as a stand-alone equation.

---

> > ### Author Rebuttal · Reviewer_xCfr · 2026-04-04
> >
> > I thank authors for their responses. However, my major concern about assuming infinite unlabeled dataset is not resolved. I will keep my score at this point.

---

> > > ### Author Response · Authors · 2026-04-07
> > >
> > > Thank you for the rebuttal acknowledgement.  Regarding the "infinite unlabeled dataset" assumption, we would first direct your attention to the response we gave to jRz6, which we rephrase and expand upon here.
> > >
> > > We do have results on the finite-sample variance of PPI++ in the Gaussian setting, which we can add to the appendix in a revision, but do not have space to include a full proof in the rebuttal here.
> > >
> > > The core result is that when $N$ is finite (using $N$ to denote the "unlabeled" data set), the condition for improvement becomes $|\rho\_{fy}| \geq 1 / \sqrt{n - 2 - 8(n-1)/(N-1)}$, which simplifies (when is $N$ large) to the condition given in the main text for the Gaussian case. Otherwise, the threshold for improvement is (intuitively) higher, since the benefit of the unlabeled data is diminished.
> > >
> > > Since we cannot provide a full proof here, we provide some intuition as to how that result can be derived, and will include all details in a revised version: As we mention in Footnote 2, the Gaussian setting is unique, where empirical covariances and variances are independent of sample means. In the Gaussian case, the quantity $\hat{\lambda}$ (based on the estimated covariance and the estimated variance of $F$) is independent of $\bar{f}\_n - \bar{f}\_N$, the difference in sample means, and we can compute terms like the expected value of $1 / \hat{\sigma}_{f}^2$ (the inverse empirical variance) using the fact that it follows an inverse chi-squared distribution.
> > >
> > > We believe this example is useful for building intuition along two dimension:
> > > 1. When $N$ is finite, then the required correlation for improvement is higher.
> > > 2. For moderate ratios of $(n-1) / (N-1)$, the results are highly similar (e.g., if there is roughly 8x as much unlabeled data as labeled data, then the condition becomes $1 / \sqrt{n - 3}$ as opposed to $1 / \sqrt{n-2}$.
> > >
> > > We will add a more extensive discussion on the "large $N$" assumption in future revisions of the paper.

---

### Decision · Program_Chairs · 2026-04-30

**Decision:**

Accept (regular)

**Comment:**

This submission investigates the recent framework of Predictive-Powered Inference in the context of (univariate) mean estimation, where a lot of cheap "pseudo-labelled" data comes cheaply, but labelled real data come at non-trivial cost. The starting point of this paper is the fact that prior PPI-type estimators have mostly been analyzed in the asymptotic regime where even the labelled dataset size is assumed to go to infinity. In such idealistic settings, the PPI approach has been shown to do well (with respect to only using the expensive labelled data) regardless of how good or bad the cheap pseudolabels are. The results of this paper demonstrate that, in the finite real-sample regime, bad pseudo-labels can indeed deteriorate performance, hence "no free lunch" in the paper title.

The main concerns from reviewers are due to (A) limited scope, and relatedly, (B) that the results of the paper are related to the control variates literature and the submission does not differentiated itself sufficiently from the control variates literature for the purposes of novelty.

I am recommending acceptance because (A) I do think PPI is a topic of relevance given the rise of LLMs and other deep learning models, and (at least according to related works section) used by application domains, so pointing out deficiencies or less-than-rosy pictures of the approach is a timely contribution, and (B) even if the techniques are from a related literature, the conceptual message of the work is still relevant and actionable for the community. Lastly, I do note that reviewers have not provided sufficient references to argue how the control variates literature might subsume this work, so I tend to give the authors benefit of doubt in this case.